# Effects of an urban sanitation intervention on childhood enteric infection and diarrhea in Maputo, Mozambique: A controlled before-and-after trial

Jackie Knee[1,2], Trent Sumner[2], Zaida Adriano[3], Claire Anderson[2], Farran Bush[4], Drew Capone[2,5], Veronica Casmo[6], David Holcomb[5,7], Pete Kolsky[5], Amy MacDougall[8], Evgeniya Molotkova[9], Judite Monteiro Braga[6], Celina Russo[2], Wolf Peter Schmidt[1], Jill Stewart[5], Winnie Zambrana[2], Valentina Zuin[10], Rassul Nalá[6], Oliver Cumming[1], Joe Brown[2,5]*

[1]London School of Hygiene & Tropical Medicine, Faculty of Infectious Tropical Diseases, Disease Control Department, London, United Kingdom; [2]Georgia Institute of Technology, School of Civil and Environmental Engineering, Atlanta, United States; [3]WE Consult ltd, Maputo, Mozambique; [4]Georgia Institute of Technology, School of Chemical and Biomolecular Engineering, Atlanta, United States; [5]University of North Carolina at Chapel Hill, Gillings School of Global Public Health, Department of Epidemiology, Chapel Hill, United States; [6]Instituto Nacional de Saúde, Maputo, Mozambique; [7]University of North Carolina at Chapel Hill, Gillings School of Global Public Health, Department of Environmental Sciences and Engineering, Chapel Hill, United States; [8]London School of Hygiene & Tropical Medicine, Faculty of Epidemiology and Population Health, Department of Medical Statistics, London, United Kingdom; [9]Georgia Institute of Technology, School of Biological Sciences, Atlanta, United States; [10]Yale-NUS College, Division of Social Science, Singapore, Singapore

*For correspondence:
joebrown@unc.edu

**Abstract** We conducted a controlled before-and-after trial to evaluate the impact of an onsite urban sanitation intervention on the prevalence of enteric infection, soil transmitted helminth reinfection, and diarrhea among children in Maputo, Mozambique. A non-governmental organization replaced existing poor-quality latrines with pour-flush toilets with septic tanks serving household clusters. We enrolled children aged 1–48 months at baseline and measured outcomes before and 12 and 24 months after the intervention, with concurrent measurement among children in a comparable control arm. Despite nearly exclusive use, we found no evidence that intervention affected the prevalence of any measured outcome after 12 or 24 months of exposure. Among children born into study sites after intervention, we observed a reduced prevalence of *Trichuris* and *Shigella* infection relative to the same age group at baseline (<2 years old). Protection from birth may be important to reduce exposure to and infection with enteric pathogens in this setting.

## Introduction

Rapid urbanization has led to the expansion of informal settlements in many low- and middle-income countries (LMICs). Such settlements often have very limited sanitation infrastructure (*UN-Habitat, 2016*). Separation of human waste from human contact can prevent exposure to enteric pathogens that cause infection, diarrhea (*Liu et al., 2016*), and potentially long-term health effects such as

environmental enteric dysfunction (EED) (*Kosek and MAL-ED Network Investigators, 2017*), linear growth deficits (*Rogawski et al., 2018*), impaired cognitive development (*MAL-ED Network Investigators, 2018*), and reduced oral vaccine immunogenicity (*Parker et al., 2018*). Children living in densely populated slum areas where fecal contamination is pervasive and sanitation infrastructure is limited may be at an increased risk of adverse health effects due to frequent exposure to enteric pathogens (*Ezeh et al., 2017*; *Fink et al., 2014*).

Household-level sewerage has demonstrated health benefits (*Barreto et al., 2010*; *Barreto et al., 2007*; *Norman et al., 2010*) and remains an important long-term goal for many urban settings despite limited evidence from controlled trials (*Norman et al., 2010*; *Wolf et al., 2018*). Such systems may not be feasible short-term solutions due to cost, space, and logistical constraints, challenges that have also impeded their evaluation via randomized trials (*Norman et al., 2010*). Further, in densely populated areas, there may not be space for household-level sanitation of any type. Shared sanitation is a subject of considerable debate but may represent the only near-term sanitation option in some settings (*Evans et al., 2017*; *Heijnen et al., 2014*; *Tidwell et al., 2020*). Yet, while shared, onsite systems may fill the growing need for safe sanitation in rapidly expanding urban areas in LMICs, to date, there has been little evidence of their health impacts in these settings. Recent large-scale, rigorous evaluations of onsite sanitation interventions and combined water, sanitation, and hygiene (WASH) interventions have demonstrated mixed effects on health (*Clasen et al., 2014*; *Sanitation Hygiene Infant Nutrition Efficacy (SHINE) Trial Team et al., 2019*; *Luby et al., 2018*; *Null et al., 2018*; *Patil et al., 2014*; *Pickering et al., 2015*) but all were conducted in rural areas with household-level interventions, and their findings may have limited generalizability to urban areas. A recent meta-analysis estimated that non-sewered interventions reduced the risk of self-reported diarrhea by 16% but did not estimate effects on objective health outcomes, such as enteric infection (*Brown and Cumming, 2020*), and could not stratify estimates by rural versus urban setting given the lack of evidence in urban areas (*Wolf et al., 2018*). To date, no controlled trials of urban onsite sanitation have been conducted despite over 740 million urban residents relying on such technologies (*Berendes et al., 2017*).

The Maputo Sanitation (MapSan) trial was the first controlled trial to evaluate an onsite, shared sanitation intervention in an urban setting and the first to use the prevalence of enteric infection, as detected by molecular methods, as the primary study outcome (*Brown et al., 2015*). The study was located in densely populated, low-income, informal neighborhoods of Maputo, Mozambique where the sanitary conditions are poor and disease burden high (*Knee et al., 2018*). As of 2017, only half of urban residents in Mozambique had access to at least basic sanitation infrastructure, 3% had access to sewerage, and 9% shared sanitation with multiple households, often in poor neighborhoods where space and resources are limited (*UNICEF/WHO, 2019*). We investigated whether an engineered, onsite, shared sanitation intervention could reduce enteric infection and diarrhea in young children living in these low-income, densely populated neighborhoods in Maputo, Mozambique.

## Results

The MapSan trial was a controlled before-and-after (CBA) trial designed to evaluate the impact of an onsite sanitation intervention on child health after 12 and 24 months of follow-up. The intervention consisted of pour-flush toilets to septic tanks with soakaway pits to discharge the liquid portion of the waste. A non-governmental organization (NGO) delivered the intervention to clusters of households known as compounds, replacing the existing poor-condition shared facilities. Control compounds did not receive the intervention and continued to use their poor-condition sanitation for the duration of the study. We assessed several measures of child health, including enteric infection measured via stool-based molecular methods, soil-transmitted helminth (STH) re-infection measured via Kato-Katz, and diarrhea measured via caregiver report in both intervention and control children during three phases: baseline (pre-intervention), 12-month follow-up, and 24-month follow-up. Children were eligible for baseline enrollment if they were less than 4 years old (1–48 months old). At follow-up, children were eligible for enrollment if they were less than 4 years old or if they would have been less than 4 years old during baseline.

We enrolled 987 children in 495 compounds during the baseline phase (February 2015 – February 2016) and collected stool samples (whole stool or diaper samples containing liquid diarrhea) from

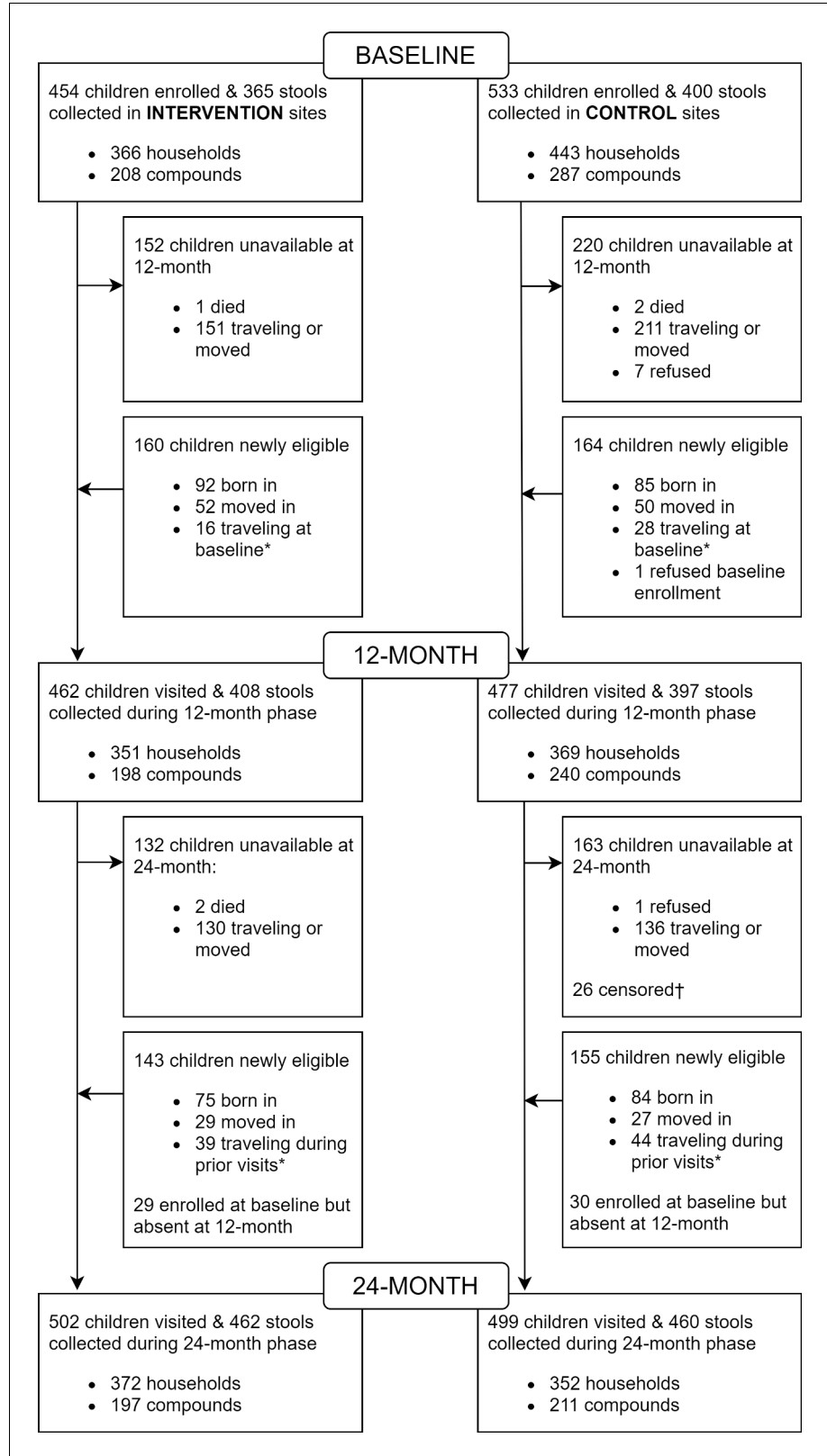

**Figure 1.** Trial profile. *Eligible for enrollment at baseline and/or 12 months but traveling at time of visit. †Children removed from 24-month analysis because their compound received an intervention after completion of the baseline phase. Source files available in *Figure 1—source data 1* and *Figure 1—source code 1*.

*Figure 1 continued on next page*

*Figure 1 continued*

The online version of this article includes the following source data and source code for figure 1:

**Source code 1.** Trial profile.
**Source data 1.** Trial profile.

765 children (78%) (*Figure 1*). During the 12-month follow-up phase (March 2016 – April 2017), we enrolled or revisited 939 children in 438 compounds and collected 805 stool samples (86%). During the 24-month follow-up phase (April 2017 – August 2018), we enrolled or revisited 1001 children in 408 compounds and collected stool samples from 922 (90%). To improve the success rate of stool sample collection during the 12- and 24-month follow-up visits, we collected rectal swabs from children who did not provide a whole stool sample after multiple collection attempts. The proportion of each type of sample (whole stool, diaper sample, and rectal swab) was similar between arms at each phase (*Appendix 1—figure 1*). Fewer than 5% of all samples were diapers and approximately 7% of 12-month samples and 25% of 24-month samples were rectal swabs (*Appendix 1—table 1*). The NGO delivered interventions to 15 control compounds after baseline and children in those compounds were censored at the time of intervention receipt (*Figure 1*). Children living in control compounds that independently upgraded their latrines were included in the main analyses. However, as inclusion of these control children may have diluted the intervention effect, they were excluded from sensitivity analyses designed to understand the impact of the intervention when compared with controls served by poor-condition sanitation throughout the study. Children in intervention and control compounds were enrolled at similar rates during each phase (*Appendix 1—figure 2*). Due to migration out of the compound, we collected longitudinal data from 62% of children (59% controls, 67% interventions) between baseline and 12-month and 51% of children (46% controls, 58% interventions) between baseline and 24-month.

At baseline enrollment, intervention compounds had more residents, households, and on-premise water taps than controls, though the number of shared latrines was similar (*Table 1*). Animals were observed in over half of all compounds. Intervention and control households had similar wealth scores, though intervention households had more members and were more crowded while control households more often had walls made of sturdy materials. All households used a municipal water tap as their primary drinking water source with 78% reporting use of a tap on the compound grounds. At baseline, latrines used by intervention households more often had pedestals or slabs, drop-hole covers, and sturdy walls compared with controls. Consistent with previous estimates in urban Maputo (*Satterthwaite et al., 2019*), open defecation was rare in our study population with only one control household reporting open defecation at baseline. Baseline characteristics of intervention and control children were similar: the average age at enrollment was 23 months (SD = 13), 51% were female, and 32% were still breastfeeding (*Table 1*). The age distributions of intervention and control children were similar at baseline and both follow-up phases (*Appendix 1—figure 3*).

We used the Luminex Gastrointestinal Pathogen Panel (GPP), a qualitative multiplex molecular assay, to simultaneously test for 15 enteric pathogens in stool samples, including nine bacteria, three protozoa, and three viruses. We detected ≥1 bacterial or protozoan enteric infection, our predefined primary outcome, in 78% (591/753) of children with stools available at baseline. We measured our pre-defined secondary outcome, ≥1 STH re-infection, using the Kato-Katz microscope method and detected ≥1 STH in 45% (308/698) of stools at baseline. The prevalences of pre-defined outcomes, individual pathogens, and pathogen types were similar between the intervention and control arms at baseline (*Table 2*). The prevalence of most bacterial, protozoan, and STH infections increased with age while the prevalence of enteric viruses decreased with age (*Appendix 1—table 2* and *Appendix 1—figure 4*).

The characteristics of children with repeated observations (including baseline) were similar to characteristics of children measured at baseline only (*Appendix 1—table 3* and *Appendix 1—table 4*) and to characteristics of children measured at 12 month and/or 24 month only with the exception of age-related characteristics (*Appendix 1—table 5* and *Appendix 1—table 6*). Over half of the children enrolled after baseline were born into study sites (336/622 [54%], *Figure 1*).

**Table 1.** Baseline characteristics of enrolled children, households, and compounds.

| | Control | | Intervention | |
|---|---|---|---|---|
| | N | n (%) or mean (SD) | N | n (%) or mean (SD) |
| **Child level variables** | | | | |
| Age at survey, days* | 520 | 700 (405) | 441 | 694 (403) |
| Sex, female | 520 | 266 (51%) | 444 | 227 (51%) |
| Child is breastfed with or without complementary feeding | 526 | 169 (32%) | 448 | 143 (32%) |
| Child is exclusively breastfed | 526 | 49 (9.3%) | 448 | 37 (8.3%) |
| Child feces reported to be disposed of in a latrine | 526 | 148 (28%) | 448 | 141 (31%) |
| Child wears diapers | 526 | 342 (65%) | 447 | 294 (66%) |
| Caregiver completed primary school | 528 | 287 (54%) | 451 | 239 (53%) |
| Child's mother is alive | 513 | 503 (98%) | 435 | 426 (98%) |
| Respondent is child's mother | 519 | 368 (71%) | 443 | 284 (64%) |
| **Household level variables** | | | | |
| Household population | 441 | 5.4 (2.4) | 365 | 6.1 (3.0) |
| Household wealth score, 0 (poorer) - 1 (wealthier)[†] | 440 | 0.45 (0.10) | 365 | 0.44 (0.10) |
| Household crowding, >3 persons/room | 440 | 54 (12%) | 365 | 60 (17%) |
| Household floor is covered[‡] | 440 | 426 (97%) | 365 | 333 (91%) |
| Household wall made of bricks, concrete, or similar[‡] | 440 | 304 (69%) | 365 | 215 (59%) |
| Household drinking water source inside compound | 435 | 324 (74%) | 360 | 294 (82%) |
| Latrine used by household has a ceramic or masonry pedestal[‡] | 432 | 153 (35%) | 359 | 142 (40%) |
| Latrine used by household has a drop-hole cover[‡] | 434 | 232 (53%) | 359 | 224 (62%) |
| **Compound level variables** | | | | |
| Number of compound members | 287 | 14 (6.2) | 208 | 19 (12) |
| Number of households | 287 | 3.8 (2.1) | 208 | 4.4 (3.7) |
| Number of water taps in compound | 283 | 0.98 (0.95) | 207 | 1.4 (1.6) |
| Number of latrines in compound | 287 | 1.0 (0.20) | 207 | 1.1 (0.57) |
| Number of people sharing a latrine | 285 | 14 (6.2) | 197 | 17 (8.9) |
| Number of households sharing a latrine | 285 | 3.7 (1.8) | 197 | 4.0 (2.8) |
| Latrine walls made of brick, concrete or similar[‡] | 282 | 72 (26%) | 204 | 67 (33%) |
| Compound population density, persons/square meter[§] | 281 | 0.071 (0.04) | 205 | 0.087 (0.05) |
| Compound has electricity that normally functions | 287 | 251 (87%) | 208 | 189 (91%) |
| Compound is prone to flooding | 287 | 184 (64%) | 208 | 120 (58%) |
| Any animals observed in compound[‡] | 287 | 170 (59%) | 208 | 132 (63%) |
| Dog(s) observed[‡] | 287 | 14 (4.9%) | 208 | 14 (6.7%) |
| Chicken(s) or duck(s) observed[‡] | 287 | 40 (14%) | 208 | 30 (14%) |
| Cat(s) observed[‡] | 287 | 149 (52%) | 208 | 116 (56%) |

Data are n (%) or mean (standard deviation) and collected by questionnaire unless otherwise noted.

* Age range 32–1819 days, IQR 339–1021 days. Age distributions available in ***Appendix 1—figure 3***.

[†]Assessed using Simple Poverty Scorecard for Mozambique (http://www.simplepovertyscorecard.com/MOZ_2008_ENG.pdf).

[‡]Data collected by direct observation.

[§]Calculated as # of people living in the compound divided by the area of the compound in square meters. Source files available in ***Table 1—source data 1*** and ***Table 1—source code 1***.

The online version of this article includes the following source data for Table 1:

**Source code 1.** Baseline characteristics.

**Source data 1.** Baseline characteristics.

**Table 2.** Effect of the intervention on bacterial, protozoan, viral, and STH infection and diarrhea at 12 and 24 months post-intervention.

| | Prevalence | | | 12-month Prevalence ratio (95% CI), p-value * | | 24-month Prevalence ratio (95% CI), p-value[†] | |
|---|---|---|---|---|---|---|---|
| | Baseline | 12 month | 24 month | Unadjusted | Adjusted[§] | Unadjusted | Adjusted[§] |
| **Any bacterial or protozoan infection‡** | | | | | | | |
| Control | 313/392 (80%) | 334/395 (85%) | 403/459 (88%) | .. | .. | .. | .. |
| Intervention | 278/361 (77%) | 347/408 (85%) | 392/462 (85%) | 1.04 (0.94–1.15), p=0.41 | 1.04 (0.94–1.15), p=0.41 | 1.00 (0.91–1.10), p=1.0 | 0.99 (0.91–1.09), p=0.89 |
| **Any STH infection‡** | | | | | | | |
| Control | 170/360 (47%) | 143/283 (51%) | 142/253 (56%) | .. | .. | .. | .. |
| Intervention | 138/329 (42%) | 150/305 (49%) | 136/292 (47%) | 1.12 (0.89–1.40), p=0.33 | 1.11 (0.89–1.38), p=0.35 | 0.94 (0.75–1.17), p=0.59 | 0.95 (0.77–1.17), p=0.62 |
| **Diarrhea‡** | | | | | | | |
| Control | 67/526 (13%) | 40/430 (9.3%) | 53/390 (14%) | .. | .. | .. | .. |
| Intervention | 59/448 (13%) | 59/436 (14%) | 53/410 (13%) | 1.41 (0.80–2.48), p=0.24 | 1.69 (0.89–3.21), p=0.11 | 0.92 (0.55–1.54), p=0.76 | 0.84 (0.47–1.51), p=0.56 |
| **Any bacteria** | | | | | | | |
| Control | 271/392 (69%) | 285/395 (72%) | 345/459 (75%) | .. | .. | .. | .. |
| Intervention | 227/361 (63%) | 292/408 (72%) | 324/462 (70%) | 1.09 (0.95–1.25), p=0.25 | 1.09 (0.95–1.26), p=0.20 | 1.03 (0.90–1.18), p=0.69 | 1.00 (0.87–1.15), p=0.96 |
| ***Shigella*** | | | | | | | |
| Control | 179/392 (46%) | 204/395 (52%) | 269/459 (59%) | .. | .. | .. | .. |
| Intervention | 152/361 (42%) | 218/408 (53%) | 245/462 (53%) | 1.13 (0.91–1.39), p=0.28 | 1.12 (0.92–1.38), p=0.27 | 0.98 (0.80–1.20), p=0.86 | 0.95 (0.79–1.16), p=0.64 |
| **ETEC** | | | | | | | |
| Control | 116/392 (30%) | 142/395 (36%) | 127/459 (28%) | .. | .. | .. | .. |
| Intervention | 110/361 (30%) | 143/408 (35%) | 126/462 (27%) | 0.93 (0.68–1.28), p=0.66 | 0.96 (0.69–1.33), p=0.81 | 0.95 (0.67–1.35), p=0.77 | 0.83 (0.57–1.19), p=0.31 |
| ***Campylobacter*** | | | | | | | |
| Control | 39/392 (9.9%) | 32/395 (8.1%) | 48/459 (10%) | .. | .. | .. | .. |
| Intervention | 21/361 (5.8%) | 35/408 (8.6%) | 34/462 (7.4%) | 1.78 (0.89–3.56), p=0.10 | 1.68 (0.82–3.45), p=0.16 | 1.20 (0.60–2.39), p=0.60 | 1.28 (0.62–2.62), 0.50 |
| ***C. difficile*** | | | | | | | |
| Control | 22/392 (5.6%) | 13/395 (3.3%) | 13/459 (2.8%) | .. | .. | .. | .. |
| Intervention | 13/361 (3.6%) | 17/408 (4.2%) | 11/462 (2.4%) | 1.95 (0.71–5.35), p=0.20 | 2.09 (0.77–5.64), p=0.15 | 1.32 (0.47–3.73), p=0.60 | 1.41 (0.46–4.30), p=0.54 |
| ***E. coli* O157** | | | | | | | |
| Control | 13/392 (3.3%) | 19/395 (4.8%) | 25/459 (5.5%) | .. | .. | .. | .. |
| Intervention | 18/361 (5.0%) | 14/408 (3.4%) | 16/462 (3.5%) | 0.48 (0.18–1.27), p=0.14 | 0.46 (0.18–1.21), p=0.12 | 0.43 (0.15–1.29), p=0.13 | 0.52 (0.17–1.59), p=0.25 |
| **STEC** | | | | | | | |
| Control | 3/392 (0.77%) | 9/395 (2.3%) | 17/459 (3.7%) | .. | .. | .. | .. |

*Table 2 continued on next page*

*Table 2 continued*

| | Prevalence | | | 12-month Prevalence ratio (95% CI), p-value * | | 24-month Prevalence ratio (95% CI), p-value[†] | |
|---|---|---|---|---|---|---|---|
| | Baseline | 12 month | 24 month | Unadjusted | Adjusted[§] | Unadjusted | Adjusted[§] |
| Intervention | 10/361 (2.8%) | 5/408 (1.2%) | 15/462 (3.3%) | 0.14 (0.03–0.67), p=0.014 | 0.15 (0.03–0.70), p=0.016 | 0.23 (0.05–1.03), p=0.055 | 0.24 (0.05–1.01), p=0.052 |
| **Any protozoa** | | | | | | | |
| Control | 205/392 (52%) | 236/395 (60%) | 303/459 (66%) | .. | .. | .. | .. |
| Intervention | 195/361 (54%) | 259/408 (63%) | 296/462 (64%) | 1.04 (0.87–1.24), p=0.69 | 1.03 (0.86–1.22), p=0.76 | 0.93 (0.78–1.11), p=0.40 | 0.91 (0.76–1.09), p=0.29 |
| *Giardia* | | | | | | | |
| Control | 201/392 (51%) | 230/395 (58%) | 294/459 (64%) | .. | .. | .. | .. |
| Intervention | 186/361 (52%) | 251/408 (62%) | 289/462 (63%) | 1.06 (0.88–1.27), p=0.55 | 1.05 (0.88–1.25), p=0.58 | 0.96 (0.80–1.14), p=0.61 | 0.93 (0.78–1.11), p=0.44 |
| *Cryptosporidium* | | | | | | | |
| Control | 8/392 (2%) | 8/395 (2%) | 14/459 (3.0%) | .. | .. | .. | .. |
| Intervention | 16/361 (4.4%) | 15/408 (3.7%) | 15/462 (3.3%) | 0.89 (0.23–3.43), p=0.87 | 0.89 (0.24–3.31), p=0.86 | 0.46 (0.11–1.93), p=0.29 | 0.53 (0.13–2.14), p=0.37 |
| **Any virus** | | | | | | | |
| Control | 53/392 (14%) | 52/395 (13%) | 59/459 (13%) | .. | .. | .. | .. |
| Intervention | 52/361 (14%) | 45/408 (11%) | 62/462 (13%) | 0.77 (0.45–1.32), p=0.35 | 0.75 (0.44–1.27), p=0.29 | 0.96 (0.55–1.68), p=0.88 | 1.03 (0.57–1.86), p=0.92 |
| **Norovirus GI/GII** | | | | | | | |
| Control | 38/392 (9.7%) | 44/395 (11%) | 47/459 (10%) | .. | .. | .. | .. |
| Intervention | 39/361 (11%) | 37/408 (9.1%) | 55/462 (12%) | 0.71 (0.38–1.33), p=0.28 | 0.68 (0.36–1.27), p=0.23 | 1.00 (0.52–1.93), p=0.99 | 1.10 (0.55–2.18), p=0.79 |
| **Adenovirus 40/41** | | | | | | | |
| Control | 13/392 (3.3%) | 9/395 (2.3%) | 7/459 (1.5%) | .. | .. | .. | .. |
| Intervention | 9/361 (2.5%) | 9/408 (2.2%) | 6/462 (1.3%) | 1.34 (0.34–5.23), p=0.68 | 1.24 (0.32–4.83), p=0.76 | 1.18 (0.23–5.98), p=0.84 | 0.97 (0.18–5.19), p=0.97 |
| **Coinfection, ≥2 GPP pathogens** | | | | | | | |
| Control | 206/392 (53%) | 237/395 (60%) | 302/459 (66%) | .. | .. | .. | .. |
| Intervention | 185/361 (51%) | 257/408 (63%) | 282/462 (61%) | 1.08 (0.90–1.29), p=0.39 | 1.08 (0.91–1.29), p=0.37 | 0.95 (0.80–1.12), p=0.54 | 0.93 (0.79–1.10), p=0.41 |
| *Trichuris* | | | | | | | |
| Control | 139/360 (39%) | 116/283 (41%) | 124/253 (49%) | .. | .. | .. | .. |
| Intervention | 117/329 (36%) | 120/305 (39%) | 117/292 (40%) | 1.05 (0.82–1.35), p=0.68 | 1.01 (0.79–1.28), p=0.96 | 0.89 (0.69–1.16), p=0.40 | 0.86 (0.67–1.10), p=0.22 |
| *Ascaris* | | | | | | | |
| Control | 95/360 (26%) | 82/283 (29%) | 78/253 (31%) | .. | .. | .. | .. |
| Intervention | 68/329 (21%) | 87/305 (29%) | 56/292 (19%) | 1.26 (0.87–1.82), p=0.22 | 1.33 (0.92–1.93), p=0.13 | 0.80 (0.52–1.21), p=29 | 0.83 (0.54–1.27), p=0.39 |
| **Coinfection, ≥2 STH** | | | | | | | |
| Control | 64/360 (18%) | 55/283 (19%) | 60/253 (24%) | .. | .. | .. | .. |
| Intervention | 47/329 (14%) | 57/305 (19%) | 37/292 (13%) | 1.16 (0.76–1.77), p=0.50 | 1.17 (0.76–1.79), p=0.49 | 0.67 (0.40–1.13), p=0.13 | 0.63 (0.37–1.07), p=0.084 |

Prevalence results are presented as (n/N (%)). All effect estimates are presented as prevalence ratios (ratio of ratios) and estimated using generalized estimating equations to fit Poisson regression models with robust standard errors.

*Analysis includes all children measured at baseline and 12-month visits.

†Analysis includes all children measured at baseline and 24 month visits.

‡Outcome was pre-specified in trial registration. All other outcomes are exploratory.

§Pathogen outcomes adjusted for child age and sex, caregiver's education, and household wealth index. Reported diarrhea was also adjusted for baseline presence of a drop-hole cover and reported use of a tap on compound grounds as primary drinking water source. Sample sizes for adjusted analyses are slightly smaller than numbers presented in prevalence estimates due to missing covariate data. *Y. enterocolitica, V. cholerae, E. histolytica*, and rotavirus were detected in <2% of samples in each arm at each phase. Descriptive data for these pathogens are available in the **Appendix 1—table 2**. Source files available in **Table 2—source data 1** and **Table 2—source code 1**.

The online version of this article includes the following source data for Table 2:

**Source code 1.** Intervention effect at 12 and 24 months.

**Source data 1.** Intervention effect at 12 and 24 months.

Our main analyses included observations from all eligible children enrolled at baseline (mean sampling age 664 days, SD = 393) and the 12-month (940 days, SD = 498) and 24-month (1137 days, SD = 603) follow-up visits (**Table 2**). We used a difference-in-difference (DID) analysis to estimate the intervention effect and adjust for baseline differences between intervention and control compounds. We present effect estimates from the DID analyses as prevalence ratios (ratio of ratios). To assess the validity of the parallel trend assumption, a key assumption of DID analyses, we ran 'placebo tests' by replacing outcomes with variables unrelated to the intervention, such as child age, respondent role, and presence of animals. Placebo tests showed no effect of the intervention on these variables, suggesting the parallel trend assumption was valid. We found no evidence that the intervention had an effect on the prevalence of any bacterial or protozoan infection (adjusted PR 1.04, 95% CI [0.94–1.15]), or any STH re-infection (1.11 [0.89–1.38]) 12 months after implementation (**Table 2**) despite household respondents reporting almost exclusive use of the intervention latrine (97%, 404/417). The prevalence of diarrhea remained fairly constant in both arms in all three phases with the exception of the 12-month measure in the control arm which was lower, resulting in a larger effect estimate with low precision (1.69 [0.89–3.21]).

The intervention had no meaningful effect at 12 months on the prevalence of infection with any of the three pathogen types measured by the GPP (bacterial, protozoan, viral), pathogen coinfection, or on any individual pathogen (**Table 2**). There was poor precision in the effect estimates for infrequently detected pathogens, evident from their wide confidence intervals. Therefore, some estimates suggestive of a large protective or detrimental effect (*Campylobacter, C. difficile, E. coli* O157, STEC, Norovirus GI/GII, Adenovirus 40/41) may have arisen by chance. While the National Deworming Campaign (NDC) provided albendazole to all compound members following baseline, during 12-month visitation only 58% of caregivers (56% control, 60% intervention) confirmed that their child was dewormed during these visits. A sensitivity analysis restricted to children confirmed to have been dewormed produced similar results to the main analysis (**Appendix 1—table 7**). By the 12-month visit, 19 control compounds (19/240 [8.0%]) had independently upgraded their facilities to pour-flush toilets. Results from sensitivity analyses excluding children living in control compounds with independently upgraded facilities were consistent with the main results (**Appendix 1—table 8**).

There was no evidence that the intervention had an effect on the prevalence of any bacterial or protozoan infection, any STH re-infection, or diarrhea after 24 months among all enrolled children (**Table 2**). We also found limited evidence of effect on the prevalence of any pathogen type or coinfection with ≥2 GPP pathogens 24 months after intervention. Results for several individual outcomes were suggestive of a protective (STEC, *E. coli* O157, *Cryptosporidium*, STH coinfection) or adverse (*Campylobacter, C. difficile*) effect, but evidence was weak as estimates were accompanied by wide confidence intervals and chance discoveries were possible given multiple comparisons. At the 24-month visits, caregivers confirmed baseline and/or 12-month deworming more frequently for intervention children (339/502 [68%]) than for control children (286/499 [57%]). Adjustment for deworming status or time since deworming had no impact on effect estimates (**Appendix 1—table 7**). Excluding children from control compounds which independently upgraded their facilities by the 24-month visit (35/211 compounds, [17%]) did not impact the results (**Appendix 1—table 8**).

**Table 3.** Effect of intervention on bacterial, protozoan, viral, and STH infection and reported diarrhea in children born into study sites post-intervention (post-baseline) but by 24-month visit compared with children of a similar age at baseline (<2 years old).

| | Prevalence (<2 years old) | | Prevalence ratio (95% CI), p-value | |
| --- | --- | --- | --- | --- |
| | Baseline | 24 month, Born-in | Unadjusted | Adjusted[†] |
| **Any bacterial or protozoan infection*** | | | | |
| Control | 158/228 (69%) | 79/106 (75%) | .. | .. |
| Intervention | 129/201 (64%) | 71/107 (66%) | 0.96 (0.77–1.21), p=0.74 | 0.99 (0.80–1.22), p=0.92 |
| **Any STH infection*** | | | | |
| Control | 67/205 (33%) | 25/68 (37%) | .. | .. |
| Intervention | 52/183 (28%) | 13/75 (17%) | 0.52 (0.26–1.05), p=0.069 | 0.51 (0.27–0.95), p=0.035 |
| **Diarrhea*** | | | | |
| Control | 46/283 (16%) | 18/105 (17%) | .. | .. |
| Intervention | 43/238 (18%) | 22/100 (22%) | 1.20 (0.57–2.5), p=0.64 | 1.37 (0.47–4.03), p=0.57 |
| **Any bacteria** | | | | |
| Control | 142/228 (62%) | 70/106 (66%) | .. | .. |
| Intervention | 102/201 (51%) | 51/107 (48%) | 0.89 (0.66–1.20), p=0.44 | 0.90 (0.67–1.19), p=0.45 |
| ***Shigella*** | | | | |
| Control | 67/228 (29%) | 36/106 (34%) | .. | .. |
| Intervention | 49/201 (24%) | 15/107 (14%) | 0.48 (0.28–0.83), p=0.009 | 0.49 (0.28–0.85), p=0.011 |
| **ETEC** | | | | |
| Control | 70/228 (31%) | 30/106 (28%) | .. | .. |
| Intervention | 58/201 (29%) | 24/107 (22%) | 0.84 (0.46–1.52), p=0.56 | 0.85 (0.48–1.51), p=0.58 |
| ***Campylobacter*** | | | | |
| Control | 27/228 (12%) | 14/106 (13%) | .. | .. |
| Intervention | 14/201 (7%) | 13/107 (12%) | 1.75 (0.63–4.87), p=0.29 | 1.75 (0.61–4.98), p=0.30 |
| ***C. difficile*** | | | | |
| Control | 20/228 (8.8%) | 7/106 (6.6%) | .. | .. |
| Intervention | 13/201 (6.5%) | 7/107 (6.5%) | 1.33 (0.36–4.86), p=0.67 | 1.49 (0.41–5.44), p=0.55 |
| ***E. coli* O157** | | | | |
| Control | 7/228 (3.1%) | 3/106 (2.8%) | .. | .. |
| Intervention | 9/201 (4.5%) | 2/107 (1.9%) | 0.45 (0.06–3.66), p=0.46 | 0.53 (0.07–4.24), p=0.55 |
| **STEC** | | | | |
| Control | 1/228 (0.44%) | 2/106 (1.9%) | .. | .. |
| Intervention | 9/201 (4.5%) | 1/107 (0.93%) | 0.05 (0.00–1.13), p=0.059 | 0.05 (0.00–1.26), p=0.070 |
| **Any protozoa** | | | | |
| Control | 82/228 (36%) | 47/106 (44%) | .. | .. |
| Intervention | 74/201 (37%) | 43/107 (40%) | 0.84 (0.55–1.28), p=0.42 | 0.90 (0.62–1.30), p=0.58 |
| ***Giardia*** | | | | |
| Control | 79/228 (35%) | 44/106 (42%) | .. | .. |
| Intervention | 68/201 (34%) | 41/107 (38%) | 0.90 (0.58–1.39), p=0.63 | 0.93 (0.64–1.36), p=0.70 |
| ***Cryptosporidium*** | | | | |
| Control | 7/228 (3.1%) | 5/106 (4.7%) | .. | .. |
| Intervention | 12/201 (6%) | 5/107 (4.7%) | 0.45 (0.08–2.57), p=0.37 | 0.64 (0.12–3.51), p=0.61 |
| **Any virus** | | | | |
| Control | 34/228 (15%) | 18/106 (17%) | .. | .. |
| Intervention | 36/201 (18%) | 18/107 (17%) | 0.83 (0.37–1.83), p=0.64 | 0.83 (0.37–1.87), p=0.66 |
| **Norovirus GI/GII** | | | | |

*Table 3 continued on next page*

*Table 3 continued*

| | Prevalence (<2 years old) | | Prevalence ratio (95% CI), p-value | |
|---|---|---|---|---|
| | Baseline | 24 month, Born-in | Unadjusted | Adjusted[†] |
| Control | 26/228 (11%) | 12/106 (11%) | .. | .. |
| Intervention | 26/201 (13%) | 17/107 (16%) | 1.24 (0.48–3.17), p=0.66 | 1.29 (0.49–3.41), p=0.61 |
| **Adenovirus 40/41** | | | | |
| Control | 7/228 (3.1%) | 4/106 (3.8%) | .. | .. |
| Intervention | 7/201 (3.5%) | 0/107 (0.0%) | ..[§] | ..[§] |
| **Coinfection, ≥2 GPP pathogens** | | | | |
| Control | 92/228 (40%) | 52/106 (49%) | .. | .. |
| Intervention | 74/201 (37%) | 39/107 (36%) | 0.82 (0.56–1.21), p=0.33 | 0.86 (0.59–1.24), p=0.41 |
| *Trichuris* | | | | |
| Control | 48/205 (23%) | 18/68 (26%) | .. | .. |
| Intervention | 41/183 (22%) | 5/75 (6.7%) | 0.25 (0.09–0.68), p=0.006 | 0.24 (0.10–0.60), p=0.002 |
| *Ascaris* | | | | |
| Control | 45/205 (22%) | 16/68 (24%) | .. | .. |
| Intervention | 29/183 (16%) | 9/75 (12%) | 0.70 (0.30–1.64), p=0.42 | 0.68 (0.30–1.54), p=0.36 |
| **Coinfection, ≥2 STH** | | | | |
| Control | 26/205 (13%) | 9/68 (13%) | .. | .. |
| Intervention | 18/183 (9.8%) | 1/75 (1.3%) | 0.13 (0.02–1.08), p=0.059 | 0.12 (0.01–1.02), p=0.052 |

Analysis includes children < 2 years old at baseline and children born into the study after baseline and <2 years old at the time of the 24-month visit. Prevalence results are presented as (n/N (%)). All effect estimates are presented as prevalence ratios (ratio of ratios) and estimated using generalized estimating equations to fit Poisson regression models with robust standard errors.

*Outcome was pre-specified in trial registration. All other outcomes are exploratory.

[†]Pathogen outcomes adjusted for child age and sex, caregiver's education, and household wealth index. Reported diarrhea was also adjusted for baseline presence of a drop-hole cover and reported use of a tap on compound grounds as primary drinking water source. Sample sizes for adjusted analyses are slightly smaller than numbers presented in prevalence estimates due to missing covariate data.

[§]Models would not converge due to sparse data. Y. enterocolitica, *V. cholerae*, *E. histolytica*, and rotavirus were detected in <2% of samples in each arm at each phase and excluded. Descriptive data for these pathogens are available in the **Appendix 1—table 2**. Source files available in **Table 3—source data 1** and **Table 3—source code 1**.

The online version of this article includes the following source data for Table 3:

**Source code 1.** Intervention effect on children born after implementation.

**Source data 1.** Intervention effect on children born after implementation.

Point estimates of effect and associated confidence intervals were largely similar in unadjusted and adjusted models with few exceptions (e.g. ETEC at 24 month) (*Table 2*). Multivariable models for GPP outcomes and STH outcomes were adjusted for covariates selected a priori (child age, sex, caregiver education, and household wealth index). No other variables met our inclusion criteria for multivariable models, which included being imbalanced between intervention and control at baseline and meaningfully changing 12-month effect estimates (>10% change in prevalence ratios) (*Appendix 1—table 9*). While the relationship between age and pathogen prevalence appeared to be non-linear for many pathogens (*Appendix 1—figure 4*), the inclusion of a higher order age term (age squared) did not meaningfully change effect estimates in the main or sub-group analyses (*Appendix 1—table 10*). Three measures of seasonality were considered for inclusion in multivariable models to adjust for any difference in seasonal distributions of data collection: (1) a binary variable defining the 'rainy' (November – April) and 'dry' seasons (May – October) in Maputo, (2) a measure of cumulative rainfall (mm) in the 30 days prior to data collection, and (3) sine and cosine terms representing dates of sample collection. While there was some imbalance between arms in data collected during the wet and dry seasons at baseline (*Appendix 1—table 9*), no measure of seasonality meaningfully changed effect estimates in the 12- and 24-month analyses and seasonality was excluded from final multivariable models (*Appendix 1—table 9* and *Appendix 1—table 11*). For

diarrhea, two variables in addition to variables selected a priori met our inclusion criteria and were included in adjusted models: presence of a latrine drop-hole cover at baseline and reported use of a water tap located within the compound grounds at baseline (*Appendix 1—table 9*). The magnitude of effect estimates were larger and confidence intervals wider for diarrhea in adjusted versus unadjusted models in the 12-month and 24-month analyses (*Table 2*).

In addition to the main analyses which included all enrolled children, we also performed two subgroup analyses. The first included children who were born after the intervention was implemented (or after baseline in control compounds) and present at the 12- and/or 24-month follow-up visit. This analysis allowed us to evaluate the impact of the intervention on young children who were never exposed to poor sanitation at baseline. The second sub-group analysis included only children with repeated measures at baseline and 12- and/or 24-month follow-up.

In sub-group analyses comparing children born into study compounds before the 24-month visit with children of similar ages at baseline (<2 years old), there was suggestive evidence that the intervention reduced the prevalence of infection with any STH by 49% (n = 522; adjusted prevalence ratio 0.51, [95% CI 0.27–0.95]), *Trichuris* by 76% (n = 522; 0.24, [0.10–0.60]), and *Shigella* by 51% (n = 630; 0.49, [0.28–0.85]) (*Table 3*). These effects were attenuated in sub-group analyses restricted to older children (>24 months) who were born before the intervention was implemented and present at the 24-month phase (*Appendix 1—table 12*). We did not observe intervention effects among children born into the study by the 12-month visit, but the sample size was small, resulting in high uncertainty in effect estimates (*Appendix 1—table 13*).

Longitudinal sub-group analyses explored the effect of the intervention on children with repeated measures at baseline and 12 month (for unadjusted analyses: n = 870 data points [435 children with repeat measures] for GPP outcomes, n = 572 [286] for Kato-Katz outcomes, and n = 1112 [556] for diarrhea) and at baseline and 24 month (n = 716 (358), n = 402 (201), n = 834 (417)). Effect estimates were consistent with results from the main analyses (*Appendix 1—table 14* and *Appendix 1—table 15*) but less precise due to the reduced sample numbers.

## Discussion

We found no evidence that this urban, onsite shared sanitation intervention was protective against our pre-specified child health outcomes of enteric infection, STH re-infection, or diarrhea. We also found no strong evidence that the intervention affected prevalence of any individual pathogen, pathogen type, or coinfection with ≥2 enteric pathogens or STH. In exploratory sub-group analyses, we found suggestive evidence that the intervention reduced the prevalence of any STH, *Trichuris*, and *Shigella* infections among children born into the study by the 24-month follow-up visit. Studying children born into intervention sites after implementation allowed us to examine the effect of the intervention from birth through the first 2 years of life. These results suggest that the intervention delayed pathogen exposure and the accumulation of enteric infections during early childhood, but need to be treated with caution as this was an exploratory subgroup analysis.

The trial was neither designed nor powered to detect differences in sub-groups of children such as those born after the intervention was implemented, potentially limiting our ability to detect small effects in such analyses. Further, all exploratory sub-group analyses included multiple comparisons, increasing the likelihood of chance discoveries. However, the magnitude of the effect estimates for the outcomes of any STH, *Trichuris*, and *Shigella* observed among children born into the study by the 24-month visit, and the directional consistency of effect estimates among most other outcomes in this sub-group analysis, strengthens the plausibility of these findings.

There are several reasons we observed suggestive evidence of an effect for some outcomes among this sub-group of young children but not among older children or in the main analyses. Children's exposures vary by age, particularly as they become mobile and begin independent exploration of their environment. It is possible that the intervention reduced exposure via pathways that are important for very young children but may represent just minor pathways of exposure among older children (*Kwong et al., 2020*). Additionally, young children may experience fewer exposures outside of the compound. Reductions in exposure and subsequent infection early in life may delay or prevent the development of environmental enteric dysfunction (EED), a subclinical condition that affects the structure and function of the gut and may increase susceptibility to future infection (*Keusch et al., 2014*; *Prendergast and Kelly, 2016*). Results from the EED sub-study of the WASH Benefits cluster

randomized controlled trial (cRCT) in Bangladesh suggest that the intervention delayed but did not prevent the onset of EED (*Lin et al., 2019*). If this intervention similarly delayed the development of EED among children born into intervention sites, they may have been less susceptible to infection than children of a similar age at baseline. Finally, some pathogens, like *Giardia* and certain STH, can cause persistent infections that can remain active for months or years if not treated (*Else et al., 2020*; *Rogawski et al., 2017*). The intervention would have no effect on such infections, highlighting the potentially important role of protection from birth.

Notably, both *Shigella* and *Trichuris* are primarily anthroponotic, and infection was strongly age-dependent in this study population (*Knee et al., 2018*). These factors may help explain the differing intervention effects observed both among pathogens and age groups. The intervention was unlikely to limit exposure to animal feces, reducing the likelihood that it would impact the infection prevalence of zoonotic pathogens like *Campylobacter* or *Giardia*. The strong positive associations between age and prevalence for *Shigella* and *Trichuris* suggest that exposure increases with age. This supports the hypothesis that the intervention may have reduced the overall frequency or intensity of exposure enough to impact *Shigella* and *Trichuris* infection among young children but not older children.

Rapid urbanization is expanding informal settlements and out-pacing the expansion of sanitation services in many cities, widening the gap in sanitation access between the urban rich and poor (*UNICEF/WHO, 2019*). To our knowledge, MapSan was the first trial to estimate the health impact of an urban, onsite shared sanitation intervention and the first to use enteric infection as the primary trial outcome. Most of the urban sanitation literature published to date has evaluated the expansion of sewerage, an important and ambitious goal that is out of reach for many cities in the near-term (*Norman et al., 2010*). Access to sewerage is associated with a 30–60% reduction of diarrheal disease depending on starting conditions, and an approximately 30% reduction in enteric parasite detection, though most studies are observational and few controlled trials exist (*Barreto et al., 2010*; *Norman et al., 2010*; *Wolf et al., 2018*).

Most studies of onsite sanitation interventions have occurred in rural areas. Despite good evidence that onsite sanitation is associated with reductions in diarrheal disease (*Freeman et al., 2017a*; *Wolf et al., 2018*), several recent rural trials of basic sanitation and combined WASH interventions with good uptake and use reported mixed effects on child health outcomes including diarrhea, linear growth, and more recently, enteric infection (*Ercumen et al., 2019*; *Grembi et al., 2020*; *Sanitation Hygiene Infant Nutrition Efficacy (SHINE) Trial Team et al., 2019*; *Lin et al., 2018*; *Luby et al., 2018*; *Null et al., 2018*; *Pickering et al., 2019*; *Rogawski McQuade et al., 2020a*).

The Sanitation, Hygiene, Infant Nutrition Efficacy (SHINE) trial in rural Zimbabwe found no impact of a combined WASH intervention on diarrhea, growth, or the prevalence of a suite of enteric pathogens among children aged <12 months old but did report a small reduction in the number of parasitic pathogens detected (*Sanitation Hygiene Infant Nutrition Efficacy (SHINE) Trial Team et al., 2019*; *Rogawski McQuade et al., 2020a*).

While the WASH Benefits Bangladesh cRCT reported no effect of any WASH intervention on child growth, the sanitation, hygiene, and combined WASH study arms reduced the prevalence of diarrheal disease from 5.7% to 3.5% (*Luby et al., 2018*), accompanied by absolute reductions in *Giardia* prevalence of 6–9% among children aged 2–3 years in the same arms (*Lin et al., 2018*). The sanitation arm also reduced the prevalence of *T. trichiura* among children 2–3 years old (from 5.2% to 3.2%) but had no impact on *A. lumbricoides* or hookworm, the only other parasites detected frequently enough to estimate effects in that study (*Ercumen et al., 2019*). In a parallel analysis, only the water treatment and combined WASH interventions of the WASH Benefits Kenya cRCT reduced the prevalence of *A. lumbricoides*, suggesting that the reduction in the combined WASH arm may be attributable to the water treatment intervention (*Pickering et al., 2019*). The sanitation-only arm had no impact on any parasite measured, although *T. trichiura* was too infrequently detected to estimate effects (*Pickering et al., 2019*). An evaluation of a comprehensive suite of 34 enteric pathogens reported reduced prevalence and quantity of enteric viruses, but not bacteria or parasites, among children aged 14 months old in the combined WASH arms in the Bangladesh trial (*Grembi et al., 2020*). Together with our findings, these results suggest that sanitation and combined WASH interventions can reduce the prevalence of enteric infection in some settings but that effects may vary by pathogen, child age, intervention, and setting.

We previously published two baseline risk factor analyses to identify demographic, environmental, and WASH-related predictors of infection and environmental fecal contamination in our study setting prior to the intervention implementation (*Holcomb et al., 2020*; *Knee et al., 2018*). Age was an important predictor of infection, although the direction of its effect varied by pathogen type. Increasing age was associated with increased risk of bacterial and protozoan infections and decreased risk of viral infections (*Knee et al., 2018*). Other socio-demographic predictors of infection included breastfeeding, which was associated with a decreased risk of any infection (driven by its strong association with protozoan infection), and female sex which was associated with an increased risk of viral infection. Few sanitation-related or environmental variables were associated with infection at baseline and the magnitude of associations were often small. The presence of a latrine superstructure and drop-hole cover were associated with small reductions in risk of bacterial or protozoan infection, often only in unadjusted analyses, but other latrine features (e.g. presence of a cleanable slab) were not. The observation of feces or used diapers around the compound grounds was associated with increased risk of bacterial and protozoan infection but most other environmental and sanitary hazards were not (*Knee et al., 2018*).

Fecal contamination was common among all environmental reservoirs tested (water, soil, food preparation surfaces) at baseline. We detected one or more microbial markers of contamination in over 95% of environmental samples (*Holcomb et al., 2020*). *E. coli* was the most frequently detected and abundant marker of contamination among all sample types, and human-associated markers were most frequently detected in soil (59%) and stored drinking water (17%) samples. Measures of latrine quality that were associated with small reductions in infection risk (e.g. drop-hole covers, latrine superstructures) were not associated with decreased odds of fecal contamination in this setting. Overall, we found few consistent relationships between markers of fecal contamination and environmental, WASH-related, and demographic characteristics at baseline (*Holcomb et al., 2020*).

While these results suggest WASH-related and environmental risk factors may be poor determinants of child health in this setting, the lack of heterogeneity in WASH conditions at baseline, given the selection criterion that compounds must share sanitation in 'poor condition,' may have limited our ability to identify strong WASH-related predictors of infection or environmental fecal contamination. Results from a forthcoming companion study suggests the intervention had mixed effects on environmental fecal contamination. The intervention may have reduced the concentration of *E. coli* by an order of magnitude in soil collected from latrine entrances after 12 months; however, there was no effect on the prevalence or concentration of indicators of fecal contamination in any other environmental compartment sampled at that time (*Holcomb et al., 2021*). It is unlikely that the observed reductions in fecal contamination in soils alone would be sufficient to impact health outcomes in this setting. Other studies that have evaluated the impact of sanitation interventions on fecal contamination of the surrounding environment have found limited evidence of effect (*Clasen et al., 2014*; *Ercumen et al., 2018a*; *Ercumen et al., 2018b*; *Fuhrmeister et al., 2020*; *Patil et al., 2014*; *Pickering et al., 2015*; *Sclar et al., 2016*; *Steinbaum et al., 2019*).

In this setting, where fecal contamination was pervasive and burden of infection high, even considerable reductions in contamination and exposure may have been insufficient to realize measurable health gains as the intervention did not address all potential transmission pathways (*Briscoe, 1984*; *Julian, 2016*; *Robb et al., 2017*). For example, the intervention did not address child feces disposal practices or handwashing behaviors and it is unlikely that the intervention infrastructure would have changed these (*Cochrane Infectious Diseases Group et al., 2019*). Previous studies of sanitation interventions have found no reduction in hand contamination (*Ercumen et al., 2018b*), which has been associated with increased incident diarrheal disease in young children (*Pickering et al., 2018*). The intervention may not have reduced exposure via consumption of contaminated food – particularly foods contaminated prior to arrival in the compound – likely an important source of enteric pathogen transmission in some settings (*Julian, 2016*; *Kwong et al., 2020*). Children's exposure to animal feces has been documented in rural, peri-urban, and urban settings and could be an important, unmitigated source of exposure to enteric pathogens in both intervention and control arms where animals were frequently observed (*Delahoy et al., 2018*; *Kwong et al., 2020*; *Penakalapati et al., 2017*). Observation of animals in compounds was examined as a potential confounder but did not change effect estimates.

The intervention was delivered at the compound level, not the community level, and was not designed to achieve any specified threshold of sanitation coverage in the study neighborhoods.

Previous studies have suggested that achieving a certain level of community sanitation coverage may be necessary to reduce disease burdens (*Barreto et al., 2007*; *Fuller and Eisenberg, 2016*; *Fuller et al., 2016*; *Harris et al., 2017*; *Jung et al., 2017*; *Spears et al., 2013*; *Wolf et al., 2018*). For example, a study of a large-scale sewerage expansion in urban Brazil found that the intervention reduced diarrheal disease by 22%, with neighborhood coverage level being the single most important explanatory variable (*Barreto et al., 2007*). We did not measure neighborhood-level sanitation coverage, but previous estimates show that while coverage is high and open defecation is limited (1%), only 9% of sanitation systems are safely managed (*Satterthwaite et al., 2019*). Further, in the Nhlamankulu district where many of our study sites are located, the majority of households (56%) rely on private pit latrines, most of which are in poor condition (*Devamani et al., 2014*; *Satterthwaite et al., 2019*). Together with our results, this suggests that both the extent and quality of community coverage are likely important to reducing overall transmission. Sanitation coverage and quality may be especially important in urban areas given the proximity of compounds and the opportunity for person-to-person contact, neighborhood-level exposure, and for external sources of contamination (e.g. a neighbor's flooded pit latrine) to influence compound-level exposures (*Barreto et al., 2007*). We did not measure neighborhood-level exposures, which may be important for young children in slum settings (*Ezeh et al., 2017*; *Medgyesi et al., 2019*), and their impact on our health outcomes is unclear. In addition to neighborhood-level exposures, the transience of the study population meant that trips to and from provinces outside of Maputo, where exposures were varied and unmeasured, were common.

It is unlikely that our findings are due to poor intervention fidelity or use, a challenge encountered in some trials of rural sanitation interventions (*Clasen et al., 2014*; *Patil et al., 2014*). The use of the intervention required minimal behavior change as compound members switched from using their existing latrine in poor condition, which was removed following construction of the intervention latrine, to using the new hygienic latrine. The results of a forthcoming process evaluation demonstrate that 96% of intervention latrines were well-maintained 2 or more years after construction, suggesting continued use by compound members (*Bick, 2021*). Further, only 3% of intervention compounds (8/270) had a secondary, non-intervention latrine in use after two or more years, indicating that members of most intervention compounds exclusively used the intervention latrines (*Bick, 2021*). It is possible that development in the study neighborhoods, including changes to sanitation facilities in control compounds, contributed to the limited effect of the intervention. However, results from sensitivity analyses that excluded control compounds with upgraded sanitation were consistent with results from the main analyses.

The two intervention designs we evaluated in this study – communal sanitation blocks and shared latrines – utilized the same basic sanitation technology but differed in the number of cabins and amenities available. While it is possible that this heterogeneity in design may have modified the effect of the intervention, this study was not powered to test this. Moreover, all intervention compounds were encouraged to independently upgrade their facilities by adding features like electricity and handwashing stations, or by connecting existing handwashing stations to the water supply, resulting in heterogeneity even within the two broad categories of intervention type.

While the NDC dewormed every study compound annually during the study period, it is possible that not all study participants received, or took, the medication and that the time between deworming and subsequent measurement of STH re-infection varied among children. Additionally, single-dose albendazole can have limited effectiveness against certain STH, notably *Trichuris* (*Moser et al., 2017*). Inadequate or ineffective deworming could have limited our ability to detect an effect on STH outcomes. Sensitivity analyses adjusting for caregiver-confirmed deworming and for estimated time between deworming and re-infection measurement produced similar results to the main analysis.

There are several important limitations of this study. As the intervention was pre-planned and not implemented by the study team, we could not randomize its allocation, increasing the risk of confounding. We assessed potential confounding variables at baseline and used a DID analysis, which accounts for baseline outcome measures, to limit the effect of unmeasured, residual confounding. While we attempted to enroll intervention and control compounds with comparable numbers of residents, the NGO which identified and implemented the intervention selected most of the largest eligible compounds for intervention. This resulted in intervention compounds having a slightly higher mean number of residents than control compounds (*Table 1*). Crowding has been identified as a risk

factor for pathogen transmission and poor health outcomes in other studies (*Halpenny et al., 2012*; *Rahman et al., 1985*; *Rogawski McQuade et al., 2020b*), although we found limited evidence of this in our study population at baseline (*Knee et al., 2018*). Further, we assessed the number of compound residents as a potential confounder but found that it did not meaningfully change the DID estimates for our pre-defined outcomes (*Appendix 1—table 9*). We consider our analysis to be robust to small differences in study arms at baseline; however, we cannot exclude the possibility of residual confounding due to such differences, a limitation of non-randomized designs.

It was not possible to mask participants to their intervention status, and our measure of caregiver-reported diarrhea could be subject to respondent and recall biases. To reduce the risk of respondent bias, the MapSan field enumerator team and implementation team were different, and respondents were not informed explicitly that the MapSan team was evaluating the health effect of the intervention. To limit recall bias, we used a 7-day recall period (*Arnold et al., 2013*). Our other pre-specified outcomes were objective measures of pathogen infection and not subject to the same biases (*Brown and Cumming, 2020*).

Due to the greater than expected losses to follow-up in both study arms, we were not able to follow all children enrolled at baseline through time as expected, but we still achieved our target enrollment numbers due to migration and births into study compounds. We conducted the originally planned longitudinal analysis as a sub-group analysis. It also served as a sensitivity analysis to estimate the impact of migration on our effect estimates. Results from this sub-group analysis were largely similar to results of the main analysis which treated measures as repeated cross-sections, although the reduction in sample size led to wider confidence intervals (*Appendix 1—table 14* and *Appendix 1—table 15*). Measures of outcomes and covariates in children with and without repeated measures were mostly similar, further limiting the likelihood that changes in the study population biased our results.

While molecular detection of enteric pathogens in stool is evidence of pathogen exposure, it is not necessarily evidence of active infection, making its clinical significance less clear (*Brown and Cumming, 2020*). We assumed pathogen detection by the GPP indicated infection because the assay's limits of detection exceeded the median infectious dose of most pathogens. While the GPP detects many enteric pathogens recognized as important causes of childhood diarrhea in LMICs, (*Liu et al., 2016*) it does not detect all enteric pathogens of importance. Further, qualitative, cross-sectional analysis of stools does not provide information on the duration or intensity of infection or pathogen carriage. Quantitative results like those produced by multiplex quantitative PCR panels can be used to aid identification of etiologic agents of diarrhea, especially in cases of coinfection, and to differentiate between low-level enteric pathogen detection of unknown clinical relevance and higher concentration shedding which is more clearly associated with disease (*Liu et al., 2014*; *Liu et al., 2016*; *Platts-Mills et al., 2013*). Some studies have demonstrated overall good performance of the GPP but observed elevated false positive detection rates for the *Salmonella* targets (*Duong et al., 2016*; *Kellner et al., 2019*). For this reason we removed *Salmonella* results from our pre-specified outcome definition. Results from analyses including and excluding *Salmonella* were similar. In addition, some studies have observed reduced sensitivity or specificity for some GPP targets compared with qPCR-based methods, including norovirus, adenovirus, *Campylobacter*, *Yersinia enterocolitica*, ETEC, and *Salmonella*, although inconsistencies between studies exist and are likely due to differences in comparator assays or sample storage and processing (*Chhabra et al., 2017*; *Deng et al., 2015*; *Duong et al., 2016*; *Huang et al., 2016*; *Zhan et al., 2020*; *Zhuo et al., 2017*). Further, the lack of an adequate reference standard in most comparative studies complicates interpretation (*Freeman et al., 2017b*).

Our ability to detect an effect on our primary outcome, the prevalence of $\geq 1$ bacterial or protozoan infection, may have been limited by (1) the extended duration of shedding of some pathogens following active infection; (2) the overall high burden of disease in our study population, particularly among older children; and (3) residual confounding by age given the strong observed relationship between age and infection status (particularly for protozoan pathogens), all of which may have biased our results toward the null. Further, the intervention may have impacted the concentration of pathogens shed (*Grembi et al., 2020*; *Lin et al., 2019*), but our binary outcome was not sensitive to such differences The qualitative nature of the GPP did not allow us to interrogate this question.

We analyzed a smaller number of stool samples for STH than for other enteric pathogens due to requirements of the Kato-Katz method used for STH detection. The Kato-Katz method can only be

performed on whole, solid stool. Diarrheal samples and rectal swabs, the latter of which were introduced during the 12-month follow-up phase, were not eligible for STH analysis by Kato-Katz. Further, when limited stool material was collected, we prioritized the molecular analysis used for the primary outcome. While the smaller sample size available for the STH analyses may have reduced our ability to detect small effects, the proportions of whole stool, diarrheal diaper samples, and rectal swabs were similar between arms at each phase (*Appendix 1—table 1*). This limited the potential impact that sample type could have on our results.

While the Kato-Katz method performs similarly to other microscope-based and molecular methods for detection of moderate- to high-intensity infections, it may be less sensitive than molecular methods in detecting low-intensity infections (*Benjamin-Chung et al., 2020*; *Cools et al., 2019*). A recent study has also suggested reduced specificity of the Kato-Katz method for detection of low-intensity *A. lumbricoides* infections (*Benjamin-Chung et al., 2020*). In settings where low-intensity infections are common, or where STH may be targeted for elimination, methods with better diagnostic accuracy, like qPCR, may be considered.

We had limited ability to evaluate the impact of seasonality or weather-related trends on our effect estimates due to drought conditions during the 2015/2016 rainy season. We adjusted models for cumulative 30-day rainfall, a binary indicator of wet/dry season, and sine/cosine terms of sample collection date (*Stolwijk et al., 1999*) but excluded all seasonality terms from final multivariable models because they did not meaningfully change effect estimates.

Our results demonstrate that access to hygienic, shared onsite sanitation systems was not sufficient to reduce enteric infection or diarrhea in children aged 6 years or younger ($\leq$4 at baseline) 12–24 months after implementation. Results from our sub-group analysis of children born into intervention sites showed a substantial reduction in the prevalence of any STH, *Trichuris*, and *Shigella* infection, suggesting that children may require protection from birth to reduce or delay infection burdens. Our results do not suggest that shared sanitation is inadvisable in this setting, as we did not compare against household-level sanitation improvements, nor do they account for the many non-health-related benefits associated with this intervention or upgraded sanitation generally (*Caruso et al., 2018*; *Sclar et al., 2018*; *Shiras et al., 2018*).

The need for effective sanitation solutions may be most urgent in densely populated, low-income, informal communities like our study setting where ubiquitous fecal contamination drives high infection burdens. Disease transmission in these settings may be driven by multiple interrelated pathways, complicated by frequent migration and the diversity of circulating pathogens, and therefore difficult to interrupt. While decades of research have demonstrated meaningful health gains following sanitation improvements, the results of this study and other rigorous trials of sanitation interventions suggest that the relationship between sanitation and health is complex, difficult to measure, and may not be generalizable across diverse settings and populations.

## Materials and methods

### Study design and intervention

MapSan was a controlled before-and-after trial, and details of the study design and analysis plan have been published previously (*Brown et al., 2015*). We conducted the study in 16 densely populated, low-income, informal neighborhoods in Maputo, Mozambique. The intervention was delivered to compounds, typically groups of three to five households (although larger and smaller compounds exist) often delineated by a wall or barrier, that shared sanitation and outdoor living space. Shared compound sanitation facilities are not considered public facilities. We collected data in an open cohort of children in intervention and control compounds at three time-points: baseline (pre-intervention), 12 months post-intervention, and 24 months post-intervention.

The NGO Water and Sanitation for the Urban Poor selected intervention compounds and designed and built 300 intervention facilities – pour-flush toilets discharging to septic tanks, the liquid effluent of which flows to the soil through soakaway pits (*Appendix 1—figure 5* and *Appendix 1—figure 6*). There were two intervention designs with the same basic sanitation technology: communal sanitation blocks (CSBs) and shared latrines (SLs) (*Appendix 1—figure 7* and *Appendix 1—figure 8*). The primary difference between CSBs and SLs was size. CSBs (n = 50) included multiple stalls with toilets and served compounds of 21 or more people with one stall allocated per

20 residents. CSBs also included rainwater harvesting systems, a municipal shared water connection, elevated water tanks for storage of municipal water, a handwashing basin, a laundry facility, and a well-drained area for bathing. Shared piped water connections were part of the municipal water system and could be used for drinking in addition to other domestic purposes. Rainwater was intended for cleaning and flushing but not drinking. Shared latrines (n = 250) were single-stall facilities serving fewer than 21 people. All septic tanks were sized to require emptying after approximately two years.

Intervention compounds were located in 11 neighborhoods of the Nhlamankulu and KaMaxakeni districts of Maputo (*Appendix 1—figure 9*). The NGO selected intervention compounds using the following criteria: (1) residents shared sanitation in poor condition as determined by an engineer; (2) the compound was located in the pre-defined implementation neighborhoods; (3) there were no fewer than 12 residents; (4) residents were willing to contribute financially to construction costs; (5) sufficient space was available for construction of the new facility; (6) the compound was accessible for transportation of construction materials and tank-emptying activities; (7) the compound had access to a legal piped water supply; and (8) the groundwater level was deep enough for construction of a septic tank. Intervention compounds were expected to pay approximately 10–15% of the construction costs (~$64 for shared latrines and ~$97 for CSBs) within one year of construction, with 25% of the total due upfront. Presence of a child was not a selection criterion and therefore not all intervention sites were included in the study. Opening of newly constructed intervention latrines occurred between February 2015 and February 2016. The study team used criteria 1, 3, 4, and 7 to select control sites that had at least one child younger than 48 months old in residence. We enrolled intervention and control compounds concurrently to limit any differential effects of seasonality or other secular trends on the outcomes (*Appendix 1—figure 2*). Additionally, we attempted to enroll control compounds with similar numbers of residents as intervention compounds. Willingness to pay for facilities among controls was assessed using hypothetical versions of questions posed to interventions. Control compounds were located within the 11 intervention neighborhoods and 5 adjacent but similar neighborhoods due to the limited availability of eligible compounds remaining within intervention neighborhoods (*Appendix 1—figure 9*). Intervention selection criteria (5, 6) and (8) were not used to select control sites as they were deemed to be related to intervention construction and maintenance and unlikely to influence our outcomes. It was not possible to blind participants or enumerators to intervention status.

## Participants

We enrolled eligible children at three time points: baseline (0 months), 12 months post-intervention, and 24 months post-intervention. Children aged 1–48 months old were eligible for baseline enrollment if we received written informed consent from a parent or guardian and if the head of the compound provided verbal assent for the compound to be included in the study. Children were eligible for enrollment at 12- and 24-month visits if they were aged 1–48 months or if they were eligible for enrollment at baseline but absent during that study visit. Children who moved into the compound fewer than 6 months before the 12-month or 24-month visit were not eligible for enrollment during that phase given their limited exposure to their new compound.

## Procedures

Trained field enumerators completed consent procedures and surveys in the participant's preferred language (Portuguese or Changana) and collected biological sampless from enrolled children (Appendix 1- Consent procedures, survey administration, and sample collection and analysis). At baseline we aimed to visit intervention compounds 2 weeks prior to the opening of the new latrines. We scheduled follow-up visits to be 12 months (±2 weeks) and 24 months (±2 weeks) from the date compound members began using their new latrines, with visits to control compounds made concurrently (±2 weeks).

We collected stool samples independently of reported symptomology. If we were unable to collect a stool sample after multiple attempts, a registered nurse collected a rectal swab after obtaining written consent for the procedure from a parent or guardian. Stool samples were kept cold and delivered to the Laboratory of Molecular Parasitology at the Instituto Nacional de Saúde (INS) within 6 hr of collection for analysis and storage at −80°C.

Samples were shipped frozen with temperatures monitors to the Georgia Institute of Technology (Atlanta, USA) where we used the xTAG GPP (Luminex Corp, Austin, USA), a qualitative multiplex molecular assay, to detect 15 enteric pathogens in stool samples: *Campylobacter jejuni/coli/lari*; *Clostridium difficile*, toxin A/B; enterotoxigenic *Escherichia coli* (ETEC) LT/ST; Shiga-like toxin producing *E. coli* (STEC) stx1/stx2; *E. coli* O157; *Salmonella*; *Shigella boydii/sonnei/flexneri/dysenteriae*; *Vibrio cholerae*; *Yersinia enterocolitica*; *Giardia lamblia*; *Cryptosporidium parvum/hominis*; *Entamoeba histolytica*; adenovirus 40/41; norovirus GI/GII; and rotavirus. The GPP has been rigorously tested and extensively used for stool-based enteric pathogen detection (*Chisenga et al., 2018*; *Claas, 2013*; *Deng et al., 2015*; *Duong et al., 2016*; *Huang et al., 2016*; *Kellner et al., 2019*; *Khare et al., 2014*; *Navidad et al., 2013*; *Patel et al., 2014*). We analyzed samples according to manufacturer instructions with the addition of elution steps for the pretreatment of rectal swabs and diaper material saturated with liquid stool (Appendix 1- Consent procedures, survey administration, and specimen collection and analysis). Technicians at INS assessed stool samples for the presence of soil-transmitted helminths (STH) using the single-slide Kato-Katz microscope method (Vestergaard Frandsen, Lausanne, Switzerland).

Representatives of the National Deworming Campaign (NDC) at the Mozambican Ministério da Saúde (MISAU) offered single-dose albendazole (400 mg, 200 mg for children aged 6–12 months) to all eligible members of intervention and control compounds following sample collection activities of each phase. Eligibility was defined by the NDC and included compound members older than 6 months who were not pregnant.

## Outcomes

For the 12-month analysis, we pre-specified the primary outcome as infection with one or more of the 12 bacterial or protozoan enteric pathogens detected by the GPP and secondary outcomes as re-infection with one or more STH as detected by Kato-Katz (following albendazole treatment at baseline), and 7-day period prevalence of caregiver-reported diarrhea. All three outcomes were considered secondary outcomes in the 24-month analysis. We defined diarrhea as the passage of three or more loose or liquid stools in a 24 hr period or any stool with blood (*Arnold et al., 2013*; *Baqui et al., 1991*). We excluded viral enteric pathogens from the primary outcome definition. The intervention may not have interrupted virus transmission due to their low infectious doses, high concentration shed in feces and extended period of shedding, environmental persistence, and capability for direct person-to-person transmission (*Julian, 2016*). Following reported specificity issues with the *Salmonella* target of the GPP, we removed it from our GPP-based outcome definitions (*Duong et al., 2016*; *Kellner et al., 2019*). In addition to the pre-specified outcomes, we evaluated the effect of the intervention on specific pathogen types (bacterial, protozoan, viral) and on individual pathogens. The results for other secondary outcomes listed in the trial registration (growth and environmental enteric dysfunction) will be published separately.

## Statistical analysis

Our sample size calculation has been described previously (*Brown et al., 2015*). We included all enrolled children at each visit and analysed data as repeated cross-sectional observations. We examined the effect of the intervention at the 12-month and 24-month phases separately. We conducted two sets of exploratory sub-group analyses. The first assessed the effect of the intervention on children with repeat observations at baseline and 12 months and at baseline and 24 months visits. These longitudinal analyses also served as sensitivity analyses of the impact of participant migration on effect estimates. The second sub-group analysis compared children who were born into study sites after the intervention (or after baseline in controls) but before the 12-month or 24-month visit with children of a similar age group at baseline. For example, children born after baseline but before the 24-month visit were compared with children aged 2 years old or younger at baseline. These analyses allowed us to explore whether exposure to the intervention from birth would reduce enteric pathogen infection during the first 1–2 years of life.

We used a DID approach to assess the impact of the intervention on all outcomes at the 12- and 24-month visits. We used generalized estimating equations (GEE) to fit Poisson regression models with robust standard errors. Our GEE models accounted for clustering at the compound level because it was the highest level of nested data and the level of the intervention allocation

(*Bottomley et al., 2016*). We estimated the effect of the intervention as the interaction of variables representing treatment status (intervention versus control) and phase (pre- or post-intervention). Therefore, effect estimates from our DID analysis are presented as ratio measures (ratio of prevalence ratios) instead of absolute differences. Multivariable models were adjusted for covariates determined a priori as potentially predictive of our outcomes, including child age and sex, caregiver's education, and household wealth. Given the important and potentially non-linear relationship between age and pathogen prevalence (*Appendix 1—figure 4*), we also considered inclusion of a higher order age term (age squared) in our models (*Appendix 1—table 10*). Additional covariates (*Appendix 1—table 9*) were considered for inclusion in multivariable models if they were imbalanced between arms at baseline (>0.1 standardized difference in prevalence or mean) and resulted in a meaningful change in the DID effect estimate (±10% change in 12-month DID prevalence ratio). We assessed the potential impact of seasonality on our results in three ways: (1) inclusion of binary indicator of wet (November – April) and dry (May – October) season in multivariable models, (2) inclusion of a variable representing cumulative rainfall (mm) 30 days prior to sample or survey collection in multivariable models, and (3) inclusion of sine and cosine functions of sample and survey dates in multivariable models (*Appendix 1—table 9* and *Appendix 1—table 11*). We used the same statistical approach for sub-group analyses. All analyses were performed on complete case data, and a missing data table is presented in Appendix 1 (*Appendix 1—table 16*). We performed all statistical analyses with Stata version 16 (StataCorp, College Station, USA).

### Registration
The trial was pre-registered at ClinicalTrials.gov (NCT02362932).

## Acknowledgements

We gratefully acknowledge data collection services and other support provided by the WE Consult team and in particular Wouter Rhebergen and Ellen de Bruijn, and the hard work of the enumerators Isabel Maninha Chiquele, Sérgio Adriano Macumbe, Carolina Zavale, Maria Celina Macuacua, Guilherme Zimba, and Anabela Mondlane. Olimpio Zavale coordinated logistics for early field work. We thank staff at the Instituto Nacional de Saúde, specifically Josina Mate, Acacio Sabonete, and Jeronimo Langa, for their support throughout the project. The study would not have been possible without our implementing partners at Water and Sanitation for the Urban Poor (WSUP), in particular Guy Norman, Carla Costa, Vasco Parente, and Jonathan Stokes. At the Georgia Institute of Technology, we gratefully acknowledge the project and laboratory support provided by Aaron Bivins, Kevin Zhu, David Berendes, Fred Goddard, Olivia Ginn, Sarah Lowry, Olivia Stehr, Haley Lewis, Jonathan Pennie, Derek Whaler, Mio Unno, Catherine Reynolds, Joel Seibel, Diana Chumak, Felicitas Schneider, Katherine Brand, Jiaxin Li, and Meredith Lockwood. We also thank Ben Arnold, Jack Colford, Radu Ban, Jay Graham, Tony Kolb, Eddy Perez, Jan Willem Rosenboom, Tom Slaymaker, Larry Moulton, and Darren Saywell for technical and study design input. We recognize the early contributions of the late Dr. Jeroen Ensink to the MapSan trial protocol. As a colleague, and as a friend, he is sorely missed. Finally, we acknowledge and thank all of the participants and their families who graciously welcomed us into their homes and were so generous with their time.

## Additional information

### Competing interests
Jackie Knee, Trent Sumner, Zaida Adriano, Claire Anderson, Farran Bush, Drew Capone, Veronica Casmo, David Holcomb, Pete Kolsky, Evgeniya Molotkova, Judite Monteiro Braga, Celina Russo, Wolf Peter Schmidt, Jill Stewart, Winnie Zambrana, Rassul Nalá, Oliver Cumming, Joe Brown: As we have indicated on the ICMJE COI form, attached, this authors's time working on this study was funded in part by the study's funders, the United States Agency for International Development and the Bill and Melinda Gates Foundation. The other authors declare that no competing interests exist.

## Funding

| Funder | Grant reference number | Author |
|---|---|---|
| Bill and Melinda Gates Foundation | OPP1137224 | Oliver Cumming Joe Brown |
| United States Agency for International Development | GHS-A-00-09-00015-00 | Oliver Cumming Joe Brown |

The funders had no role in study design, data collection and interpretation, or the decision to submit the work for publication.

## Author contributions

Jackie Knee, Data curation, Formal analysis, Investigation, Methodology, Writing - original draft, Writing - review and editing; Trent Sumner, Data curation, Formal analysis, Investigation, Methodology, Writing - review and editing; Zaida Adriano, Drew Capone, Investigation, Methodology, Writing - review and editing; Claire Anderson, Farran Bush, Veronica Casmo, David Holcomb, Evgeniya Molotkova, Judite Monteiro Braga, Celina Russo, Winnie Zambrana, Investigation, Writing - review and editing; Pete Kolsky, Conceptualization, Writing - review and editing; Amy MacDougall, Formal analysis, Methodology, Writing - review and editing; Wolf Peter Schmidt, Conceptualization, Formal analysis, Methodology, Writing - review and editing; Jill Stewart, Methodology, Project administration, Writing - review and editing; Valentina Zuin, Conceptualization, Methodology, Writing - review and editing; Rassul Nalá, Conceptualization, Data curation, Project administration, Writing - review and editing; Oliver Cumming, Conceptualization, Supervision, Funding acquisition, Methodology, Project administration, Writing - review and editing; Joe Brown, Conceptualization, Resources, Formal analysis, Supervision, Funding acquisition, Investigation, Methodology, Project administration, Writing - review and editing

## Author ORCIDs

Jackie Knee ![ORCID] https://orcid.org/0000-0002-0834-8488
Drew Capone ![ORCID] https://orcid.org/0000-0002-2138-6382
David Holcomb ![ORCID] https://orcid.org/0000-0003-4055-7164
Jill Stewart ![ORCID] https://orcid.org/0000-0002-3474-5233
Oliver Cumming ![ORCID] https://orcid.org/0000-0002-5074-8709
Joe Brown ![ORCID] https://orcid.org/0000-0002-5200-4148

## Ethics

Clinical trial registration ClinicalTrials.gov, number NCT02362932.
Human subjects: The study protocol was approved by the Comité Nacional de Bioética para a Saúde (CNBS), Ministério da Saúde (333/CNBS/14), the Research Ethics Committee of the London School of Hygiene & Tropical Medicine (reference # 8345), and the Institutional Review Board of the Georgia Institute of Technology (protocol # H15160).

## Decision letter and Author response

Decision letter https://doi.org/10.7554/eLife.62278.sa1
Author response https://doi.org/10.7554/eLife.62278.sa2

# Additional files

## Supplementary files

- Transparent reporting form
- Reporting standard 1. Consort 2010 checklist extension for cluster trials.

## Data availability

All data generated or analysed during this study are included in the manuscript and supporting files. Source data files and code have been provided for all analyses and specifically for Figure 1 and

Tables 1, 2, and 3. Additionally, we have archived all data and code at Open Science Framework (https://osf.io/me2tx, DOI 17605/OSF.IO/ME2TX).

The following dataset was generated:

| Author(s) | Year | Dataset title | Dataset URL | Database and Identifier |
|---|---|---|---|---|
| Brown J | 2020 | Effects of an urban sanitation intervention on childhood enteric infection and diarrhoea in Mozambique | https://osf.io/me2tx/ | Open Science Framework, me2tx |

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

## Appendix 1

## Consent procedures, survey administration, and specimen collection and analysis

Enumerators visited households with enrolled children at least twice at each point of follow-up. On the first visit of each phase, enumerators completed consent procedures, administered child-, household-, and compound-level surveys, and delivered stool sample collection supplies. The child's mother was the target respondent for child and household surveys, although the father or another guardian was also eligible. For compound-level surveys, the head of the compound or his or her spouse was the preferred respondent. We sought written, informed consent from the parent or guardian of each eligible child prior to initial enrollment. We sought verbal assent from parents or guardians at each follow-up visit. Consent procedures, surveys, and all study-related verbal communication was performed in Portuguese or Changana as requested by the participant. Written materials were provided in Portuguese.

Enumerators provided each caregiver with stool collection supplies, including disposable diapers, a plastic potty if the child was no longer wearing diapers, and a pre-labeled sterile sample bag. Enumerators returned the next day to collect the samples. If a sample was unavailable during the scheduled pickup, caregivers called the field team, using phone credit provided by the study, as soon as one was available or if fresh collection supplies were needed. If field enumerators were unable to collect a stool sample after multiple attempts, a registered nurse used an anatomically designed rectal swab (Copan Diagnostics Inc, Murrieta, CA, USA) to collect fecal material. Parents or guardians were required to complete a separate written consent procedure prior to collection of rectal swabs. Stool samples and rectal swabs were stored in coolers with cold packs and delivered to the Medical Parasitology Laboratory at the Mozambican Ministry of Health (MISAU/INS) within 6 hr of collection. Technicians at INS prepared Kato-Katz slides for soil-transmitted helminth (STH) detection the day of receipt and read results within 30 min of preparation for hookworm and within 24 hr for other STH. In addition to STH analysis, laboratory technicians at INS also aliquoted stools into several sterile tubes and stored them, and any rectal swabs, at $-80°C$. If a child produced a liquid stool, lab technicians stored a piece of the saturated diaper material ('diaper samples') at $-80°C$. Stool samples were shipped frozen on dry ice with temperature probes to the Georgia Institute of Technology in Atlanta, Georgia, USA where they were stored at $-80°C$ until analysis.

We followed manufacturer instructions for the pretreatment, extraction, and analysis of stool samples by the Luminex Gastrointestinal Pathogen Panel (GPP), with additional elution steps added to the pretreatment protocol for rectal swabs and diaper samples. We eluted diaper samples in 2.5 mL of lysis buffer (ASL buffer, Qiagen, Hilden, Germany). We used a sterile 10 mL syringe to facilitate elution via agitation by taking in and expelling the buffer five times. We used 1 mL of the final eluate in the pretreatment. We agitated rectal swabs in 1 mL of lysis buffer for 1 min and used the eluate in the pretreatment. Following pretreatment, we extracted DNA and RNA using the QIAcube HT platform and the QIAamp 96 Virus QIAcube HT Kit (Qiagen, Hilden, Germany). We added MS2, a nonpathogenic RNA virus, to each sample prior to nucleic acid extraction as an extraction and RT-PCR inhibition control. We included at least one sample process control (containing only lysis buffer and MS2) and negative extraction control (containing only lysis buffer) with each set of extractions. During the PCR step we included at least one no-template control, containing molecular grade water and all PCR reagents with each run. To assess elution and extraction of nucleic acid from swab and diaper samples, we measured the concentration of double-stranded DNA (dsDNA) present in a subset of extracts using the Qubit High Sensitivity dsDNA kit (Invitrogen, Carlsbad, CA, USA) and Qubit 4 Fluorimeter (Invitrogen, Carlsbad, CA, USA). The mean concentration of dsDNA recovered from rectal swabs was 26.3 ng/μL (SD 15.5, n = 195, 25 swabs with measures above assay detection limit) and from diaper samples was 28.7 ng/μL (SD 16.9, n = 61, 16 diapers with measures above assay detection limit). The concentration of dsDNA recovered from whole stool exceeded the assay detection limits in most cases. The mean concentration of dsDNA in the subset of stools with measurable results was 40.8 ng/μL (SD = 16.5, n = 33, 57 samples had concentrations above the assay detection limit). Following extraction, we stored all extracts at 4°C and analyzed them by GPP within 24 hr. For long-term storage, we archived samples at $-80°C$. We extracted and analyzed approximately 10% of samples in duplicate (biological replicates). If duplicate analyses yielded different results, we

combined the results from all analyses such that the final result captured all positive detections for a given sample. If we could not detect a MS2 signal in a given sample, we either re-extracted or diluted the extract 1:10 in molecular grade water and re-assayed by GPP.

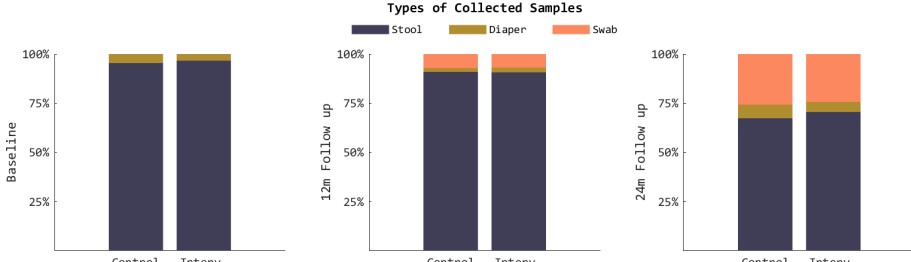

**Appendix 1—figure 1.** Proportion of each type of sample collected during the baseline, 12-month, and 24-month phases. Results stratified by study arm. Rectal swabs were not introduced until the 12-month phase of the study.

**Appendix 1—table 1.** Number and proportion of sample types collected in each arm at each phase.

|  | Baseline | | 12 month | | 24 month | |
|---|---|---|---|---|---|---|
|  | Control | Intervention | Control | Intervention | Control | Intervention |
| Whole stool | 377 (96%) | 351 (97%) | 361 (91%) | 380 (93%) | 307 (67%) | 333 (72%) |
| Diarrheal diaper | 15 (3.8%) | 10 (2.8%) | 4 (1.0%) | 4 (0.98%) | 32 (7.0%) | 20 (4.3%) |
| Rectal swab* | 0 (0%) | 0 (0%) | 30 (7.6%) | 24 (5.9%) | 120 (26%) | 109 (24%) |

*Mean concentration of double-stranded DNA recovered from whole stool was 40.8 ng/μL (SD = 16.5, n = 33 with 57 samples excluded as their concentrations exceeded the upper detection limit of the assay), diaper samples was 28.7 ng/μL (SD = 16.9, n = 61 with 16 samples excluded as concentrations exceeded upper detection limit of assay), and rectal swabs was 26.3 ng/μL (SD = 15.5, n = 195 with 25 samples excluded as concentrations exceeded upper detection limit of assay). Only a subset of each sample type assayed for dsDNA concentration.

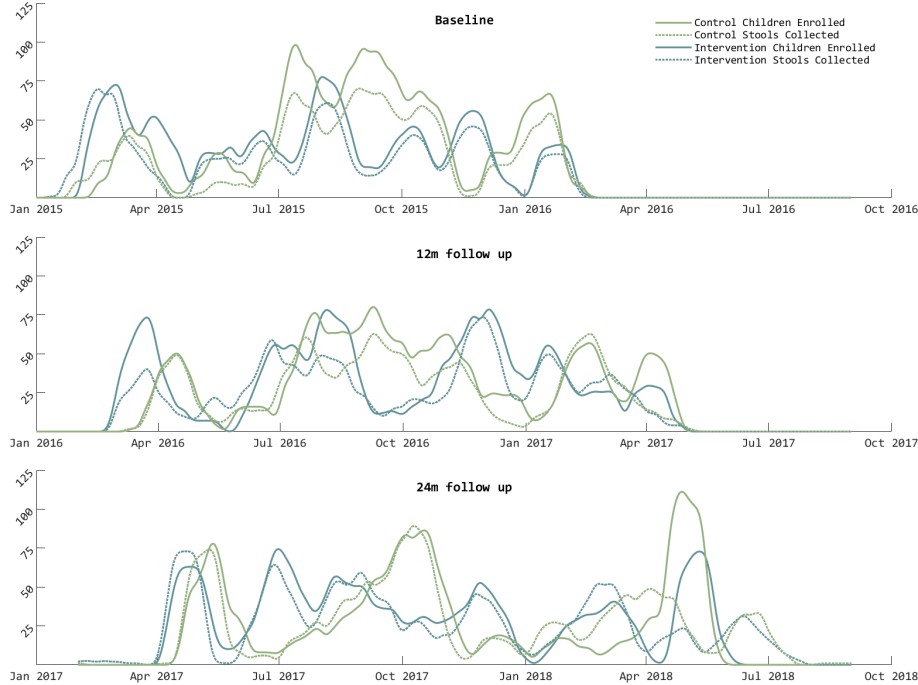

**Appendix 1—figure 2.** Enrollment and stool sample collection profile. Graphs depict 4-week rolling average of the number of intervention and control children enrolled/visited (solid lines) and the number of stool samples collected (including whole stool, diaper samples, and rectal swabs) during the baseline, 12-month, and 24-month phases. The overall success of stool sample collection was 78% at baseline, 86% at 12 month, and 90% at 24 month. The increase in success rate was due to the introduction of rectal swab collection during the 12-month phase.

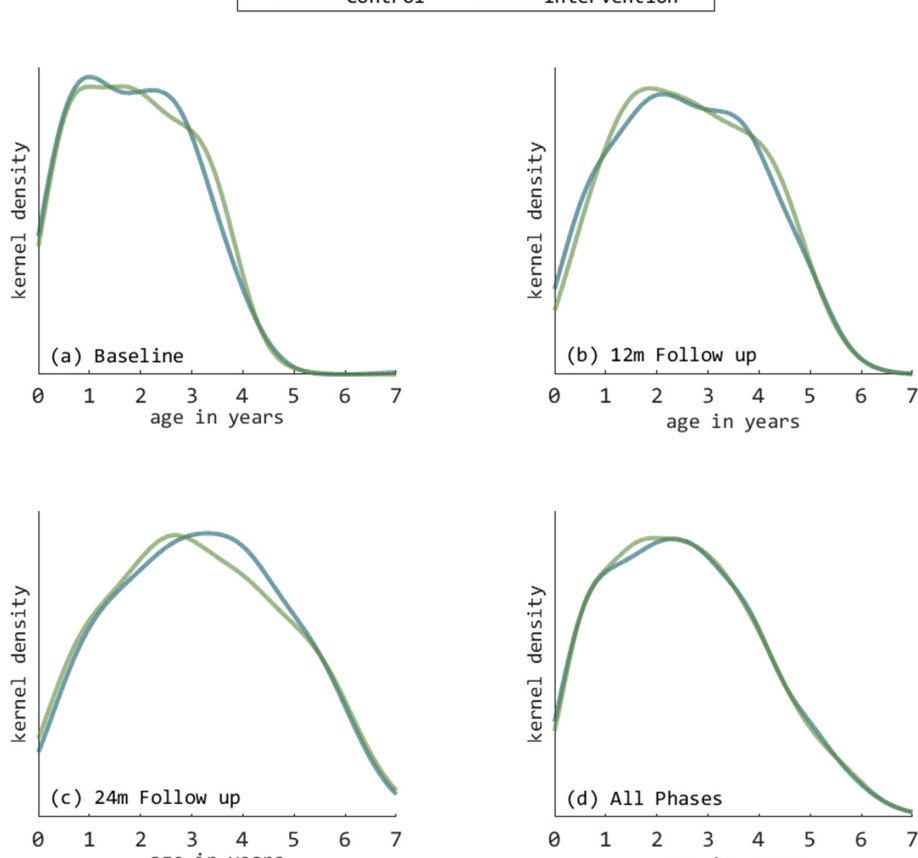

**Appendix 1—figure 3.** Distribution of age (years) of enrolled children at each phase. Results are presented as kernel density plots and stratified by study arm (intervention=blue, control=green) and phase: (**a**) Baseline phase, (**b**) 12-month follow-up, (**c**) 24-month follow-up, and (**d**) All phases combined.

**Appendix 1—table 2.** Age stratified baseline prevalence of health outcomes.

| | Baseline Prevalence | | |
|---|---|---|---|
| | 1–11 months | 12–23 months | 24–48 months |
| **Any bacterial or protozoan infection** | | | |
| All children | 108/208 (52%) | 179/221 (81%) | 277/297 (93%) |
| Control | 57/109 (52%) | 101/119 (85%) | 143/152 (94%) |
| Intervention | 51/99 (52%) | 78/102 (76%) | 134/145 (92%) |
| **Any STH infection** | | | |
| All children | 30/185 (16%) | 89/203 (44%) | 171/277 (62%) |
| Control | 17/93 (18%) | 50/112 (45%) | 94/144 (65%) |
| Intervention | 13/92 (14%) | 39/91 (43%) | 77/133 (58%) |
| **Diarrhea** | | | |
| All children | 37/258 (14%) | 52/264 (20%) | 36/427 (8.4%) |
| Control | 19/138 (14%) | 27/146 (18%) | 20/234 (8.6%) |
| Intervention | 18/120 (15%) | 25/118 (21%) | 16/193 (8.3%) |
| **Any bacterial infection** | | | |
| All children | 94/208 (45%) | 150/221 (68%) | 229/297 (77%) |

*Continued on next page*

*Appendix 1—table 2 continued*

| | Baseline Prevalence | | |
| | 1–11 months | 12–23 months | 24–48 months |
| --- | --- | --- | --- |
| Intervention | 53/109 (49%) | 89/119 (75%) | 117/152 (77%) |
| All children | 41/99 (41%) | 61/102 (60%) | 112/145 (77%) |
| **Shigella** | | | |
| All children | 19/208 (9.1%) | 97/221 (44%) | 192/297 (65%) |
| Control | 10/109 (9.2%) | 57/119 (48%) | 101/152 (66%) |
| Intervention | 9/99 (9.1%) | 40/102 (39%) | 91/145 (63%) |
| **ETEC** | | | |
| All children | 47/208 (23%) | 81/221 (37%) | 90/297 (30%) |
| Control | 25/109 (23%) | 45/119 (38%) | 43/152 (28%) |
| Intervention | 22/99 (22%) | 36/102 (35%) | 47/145 (32%) |
| **Campylobacter** | | | |
| All children | 22/208 (11%) | 19/221 (8.6%) | 16/297 (5.4%) |
| Control | 14/109 (13%) | 13/119 (11%) | 10/152 (6.6%) |
| Intervention | 8/99 (8.1%) | 6/102 (5.9%) | 6/145 (4.1%) |
| **C. difficile** | | | |
| All children | 23/208 (11%) | 10/221 (4.5%) | 2/297 (0.67%) |
| Control | 13/109 (12%) | 7/119 (5.9%) | 2/152 (1.3%) |
| Intervention | 10/99 (10%) | 3/102 (2.9%) | 0/145 (0.0%) |
| **E. coli o157** | | | |
| All children | 6/208 (2.9%) | 10/221 (4.5%) | 15/297 (5%) |
| Control | 4/109 (3.7%) | 3/119 (2.5%) | 6/152 (4%) |
| Intervention | 2/99 (2%) | 7/102 (6.9%) | 9/145 (6.2%) |
| **STEC** | | | |
| All children | 3/208 (1.4%) | 7/221 (3.2%) | 3/297 (1%) |
| Control | 0/109 (0.0%) | 1/119 (0.84%) | 2/152 (1.3%) |
| Intervention | 3/99 (3%) | 6/102 (5.9%) | 1/145 (0.69%) |
| **Y. enterocolitica** | | | |
| All children | 0/208 (0.0%) | 1/221 (0.45%) | 0/297 (0.0%) |
| Control | 0/109 (0.0%) | 0/119 (0.0%) | 0/152 (0.0%) |
| Intervention | 0/99 (0.0%) | 1/102 (0.98%) | 0/145 (0.0%) |
| **V. cholerae** | | | |
| All children | 0/208 (0.0%) | 0/221 (0.0%) | 0/297 (0.0%) |
| Control | 0/109 (0.0%) | 0/119 (0.0%) | 0/152 (0.0%) |
| Intervention | 0/99 (0.0%) | 0/102 (0.0%) | 0/145 (0.0%) |
| **Any Protozoa** | | | |
| All children | 36/208 (17%) | 120/221 (54%) | 223/297 (75%) |
| Control | 14/109 (13%) | 68/119 (57%) | 114/152 (75%) |
| Intervention | 22/99 (22%) | 52/102 (51%) | 109/145 (75%) |
| **Giardia** | | | |
| All children | 28/208 (13%) | 119/221 (54%) | 219/297 (74%) |
| Control | 12/109 (11%) | 67/119 (56%) | 113/152 (74%) |
| Intervention | 16/99 (16%) | 52/102 (51%) | 106/145 (73%) |
| **Cryptosporidium** | | | |

*Continued on next page*

*Appendix 1—table 2 continued*

| | Baseline Prevalence | | |
| | 1–11 months | 12–23 months | 24–48 months |
| --- | --- | --- | --- |
| All children | 10/208 (4.8%) | 9/221 (4.1%) | 5/297 (1.7%) |
| Control | 2/109 (1.8%) | 5/119 (4.2%) | 1/152 (0.66%) |
| Intervention | 8/99 (8.1%) | 4/102 (3.9%) | 4/145 (2.8%) |
| *E. histolytica* | | | |
| All children | 1/208 (0.48%) | 0/221 (0.0%) | 3/297 (1%) |
| Control | 0/109 (0.0%) | 0/119 (0.0%) | 0/152 (0.0%) |
| Intervention | 1/99 (1%) | 0/102 (0.0%) | 3/145 (2.1%) |
| **Any virus** | | | |
| All children | 36/208 (17%) | 34/221 (15%) | 33/297 (11%) |
| Control | 15/109 (14%) | 19/119 (16%) | 19/152 (13%) |
| Intervention | 21/99 (21%) | 15/102 (15%) | 14/145 (9.7%) |
| **Norovirus GI/GII** | | | |
| All children | 27/208 (13%) | 25/221 (11%) | 23/297 (7.7%) |
| Control | 12/109 (11%) | 14/119 (12%) | 12/152 (7.9%) |
| Intervention | 15/99 (15%) | 11/102 (11%) | 11/145 (7.6%) |
| **Adenovirus 40/41** | | | |
| All children | 7/208 (3.4%) | 7/221 (3.2%) | 8/297 (2.7%) |
| Control | 4/109 (3.7%) | 3/119 (2.5%) | 6/152 (4%) |
| Intervention | 3/99 (3%) | 4/102 (3.9%) | 2/145 (1.4%) |
| **Rotavirus A** | | | |
| All children | 3/208 (1.4%) | 5/221 (2.3%) | 2/297 (0.67%) |
| Control | 0/109 (0.0%) | 2/119 (1.7%) | 1/152 (0.66%) |
| Intervention | 3/99 (3%) | 3/102 (2.9%) | 1/145 (0.69%) |
| **Coinfection, ≥2 GPP pathogens** | | | |
| All children | 48/208 (23%) | 118/221 (53%) | 203/297 (68%) |
| Control | 23/109 (21%) | 69/119 (58%) | 104/152 (68%) |
| Intervention | 25/99 (25%) | 49/102 (48%) | 99/145 (68%) |
| *Trichuris* | | | |
| All children | 20/185 (11%) | 69/203 (34%) | 150/277 (54%) |
| Control | 10/93 (11%) | 38/112 (34%) | 82/144 (57%) |
| Intervention | 10/92 (11%) | 31/91 (34%) | 68/133 (51%) |
| *Ascaris* | | | |
| All children | 21/185 (11%) | 53/203 (26%) | 81/277 (29%) |
| Control | 12/93 (13%) | 33/112 (29%) | 47/144 (33%) |
| Intervention | 9/92 (9.8%) | 20/91 (22%) | 34/133 (26%) |
| **Coinfection, ≥2 STH** | | | |
| All children | 11/185 (6%) | 33/203 (16%) | 60/277 (22%) |
| Control | 5/93 (5.4%) | 21/112 (19%) | 35/144 (24%) |
| Intervention | 6/92 (6.5%) | 12/91 (13%) | 25/133 (19%) |
| **Number of GPP infections** | | | |
| All children | 0.94 (1.1) | 1.8 (1.2) | 1.9 (0.95) |
| Control | 0.88 (1.1) | 1.8 (1.1) | 2 (0.93) |
| Intervention | 1 (1.1) | 1.7 (1.3) | 1.9 (0.98) |

*Continued on next page*

*Appendix 1—table 2 continued*

| | Baseline Prevalence | | |
| --- | --- | --- | --- |
| | **1–11 months** | **12–23 months** | **24–48 months** |
| **Number of STH infections** | | | |
| All children | 0.23 (0.55) | 0.61 (0.75) | 0.86 (0.76) |
| Control | 0.24 (0.54) | 0.64 (0.78) | 0.9 (0.76) |
| Intervention | 0.23 (0.56) | 0.57 (0.72) | 0.8 (0.76) |

Data presented n/N (%) or mean (standard deviation). All bacterial, protozoan, and viral pathogens were measured using the Luminex Gastrointestinal Pathogen panel. STH were measured using the Kato-Katz method. Diarrhea was measured via caregiver report in household surveys.

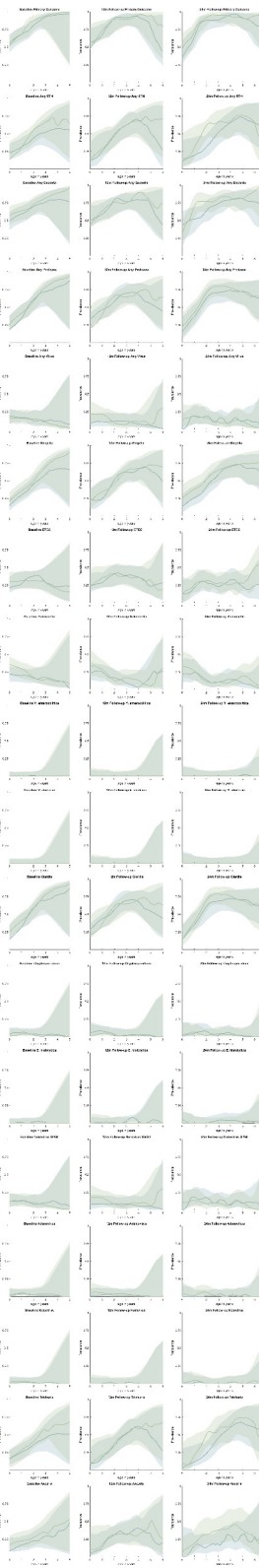

**Appendix 1—figure 4.** Prevalence of pathogens by age at baseline, 12-month, and 24-month phases. Results are smoothed averages stratified by study arm with 95% confidence intervals represented by shaded areas.

**Appendix 1—table 3.** Baseline enrollment characteristics of children with and without repeated measures at the 12-month phase.
Results are presented for all children combined and stratified by study arm.

| | All children | | | Control | | | Intervention | | |
|---|---|---|---|---|---|---|---|---|---|
| | BL and 12M* | BL only† | Std. diff.‡ | BL and 12M | BL only | Std. diff. | BL and 12M | BL only | Std. diff. |
| **Outcomes** | | | | | | | | | |
| Diarrhea | 83/609 (14%) | 43/365 (12%) | 0.06 | 38/310 (12%) | 29/216 (13%) | 0.03 | 45/299 (15%) | 14/149 (9.4%) | 0.17 |
| Any bacterial or protozoan infection | 376/485 (78%) | 215/268 (80%) | 0.07 | 184/234 (79%) | 129/158 (82%) | 0.08 | 192/251 (76%) | 86/110 (78%) | 0.04 |
| Any GPP infection | 390/485 (80%) | 225/268 (84%) | 0.09 | 188/234 (80%) | 135/158 (85%) | 0.14 | 202/251 (80%) | 90/110 (82%) | 0.03 |
| Any bacterial infection | 311/485 (64%) | 187/268 (70%) | 0.12 | 157/234 (67%) | 114/158 (72%) | 0.11 | 154/251 (61%) | 73/110 (66%) | 0.10 |
| Shigella | 200/485 (41%) | 131/268 (49%) | 0.15 | 101/234 (43%) | 78/158 (49%) | 0.12 | 99/251 (39%) | 53/110 (48%) | 0.18 |
| ETEC | 147/485 (30%) | 79/268 (29%) | 0.02 | 68/234 (29%) | 48/158 (30%) | 0.03 | 79/251 (31%) | 31/110 (28%) | 0.07 |
| Campylobacter | 37/485 (7.6%) | 23/268 (8.6%) | 0.03 | 22/234 (9.4%) | 17/158 (11%) | 0.05 | 15/251 (6%) | 6/110 (5.5%) | 0.02 |
| C. difficile | 23/485 (4.7%) | 12/268 (4.5%) | 0.01 | 15/234 (6.4%) | 7/158 (4.4%) | 0.09 | 8/251 (3.2%) | 5/110 (4.5%) | 0.07 |
| E. coli O157 | 19/485 (3.9%) | 12/268 (4.5%) | 0.03 | 9/234 (3.9%) | 4/158 (2.5%) | 0.07 | 10/251 (4%) | 8/110 (7.3%) | 0.14 |
| STEC | 7/485 (1.4%) | 6/268 (2.2%) | 0.06 | 1/234 (0.43%) | 2/158 (1.3%) | 0.09 | 6/251 (2.4%) | 4/110 (3.6%) | 0.07 |
| Any protozoan infection | 257/485 (53%) | 143/268 (53%) | 0.01 | 126/234 (54%) | 79/158 (50%) | 0.08 | 131/251 (52%) | 64/110 (58%) | 0.12 |
| Giardia | 247/485 (51%) | 140/268 (52%) | 0.03 | 122/234 (52%) | 79/158 (50%) | 0.04 | 125/251 (50%) | 61/110 (55%) | 0.11 |
| Cryptosporidium | 20/485 (4.1%) | 4/268 (1.5%) | 0.16 | 7/234 (3%) | 1/158 (0.63%) | 0.18 | 13/251 (5.2%) | 3/110 (2.7%) | 0.13 |
| E. histolytica | 2/485 (0.41%) | 2/268 (0.75%) | 0.04 | 0/234 (0.0%) | 0/158 (0.0%) | . .** | 2/251 (0.80%) | 2/110 (1.8%) | 0.09 |
| Any viral infection | 66/485 (14%) | 39/268 (15%) | 0.03 | 31/234 (13%) | 22/158 (14%) | 0.02 | 35/251 (14%) | 17/110 (15%) | 0.04 |
| Adenovirus 40/41 | 14/485 (2.9%) | 8/268 (3%) | 0.01 | 8/234 (3.4%) | 5/158 (3.2%) | 0.01 | 6/251 (2.4%) | 3/110 (2.7%) | 0.02 |
| Norovirus GI/GII | 50/485 (10%) | 27/268 (10%) | 0.01 | 23/234 (9.8%) | 15/158 (9.5%) | 0.01 | 27/251 (11%) | 12/110 (11%) | 0.00 |
| Rotavirus A | 5/485 (1%) | 5/268 (1.9%) | 0.07 | 1/234 (0.43%) | 2/158 (1.3%) | 0.09 | 4/251 (1.6%) | 3/110 (2.7%) | 0.08 |
| Coinfection, ≥2 GPP infections | 251/485 (52%) | 140/268 (52%) | 0.01 | 126/234 (54%) | 80/158 (51%) | 0.06 | 125/251 (50%) | 60/110 (55%) | 0.10 |
| Any STH infection | 202/447 (45%) | 106/242 (44%) | 0.03 | 106/218 (49%) | 64/142 (45%) | 0.07 | 96/229 (42%) | 42/100 (42%) | 0.00 |
| Ascaris | 109/447 (24%) | 54/242 (22%) | 0.05 | 65/218 (30%) | 30/142 (21%) | 0.20 | 44/229 (19%) | 24/100 (24%) | 0.12 |
| Trichuris | 170/447 (38%) | 86/242 (36%) | 0.05 | 85/218 (39%) | 54/142 (38%) | 0.02 | 85/229 (37%) | 32/100 (32%) | 0.11 |
| Coinfection,≥2 STH infections | 77/447 (17%) | 34/242 (14%) | 0.09 | 44/218 (20%) | 20/142 (14%) | 0.16 | 33/229 (14%) | 14/100 (14%) | 0.01 |
| Number of GPP infections | 1.6 (1.1) | 1.7 (1.1) | 0.07 | 1.6 (1.1) | 1.6 (1.1) | 0.02 | 1.6 (1.1) | 1.7 (1.2) | 0.14 |

*Continued on next page*

*Appendix 1—table 3 continued*

| | All children | | | Control | | | Intervention | | |
|---|---|---|---|---|---|---|---|---|---|
| | BL and 12M* | BL only† | Std. diff.‡ | BL and 12M | BL only | Std. diff. | BL and 12M | BL only | Std. diff. |
| Number of STH infections | 0.64 (0.77) | 0.58 (0.73) | 0.08 | 0.7 (0.79) | 0.59 (0.73) | 0.14 | 0.59 (0.75) | 0.57 (0.73) | 0.03 |
| **Child-, household-, compound-level characteristics** | | | | | | | | | |
| Child sex, female | 319/614 (52%) | 174/350 (50%) | 0.04 | 169/312 (54%) | 97/208 (47%) | 0.15 | 150/302 (50%) | 77/142 (54%) | 0.09 |
| Child breastfed | 206/609 (34%) | 106/365 (29%) | 0.10 | 107/310 (35%) | 62/216 (29%) | 0.13 | 99/299 (33%) | 44/149 (30%) | 0.08 |
| Child exclusively breastfed | 51/609 (8.4%) | 35/365 (9.6%) | 0.04 | 27/310 (8.7%) | 22/216 (10%) | 0.05 | 24/299 (8%) | 13/149 (8.7%) | 0.03 |
| Child age at survey, days | 697 (409) | 697 (396) | 0.00 | 698 (409) | 703 (400) | 0.01 | 696 (409) | 689 (391) | 0.02 |
| Child age at sampling, days | 668 (399) | 656 (382) | 0.03 | 661 (397) | 655 (395) | 0.02 | 674 (402) | 657 (364) | 0.04 |
| Child wears diapers | 402/609 (66%) | 234/364 (64%) | 0.04 | 209/310 (67%) | 133/216 (62%) | 0.12 | 193/299 (65%) | 101/148 (68%) | 0.08 |
| Child feces disposed in latrine | 173/609 (28%) | 116/365 (32%) | 0.07 | 79/310 (25%) | 69/216 (32%) | 0.14 | 94/299 (31%) | 47/149 (32%) | 0.00 |
| Caregiver completed primary school | 333/614 (54%) | 193/365 (53%) | 0.03 | 163/312 (52%) | 124/216 (57%) | 0.10 | 170/302 (56%) | 69/149 (46%) | 0.20 |
| Mother alive | 576/590 (98%) | 353/358 (99%) | 0.07 | 295/301 (98%) | 208/212 (98%) | 0.01 | 281/289 (97%) | 145/146 (99%) | 0.16 |
| Respondent is child's mother | 414/605 (68%) | 238/357 (67%) | 0.04 | 222/307 (72%) | 146/212 (69%) | 0.08 | 192/298 (64%) | 92/145 (63%) | 0.02 |
| Household floors covered | 575/615 (94%) | 349/368 (95%) | 0.06 | 300/313 (96%) | 211/217 (97%) | 0.08 | 275/302 (91%) | 138/151 (91%) | 0.01 |
| Household walls made of sturdy material | 399/615 (65%) | 243/368 (66%) | 0.02 | 216/313 (69%) | 154/217 (71%) | 0.04 | 183/302 (61%) | 89/151 (59%) | 0.03 |
| Latrine has drop-hole | 359/604 (59%) | 193/364 (53%) | 0.13 | 169/307 (55%) | 109/214 (51%) | 0.08 | 190/297 (64%) | 84/150 (56%) | 0.16 |
| Latrine has vent-pipe | 93/605 (15%) | 44/364 (12%) | 0.10 | 21/308 (6.8%) | 12/214 (5.6%) | 0.05 | 72/297 (24%) | 32/150 (21%) | 0.07 |
| Latrine has ceramic or concrete slab or pedestal | 224/602 (37%) | 133/363 (37%) | 0.01 | 101/305 (33%) | 80/213 (38%) | 0.09 | 123/297 (41%) | 53/150 (35%) | 0.13 |
| Latrine has sturdy walls | 193/605 (32%) | 110/363 (30%) | 0.03 | 84/306 (27%) | 58/215 (27%) | 0.01 | 109/299 (36%) | 52/148 (35%) | 0.03 |
| Water tap on compound grounds | 468/606 (77%) | 285/364 (78%) | 0.03 | 224/308 (73%) | 162/214 (76%) | 0.07 | 244/298 (82%) | 123/150 (82%) | 0.00 |
| Household crowding, ≥3 persons/room | 122/615 (20%) | 45/368 (12%) | 0.21 | 55/313 (18%) | 22/217 (10%) | 0.22 | 67/302 (22%) | 23/151 (15%) | 0.18 |
| Compound electricity normally functions | 556/615 (90%) | 331/372 (89%) | 0.05 | 272/313 (87%) | 195/220 (89%) | 0.05 | 284/302 (94%) | 136/152 (89%) | 0.17 |
| Standing water observed in compound | 44/605 (7.3%) | 26/363 (7.2%) | 0.00 | 7/306 (2.3%) | 7/215 (3.3%) | 0.06 | 37/299 (12%) | 19/148 (13%) | 0.01 |
| Leaking or standing wastewater observed in compound | 371/605 (61%) | 233/363 (64%) | 0.06 | 214/306 (70%) | 149/215 (69%) | 0.01 | 157/299 (53%) | 84/148 (57%) | 0.09 |
| Any animal observed | 395/615 (64%) | 226/372 (61%) | 0.07 | 189/313 (60%) | 129/220 (59%) | 0.04 | 206/302 (68%) | 97/152 (64%) | 0.09 |
| Dog observed | 51/615 (8.3%) | 23/372 (6.2%) | 0.08 | 18/313 (5.8%) | 10/220 (4.5%) | 0.05 | 33/302 (11%) | 13/152 (8.6%) | 0.08 |
| Chicken or duck observed | 94/615 (15%) | 36/372 (9.7%) | 0.17 | 43/313 (14%) | 27/220 (12%) | 0.04 | 51/302 (17%) | 9/152 (5.9%) | 0.35 |

*Continued on next page*

*Appendix 1—table 3 continued*

| | All children | | | Control | | | Intervention | | |
|---|---|---|---|---|---|---|---|---|---|
| | BL and 12M* | BL only† | Std. diff.‡ | BL and 12M | BL only | Std. diff. | BL and 12M | BL only | Std. diff. |
| Cat observed | 341/615 (55%) | 205/372 (55%) | 0.01 | 167/313 (53%) | 120/220 (55%) | 0.02 | 174/302 (58%) | 85/152 (56%) | 0.03 |
| Faeces or used diapers observed around compound | 276/605 (46%) | 177/363 (49%) | 0.06 | 166/306 (54%) | 116/215 (54%) | 0.01 | 110/299 (37%) | 61/148 (41%) | 0.09 |
| Compound floods during rain | 377/615 (61%) | 226/372 (61%) | 0.01 | 211/313 (67%) | 137/220 (62%) | 0.11 | 166/302 (55%) | 89/152 (59%) | 0.07 |
| Number of household members | 6.4 (3.3) | 5.6 (2.6) | 0.27 | 6 (3) | 5.2 (2.1) | 0.33 | 6.8 (3.5) | 6.3 (3.1) | 0.18 |
| Household wealth score, 0–1 | 0.43 (0.1) | 0.44 (0.099) | 0.10 | 0.44 (0.1) | 0.45 (0.097) | 0.15 | 0.43 (0.1) | 0.43 (0.1) | 0.01 |
| Number of households in compound | 5.2 (4.6) | 4.7 (4.4) | 0.11 | 4.4 (2.9) | 3.8 (1.7) | 0.21 | 6.1 (5.6) | 6 (6.4) | 0.02 |
| Compound population | 21 (15) | 19 (14) | 0.18 | 17 (8.1) | 15 (6.1) | 0.22 | 26 (18) | 24 (20) | 0.11 |
| Number of water taps in compound | 1.5 (2.2) | 1.2 (1) | 0.22 | 1 (1.1) | 0.97 (0.83) | 0.04 | 2.1 (2.8) | 1.4 (1.2) | 0.30 |
| Number of latrines/drop-holes in compound | 1.1 (0.63) | 1.1 (0.65) | 0.00 | 1 (0.24) | 1 (0.2) | 0.04 | 1.2 (0.86) | 1.3 (0.97) | 0.03 |
| Compound population density | 0.084 (0.046) | 0.078 (0.045) | 0.13 | 0.076 (0.04) | 0.07 (0.039) | 0.14 | 0.092 (0.051) | 0.089 (0.05) | 0.06 |

Results are presented as prevalence (n/N (%)) or mean (standard deviation) at baseline.

*Prevalence (or mean (SD)) for children with repeated observations at baseline and 12-month visits.

†Prevalence (or mean (SD)) for children with observations at baseline visit and not the 12-month visit.

‡Standardized mean difference between observations of children with and without repeated measures at baseline and 12-month visits.

§ Could not be calculated.

**Appendix 1—table 4.** Baseline enrollment characteristics of children with and without repeated measures at the 24-month phase.
Results are presented for all children combined and stratified by study arm.

| | All children | | | Control | | | Intervention | | |
|---|---|---|---|---|---|---|---|---|---|
| | BL and 24M* | BL only† | Std. Diff.‡ | BL and 24M | BL only | Std. Diff. | BL and 24M | BL only | Std. Diff. |
| **Outcomes** | | | | | | | | | |
| Diarrhea | 75/504 (15%) | 51/470 (11%) | 0.12 | 35/244 (14%) | 32/282 (11%) | 0.09 | 40/260 (15%) | 19/188 (10%) | 0.16 |
| Any bacterial or protozoan infection | 310/394 (79%) | 281/359 (78%) | 0.01 | 144/183 (79%) | 169/209 (81%) | 0.05 | 166/211 (79%) | 112/150 (75%) | 0.09 |
| Any GPP infection | 322/394 (82%) | 293/359 (82%) | 0.00 | 148/183 (81%) | 175/209 (84%) | 0.07 | 174/211 (82%) | 118/150 (79%) | 0.10 |
| Any bacterial infection | 251/394 (64%) | 247/359 (69%) | 0.11 | 120/183 (66%) | 151/209 (72%) | 0.14 | 131/211 (62%) | 96/150 (64%) | 0.04 |
| *Shigella* | 158/394 (40%) | 173/359 (48%) | 0.16 | 74/183 (40%) | 105/209 (50%) | 0.20 | 84/211 (40%) | 68/150 (45%) | 0.11 |
| ETEC | 115/394 (29%) | 111/359 (31%) | 0.04 | 53/183 (29%) | 63/209 (30%) | 0.03 | 62/211 (29%) | 48/150 (32%) | 0.06 |
| *Campylobacter* | 31/394 (7.9%) | 29/359 (8.1%) | 0.01 | 18/183 (9.8%) | 21/209 (10%) | 0.01 | 13/211 (6.2%) | 8/150 (5.3%) | 0.04 |

*Continued on next page*

*Appendix 1—table 4 continued*

| | All children | | | Control | | | Intervention | | |
|---|---|---|---|---|---|---|---|---|---|
| | BL and 24M* | BL only† | Std. Diff.‡ | BL and 24M | BL only | Std. Diff. | BL and 24M | BL only | Std. Diff. |
| *C. difficile* | 18/394 (4.6%) | 17/359 (4.7%) | 0.01 | 10/183 (5.5%) | 12/209 (5.7%) | 0.01 | 8/211 (3.8%) | 5/150 (3.3%) | 0.02 |
| *E. coli* O157 | 17/394 (4.3%) | 14/359 (3.9%) | 0.02 | 7/183 (3.8%) | 6/209 (2.9%) | 0.05 | 10/211 (4.7%) | 8/150 (5.3%) | 0.03 |
| STEC | 6/394 (1.5%) | 7/359 (1.9%) | 0.03 | 2/183 (1.1%) | 1/209 (0.48%) | 0.07 | 4/211 (1.9%) | 6/150 (4%) | 0.12 |
| Any protozoan infection | 214/394 (54%) | 186/359 (52%) | 0.05 | 96/183 (52%) | 109/209 (52%) | 0.01 | 118/211 (56%) | 77/150 (51%) | 0.09 |
| *Giardia* | 204/394 (52%) | 183/359 (51%) | 0.02 | 92/183 (50%) | 109/209 (52%) | 0.04 | 112/211 (53%) | 74/150 (49%) | 0.08 |
| *Cryptosporidium* | 20/394 (5.1%) | 4/359 (1.1%) | 0.23 | 7/183 (3.8%) | 1/209 (0.48%) | 0.23 | 13/211 (6.2%) | 3/150 (2%) | 0.21 |
| *E. histolytica* | 2/394 (0.51%) | 2/359 (0.56%) | 0.01 | 0/183 (0.0%) | 0/209 (0.0%) | ..§ | 2/211 (0.95%) | 2/150 (1.3%) | 0.04 |
| Any viral infection | 55/394 (14%) | 50/359 (14%) | 0.00 | 22/183 (12%) | 31/209 (15%) | 0.08 | 33/211 (16%) | 19/150 (13%) | 0.09 |
| Adenovirus 40/41 | 14/394 (3.5%) | 8/359 (2.2%) | 0.08 | 7/183 (3.8%) | 6/209 (2.9%) | 0.05 | 7/211 (3.3%) | 2/150 (1.3%) | 0.13 |
| Norovirus GI/GII | 42/394 (11%) | 35/359 (9.8%) | 0.03 | 15/183 (8.2%) | 23/209 (11%) | 0.10 | 27/211 (13%) | 12/150 (8%) | 0.16 |
| Rotavirus A | 3/394 (0.76%) | 7/359 (1.9%) | 0.10 | 1/183 (0.55%) | 2/209 (0.96%) | 0.05 | 2/211 (0.95%) | 5/150 (3.3%) | 0.17 |
| Coinfection,≥2 GPP infections | 206/394 (52%) | 185/359 (52%) | 0.02 | 97/183 (53%) | 109/209 (52%) | 0.02 | 109/211 (52%) | 76/150 (51%) | 0.02 |
| Any STH infection | 156/362 (43%) | 152/327 (46%) | 0.07 | 80/171 (47%) | 90/189 (48%) | 0.02 | 76/191 (40%) | 62/138 (45%) | 0.10 |
| *Ascaris* | 85/362 (23%) | 78/327 (24%) | 0.01 | 50/171 (29%) | 45/189 (24%) | 0.12 | 35/191 (18%) | 33/138 (24%) | 0.14 |
| *Trichuris* | 128/362 (35%) | 128/327 (39%) | 0.08 | 63/171 (37%) | 76/189 (40%) | 0.07 | 65/191 (34%) | 52/138 (38%) | 0.08 |
| Coinfection,≥2 STH infections | 57/362 (16%) | 54/327 (17%) | 0.02 | 33/171 (19%) | 31/189 (16%) | 0.08 | 24/191 (13%) | 23/138 (17%) | 0.12 |
| Number of GPP infections | 1.6 (1.1) | 1.6 (1.2) | 0.04 | 1.6 (1.1) | 1.7 (1.1) | 0.10 | 1.6 (1.1) | 1.6 (1.2) | 0.01 |
| Number of STH infections | 0.61 (0.75) | 0.64 (0.76) | 0.04 | 0.67 (0.78) | 0.65 (0.75) | 0.03 | 0.55 (0.72) | 0.63 (0.77) | 0.10 |
| **Child-, household-, compound-level characteristics** | | | | | | | | | |
| Child sex, female | 260/503 (52%) | 233/461 (51%) | 0.02 | 124/241 (51%) | 142/279 (51%) | 0.01 | 136/262 (52%) | 91/182 (50%) | 0.04 |
| Child breastfed | 172/504 (34%) | 140/470 (30%) | 0.09 | 87/244 (36%) | 82/282 (29%) | 0.14 | 85/260 (33%) | 58/188 (31%) | 0.04 |
| Child exclusively breastfed | 35/504 (6.9%) | 51/470 (11%) | 0.14 | 19/244 (7.8%) | 30/282 (11%) | 0.10 | 16/260 (6.2%) | 21/188 (11%) | 0.18 |
| Child age at survey, days | 698 (403) | 696 (405) | 0.01 | 689 (400) | 709 (410) | 0.05 | 707 (406) | 675 (398) | 0.08 |
| Child age at sampling, days | 675 (406) | 651 (379) | 0.06 | 666 (403) | 652 (390) | 0.04 | 682 (409) | 650 (364) | 0.08 |
| Child wears diapers | 343/504 (68%) | 293/469 (62%) | 0.12 | 171/244 (70%) | 171/282 (61%) | 0.20 | 172/260 (66%) | 122/187 (65%) | 0.02 |
| Child feces disposed in latrine | 138/504 (27%) | 151/470 (32%) | 0.10 | 57/244 (23%) | 91/282 (32%) | 0.20 | 81/260 (31%) | 60/188 (32%) | 0.02 |

*Continued on next page*

Appendix 1—table 4 continued

| | All children | | | Control | | | Intervention | | |
|---|---|---|---|---|---|---|---|---|---|
| | BL and 24M* | BL only† | Std. Diff.‡ | BL and 24M | BL only | Std. Diff. | BL and 24M | BL only | Std. Diff. |
| Caregiver completed primary school | 274/507 (54%) | 252/472 (53%) | 0.01 | 131/245 (53%) | 156/283 (55%) | 0.03 | 143/262 (55%) | 96/189 (51%) | 0.08 |
| Mother alive | 474/486 (98%) | 455/462 (98%) | 0.07 | 232/236 (98%) | 271/277 (98%) | 0.03 | 242/250 (97%) | 184/185 (99%) | 0.20 |
| Respondent is child's mother | 337/500 (67%) | 315/462 (68%) | 0.02 | 173/241 (72%) | 195/278 (70%) | 0.04 | 164/259 (63%) | 120/184 (65%) | 0.04 |
| Household floors covered | 469/507 (93%) | 455/476 (96%) | 0.13 | 233/245 (95%) | 278/285 (98%) | 0.13 | 236/262 (90%) | 177/191 (93%) | 0.09 |
| Household walls made of sturdy material | 337/507 (66%) | 305/476 (64%) | 0.05 | 184/245 (75%) | 186/285 (65%) | 0.22 | 153/262 (58%) | 119/191 (62%) | 0.08 |
| Latrine has drop-hole | 294/497 (59%) | 258/471 (55%) | 0.09 | 133/239 (56%) | 145/282 (51%) | 0.08 | 161/258 (62%) | 113/189 (60%) | 0.05 |
| Latrine has vent-pipe | 80/497 (16%) | 57/472 (12%) | 0.12 | 18/239 (7.5%) | 15/283 (5.3%) | 0.09 | 62/258 (24%) | 42/189 (22%) | 0.04 |
| Latrine has ceramic or concrete slab or pedestal | 184/494 (37%) | 173/471 (37%) | 0.01 | 77/236 (33%) | 104/282 (37%) | 0.09 | 107/258 (41%) | 69/189 (37%) | 0.10 |
| Latrine has sturdy walls | 165/501 (33%) | 138/467 (30%) | 0.07 | 67/240 (28%) | 75/281 (27%) | 0.03 | 98/261 (38%) | 63/186 (34%) | 0.08 |
| Water tap on compound grounds | 389/498 (78%) | 364/472 (77%) | 0.02 | 171/239 (72%) | 215/283 (76%) | 0.10 | 218/259 (84%) | 149/189 (79%) | 0.14 |
| Household crowding,≥3 persons/room | 114/507 (22%) | 53/476 (11%) | 0.31 | 45/245 (18%) | 32/285 (11%) | 0.20 | 69/262 (26%) | 21/191 (11%) | 0.40 |
| Compound electricity normally functions | 454/507 (90%) | 433/480 (90%) | 0.02 | 214/245 (87%) | 253/288 (88%) | 0.02 | 240/262 (92%) | 180/192 (94%) | 0.08 |
| Standing water observed in compound | 39/501 (7.8%) | 31/467 (6.6%) | 0.04 | 7/240 (2.9%) | 7/281 (2.5%) | 0.03 | 32/261 (12%) | 24/186 (13%) | 0.02 |
| Leaking or standing wastewater observed in compound | 308/501 (61%) | 296/467 (63%) | 0.04 | 164/240 (68%) | 199/281 (71%) | 0.05 | 144/261 (55%) | 97/186 (52%) | 0.06 |
| Any animal observed | 337/507 (66%) | 284/480 (59%) | 0.15 | 156/245 (64%) | 162/288 (56%) | 0.15 | 181/262 (69%) | 122/192 (64%) | 0.12 |
| Dog observed | 49/507 (9.7%) | 25/480 (5.2%) | 0.17 | 17/245 (6.9%) | 11/288 (3.8%) | 0.14 | 32/262 (12%) | 14/192 (7.3%) | 0.17 |
| Chicken or duck observed | 71/507 (14%) | 59/480 (12%) | 0.05 | 32/245 (13%) | 38/288 (13%) | 0.00 | 39/262 (15%) | 21/192 (11%) | 0.12 |
| Cat observed | 294/507 (58%) | 252/480 (53%) | 0.11 | 143/245 (58%) | 144/288 (50%) | 0.17 | 151/262 (58%) | 108/192 (56%) | 0.03 |
| Feces or used diapers observed around compound | 218/501 (44%) | 235/467 (50%) | 0.14 | 120/240 (50%) | 162/281 (58%) | 0.15 | 98/261 (38%) | 73/186 (39%) | 0.03 |
| Compound floods during rain | 310/507 (61%) | 293/480 (61%) | 0.00 | 166/245 (68%) | 182/288 (63%) | 0.10 | 144/262 (55%) | 111/192 (58%) | 0.06 |
| Number of household members | 6.7 (3.4) | 5.5 (2.6) | 0.39 | 6.3 (3) | 5.2 (2.2) | 0.42 | 7.1 (3.6) | 6.1 (3) | 0.31 |
| Household wealth score, 0–1 | 0.43 (0.11) | 0.44 (0.097) | 0.12 | 0.44 (0.1) | 0.45 (0.095) | 0.10 | 0.42 (0.11) | 0.43 (0.1) | 0.11 |
| Number of households in compound | 5.3 (4.7) | 4.7 (4.3) | 0.13 | 4.4 (3.1) | 3.9 (1.8) | 0.21 | 6.1 (5.7) | 5.9 (6.2) | 0.03 |
| Compound population | 22 (15) | 18 (14) | 0.26 | 17 (8.1) | 15 (6.5) | 0.27 | 27 (18) | 23 (19) | 0.18 |
| Number of water taps in compound | 1.6 (2.2) | 1.2 (1.3) | 0.24 | 1 (1) | 0.99 (0.92) | 0.02 | 2.2 (2.8) | 1.4 (1.8) | 0.31 |
| Number of latrines in compound | 1.1 (0.62) | 1.1 (0.65) | 0.01 | 1 (0.25) | 1 (0.19) | 0.04 | 1.2 (0.82) | 1.3 (0.99) | 0.08 |
| Compound population density | 0.084 (0.049) | 0.079 (0.042) | 0.13 | 0.072 (0.038) | 0.075 (0.04) | 0.05 | 0.096 (0.055) | 0.084 (0.044) | 0.23 |

Results are presented as prevalence (n/N (%)) or mean (standard deviation) at baseline.

*Prevalence (or mean (SD)) for children with repeated observations at baseline and 24-month visits.

†Prevalence (or mean (SD)) for children with observations at the baseline visit and not the 24-month visit.

‡Standardized mean difference between observations of children with and without repeated measures at baseline and 24-month visits.

§Could not be calculated.

**Appendix 1—table 5.** Balance of characteristics measured at 12-month visits between children with repeat observations at baseline and 12-month and children with observations at the 12-month phase only.

| | All Children | | | Control | | | Intervention | | | |
|---|---|---|---|---|---|---|---|---|---|---|
| | BL and 12M* | 12M only† | Std. Diff.‡ | BL and 12M | 12M only | Std. Diff. | BL and 12M | 12M only | Std. Diff. | Std. Diff. Control v. Interv.§ |
| Child sex, female | 319/614 (52%) | 156/313 (50%) | 0.04 | 169/312 (54%) | 73/155 (47%) | 0.14 | 150/302 (50%) | 83/158 (53%) | 0.06 | 0.11 |
| Child breastfed | 27/562 (4.8%) | 161/305 (53%) | 1.25 | 13/280 (4.6%) | 76/151 (50%) | 1.19 | 14/282 (5%) | 85/154 (55%) | 1.31 | 0.10 |
| Child exclusively breastfed | 3/562 (0.53%) | 38/305 (12%) | 0.50 | 2/280 (0.71%) | 16/151 (11%) | 0.44 | 1/282 (0.35%) | 22/154 (14%) | 0.56 | 0.11 |
| Caregiver completed primary school | 305/614 (50%) | 144/309 (47%) | 0.06 | 156/312 (50%) | 62/153 (41%) | 0.19 | 149/302 (49%) | 82/156 (53%) | 0.06 | 0.24 |
| Child wears diapers | 83/563 (15%) | 194/305 (64%) | 1.16 | 40/281 (14%) | 92/151 (61%) | 1.10 | 43/282 (15%) | 102/154 (66%) | 1.21 | 0.11 |
| Respondent is child's mother | 365/563 (65%) | 236/305 (77%) | 0.28 | 188/281 (67%) | 121/151 (80%) | 0.30 | 177/282 (63%) | 115/154 (75%) | 0.26 | 0.13 |
| Household floors covered | 584/615 (95%) | 305/321 (95%) | 0.00 | 299/313 (96%) | 155/163 (95%) | 0.02 | 285/302 (94%) | 150/158 (95%) | 0.03 | 0.01 |
| Household walls made of sturdy material | 398/615 (65%) | 189/321 (59%) | 0.12 | 212/313 (68%) | 101/163 (62%) | 0.12 | 186/302 (62%) | 88/158 (56%) | 0.12 | 0.13 |
| Household crowding,≥3 persons/room | 210/615 (34%) | 106/321 (33%) | 0.02 | 111/313 (35%) | 54/163 (33%) | 0.05 | 99/302 (33%) | 52/158 (33%) | 0.00 | 0.00 |
| Compound electricity normally functions | 575/615 (94%) | 304/324 (94%) | 0.01 | 286/313 (91%) | 152/164 (93%) | 0.05 | 289/302 (96%) | 152/160 (95%) | 0.03 | 0.10 |
| Any animal observed | 505/611 (83%) | 275/324 (85%) | 0.06 | 235/309 (76%) | 131/164 (80%) | 0.09 | 270/302 (89%) | 144/160 (90%) | 0.02 | 0.29 |
| Dog observed | 134/611 (22%) | 81/324 (25%) | 0.07 | 57/309 (18%) | 37/164 (23%) | 0.10 | 77/302 (26%) | 44/160 (28%) | 0.05 | 0.11 |
| Chicken or duck observed | 77/611 (13%) | 42/324 (13%) | 0.01 | 34/309 (11%) | 18/164 (11%) | 0.00 | 43/302 (14%) | 24/160 (15%) | 0.02 | 0.12 |
| Cat observed | 469/611 (77%) | 249/324 (77%) | 0.00 | 218/309 (71%) | 118/164 (72%) | 0.03 | 251/302 (83%) | 131/160 (82%) | 0.03 | 0.24 |

*Continued on next page*

*Appendix 1—table 5 continued*

| | All Children | | | Control | | | Intervention | | | |
|---|---|---|---|---|---|---|---|---|---|---|
| | BL and 12M* | 12M only[†] | Std. Diff.[‡] | BL and 12M | 12M only | Std. Diff. | BL and 12M | 12M only | Std. Diff. | Std. Diff. Control v. Interv.[§] |
| Compound floods during rain | 220/615 (36%) | 119/324 (37%) | 0.02 | 132/313 (42%) | 64/164 (39%) | 0.06 | 88/302 (29%) | 55/160 (34%) | 0.11 | 0.10 |
| Child age at survey, days | 1114 (415) | 622 (502) | 1.07 | 1105 (413) | 684 (535) | 0.88 | 1122 (417) | 560 (461) | 1.28 | 0.25 |
| Child age at sampling, days | 1102 (417) | 605 (484) | 1.10 | 1080 (414) | 649 (516) | 0.92 | 1122 (420) | 563 (450) | 1.29 | 0.18 |
| Number of household members | 6.5 (3.2) | 6.3 (3.3) | 0.06 | 6.2 (3) | 6.4 (3.5) | 0.05 | 6.8 (3.3) | 6.2 (3.2) | 0.17 | 0.05 |
| Household wealth score, 0–1 | 0.4 (0.11) | 0.39 (0.11) | 0.02 | 0.4 (0.11) | 0.39 (0.11) | 0.12 | 0.39 (0.1) | 0.4 (0.1) | 0.10 | 0.11 |
| Number of households in compound | 5.2 (4.7) | 5.4 (5.5) | 0.04 | 4.2 (2.9) | 4 (2.3) | 0.09 | 6.3 (5.9) | 6.9 (7.3) | 0.09 | 0.53 |
| Compound population | 23 (22) | 24 (26) | 0.04 | 18 (9.7) | 18 (8.7) | 0.05 | 28 (29) | 30 (35) | 0.07 | 0.50 |
| Compound population density | 0.086 (0.049) | 0.084 (0.051) | 0.04 | 0.08 (0.043) | 0.078 (0.044) | 0.05 | 0.091 (0.054) | 0.089 (0.058) | 0.03 | 0.22 |

Results are presented as prevalence (n/N (%)) or mean (standard deviation) at 12-month visit.

*Prevalence (or mean (SD)) for children with repeated observations at baseline and 12-month visits.

[†]Prevalence (or mean (SD)) for children with observations at the 12-month visit only.

[‡]Standardized mean difference between observations of children with and without repeated measures at baseline and 12-month visits.

[§]Standardized mean difference between observations from control and intervention children measured at 12-month visit only.

**Appendix 1—table 6.** Balance of characteristics measured at 24-month visits between children with repeat observations at baseline and 24-month and children with observations at the 24-month phase only.

| | All Children | | | Control | | | Intervention | | | |
|---|---|---|---|---|---|---|---|---|---|---|
| | BL and 24M* | 24M only[†] | Std. Diff.[†] | BL and 24M | 24M only | Std. Diff. | BL and 24M | 24M only | Std. Diff. | Std. Diff. Control v. Interv.[§] |
| Child sex, female | 260/503 (52%) | 190/428 (44%) | 0.15 | 124/241 (51%) | 96/222 (43%) | 0.16 | 136/262 (52%) | 94/206 (46%) | 0.13 | 0.05 |
| Child breastfed | 0/418 (0.0%) | 129/381 (34%) | 1.01 | 0/195 (0.0%) | 68/194 (35%) | 1.04 | 0/223 (0.0%) | 61/187 (33%) | 0.98 | 0.05 |
| Child exclusively breastfed | 0/418 (0.0%) | 36/381 (9.4%) | 0.46 | 0/195 (0.0%) | 16/194 (8.3%) | 0.42 | 0/223 (0.0%) | 20/187 (11%) | 0.49 | 0.08 |
| Caregiver completed primary school | 199/507 (39%) | 164/427 (38%) | 0.02 | 88/245 (36%) | 82/221 (37%) | 0.02 | 111/262 (42%) | 82/206 (40%) | 0.05 | 0.06 |
| Child wears diapers | 3/419 (0.72%) | 196/381 (51%) | 1.42 | 1/196 (0.51%) | 101/194 (52%) | 1.44 | 2/223 (0.9%) | 95/187 (51%) | 1.39 | 0.03 |
| Respondent is child's mother | 259/419 (62%) | 298/381 (78%) | 0.36 | 129/196 (66%) | 161/194 (83%) | 0.40 | 130/223 (58%) | 137/187 (73%) | 0.32 | 0.24 |

*Continued on next page*

*Appendix 1—table 6 continued*

| | All Children | | | Control | | | Intervention | | | |
|---|---|---|---|---|---|---|---|---|---|---|
| | BL and 24M* | 24M only[†] | Std. Diff.[†] | BL and 24M | 24M only | Std. Diff. | BL and 24M | 24M only | Std. Diff. | Std. Diff. Control v. Interv.[§] |
| Household floors covered | 484/ 507 (95%) | 459/ 467 (98%) | 0.16 | 237/ 245 (97%) | 234/ 239 (98%) | 0.07 | 247/ 262 (94%) | 225/ 228 (99%) | 0.24 | 0.06 |
| Household walls made of sturdy material | 352/ 507 (69%) | 296/ 467 (63%) | 0.13 | 180/ 245 (73%) | 157/ 239 (66%) | 0.17 | 172/ 262 (66%) | 139/ 228 (61%) | 0.10 | 0.10 |
| Household crowding,≥3 persons/ room | 137/ 507 (27%) | 108/ 467 (23%) | 0.09 | 74/245 (30%) | 66/239 (28%) | 0.06 | 63/262 (24%) | 42/228 (18%) | 0.14 | 0.22 |
| Compound electricity normally functions | 485/ 507 (96%) | 472/ 494 (96%) | 0.01 | 230/ 245 (94%) | 237/ 254 (93%) | 0.02 | 255/ 262 (97%) | 235/ 240 (98%) | 0.04 | 0.23 |
| Any animal observed | 384/ 507 (76%) | 359/ 494 (73%) | 0.07 | 162/ 245 (66%) | 182/ 254 (72%) | 0.12 | 222/ 262 (85%) | 177/ 240 (74%) | 0.27 | 0.05 |
| Dog observed | 70/507 (14%) | 78/494 (16%) | 0.06 | 30/245 (12%) | 40/254 (16%) | 0.10 | 40/262 (15%) | 38/240 (16%) | 0.02 | 0.00 |
| Chicken or duck observed | 63/507 (12%) | 52/494 (11%) | 0.06 | 22/245 (9%) | 32/254 (13%) | 0.12 | 41/262 (16%) | 20/240 (8.3%) | 0.23 | 0.14 |
| Cat observed | 360/ 507 (71%) | 340/ 494 (69%) | 0.05 | 154/ 245 (63%) | 174/ 254 (69%) | 0.12 | 206/ 262 (79%) | 166/ 240 (69%) | 0.22 | 0.01 |
| Compound floods during rain | 182/ 507 (36%) | 184/ 494 (37%) | 0.03 | 89/245 (36%) | 107/ 254 (42%) | 0.12 | 93/262 (36%) | 77/240 (32%) | 0.07 | 0.21 |
| Child age at survey, days | 1518 (407) | 740 (518) | 1.67 | 1520 (406) | 749 (541) | 1.61 | 1516 (408) | 731 (494) | 1.73 | 0.04 |
| Child age at sampling, days | 1510 (415) | 694 (478) | 1.82 | 1505 (408) | 716 (512) | 1.70 | 1516 (422) | 672 (439) | 1.96 | 0.09 |
| Number of household members | 6.6 (3.1) | 6.3 (3.4) | 0.10 | 6.5 (3) | 6.6 (3.8) | 0.04 | 6.7 (3.1) | 6 (2.8) | 0.26 | 0.20 |
| Household wealth score, 0–1 | 0.41 (0.11) | 0.41 (0.11) | 0.01 | 0.41 (0.12) | 0.4 (0.11) | 0.11 | 0.41 (0.1) | 0.42 (0.097) | 0.15 | 0.19 |
| Number of households in compound | 5.3 (4.9) | 5.5 (5.5) | 0.04 | 4.3 (2.8) | 4.4 (3.2) | 0.03 | 6.2 (6.1) | 6.6 (6.9) | 0.06 | 0.41 |
| Compound population | 21 (15) | 21 (16) | 0.04 | 18 (9.5) | 17 (8.9) | 0.07 | 25 (19) | 25 (21) | 0.00 | 0.47 |
| Compound population density | 0.08 (0.047) | 0.08 (0.047) | 0.01 | 0.074 (0.037) | 0.075 (0.042) | 0.03 | 0.087 (0.053) | 0.085 (0.052) | 0.03 | 0.22 |

Results are presented as prevalence (n/N (%)) or mean (standard deviation) at 24-month visit.

* Prevalence (or mean (SD)) for children with repeated observations at baseline and 24-month visits.

[†]Prevalence (or mean (SD)) for children with observations at the 24-month visit only.

[‡]Standardized mean difference between observations of children with and without repeated measures at baseline and 24-month visits.

[§]Standardized mean difference between observations from control and intervention children measured at 24-month visit only.

**Appendix 1—table 7.** Sensitivity analysis assessing the impact of reported deworming on STH effect estimates 12 and 24 months after the intervention.

| | 12-month Prevalence ratio | | | 24-month Prevalence ratio | | |
|---|---|---|---|---|---|---|
| | Main analysis, all children* | Adjusted for reported deworming † | Restricted to children dewormed at baseline ‡ | Main analysis, all children* | Adjusted for reported deworming † | Adjusted for time since deworming§ |
| | n = 1239 | n = 1239 | n = 1031 | n = 1161 | n = 1161 | N = 1159 |
| Any STH infection | 1.11 (0.89–1.38) | 1.09 (0.87–1.35) | 1.06 (0.84–1.33) | 0.95 (0.77–1.17) | 0.93 (0.77–1.16) | 0.93 (0.75–1.14) |
| Trichuris | 1.01 (0.79–1.28) | 0.98 (0.77–1.24) | 0.96 (0.74–1.23) | 0.86 (0.67–1.10) | 0.85 (0.66–1.08) | 0.86 (0.67–1.09) |
| Ascaris | 1.33 (0.92–1.93) | 1.30 (0.90–1.88) | 1.30 (0.87–1.94) | 0.83 (0.54–1.27) | 0.84 (0.55–1.29) | 0.78 (0.51–1.18) |
| Coinfection,≥2 STH | 1.17 (0.76–1.79) | 1.12 (0.73–1.71) | 1.16 (0.73–1.85) | 0.63 (0.37–1.07) | 0.63 (0.37–1.08) | 0.60 (0.35–1.03) |

All effect estimates are presented as prevalence ratios (ratio of ratios) with 95% confidence intervals and estimated using generalized estimating equations to fit Poisson regression models with robust standard errors. All models adjusted for child age, sex, caregiver education level, and household wealth.

*Analysis includes all children regardless of caregiver-reported deworming status.

†Analysis is adjusted for reported deworming status. Effect estimates at 12 month are adjusted for baseline deworming confirmation, effect estimates at 24 month are adjusted for baseline and/or 12 month deworming confirmation.

‡Analysis is restricted to children whose caregivers confirmed baseline deworming.

§ Adjusted for time between 12 month deworming and 24 month sample collection, time broken into three intervals: 0–3 months, 4–6 months, and >6 months. The NDC performed 12 month deworming activities at the end of the 12 month phase instead of concurrent to 12 month sample collection resulting in some variation in the amount of time between 12 month deworming and 24 month sample collection among participants. All samples collected during 12 month phase were collected >6 months after deworming and no adjustment for time since deworming was made.

**Appendix 1—table 8.** Sensitivity analysis assessing impact of independent upgrading of control sanitation facilities on effect estimates.

| | 12-month adjusted prevalence ratio | | 24 month adjusted prevalence ratio | |
|---|---|---|---|---|
| | Main analysis, all children* | Excluding controls with upgraded sanitation† | Main analysis, all children* | Excluding controls with upgraded sanitation† |
| Any bacterial or protozoan infection | 1.04 (0.94–1.15), n = 1510 | 1.05 (0.95–1.16), n = 1491 | 0.99 (0.91–1.09), n = 1536 | 1.00 (0.91–1.10), n = 1502 |
| Any STH infection | 1.11 (0.89–1.38), n = 1239 | 1.11 (0.89–1.38), n = 1225 | 0.95 (0.77–1.17), n = 1161 | 0.94 (0.76–1.16), n = 1148 |
| Diarrhea | 1.69 (0.89–3.21), n = 1594 | 1.76 (0.91–3.39), n = 1575 | 0.84 (0.47–1.51), n = 1502 | 0.81 (0.45–1.48), n = 1471 |

All effect estimates are presented as prevalence ratios (ratio of ratios) with 95% confidence intervals and estimated using generalized estimating equations to fit Poisson regression models with robust standard errors. All infection outcomes are adjusted for child age and sex, caregiver's education, and household wealth index, and the diarrhea outcome is also adjusted for baseline presence of a drop-hole cover and reported use of a tap on compound grounds as primary drinking water source.

* Results represent effect estimates for the main analyses which included control children irrespective of whether their latrines had been independently upgraded (results also presented in *Table 2* in main text).

† Results from sensitivity analyses which exclude control children living in compounds that independently upgraded their latrines to be similar to the intervention.

**Appendix 1—table 9.** Confounding assessment for primary outcome and both secondary outcomes (any STH, diarrhea) at 12 months.

| Variable | n/N (%) or mean (SD) at Baseline Control | Inter-vention. | Std diff.* | Primary outcome Unadjusted† Comparator PR: 1.04 (0.94–1.15) | Primary outcome Adjusted‡ Comparator aPR: 1.04 (0.94–1.15) | Any STH Unadjusted† Comparator PR: 1.12 (0.89–1.40) | Any STH Adjusted‡ Comparator aPR: 1.11 (0.90–1.38) | Diarrhea Unadjusted† Comparator PR: 1.41 (0.80–2.48) | Diarrhea Adjusted‡ Comparator aPR: 1.32 (0.75–2.33) |
|---|---|---|---|---|---|---|---|---|---|
| Female | 266/520 (51%) | 227/444 (51%) | 0.00 | 1.04 (0.94–1.15) | 1.04 (0.94–1.15) | 1.14 (0.91–1.42) | 1.11 (0.89–1.38) | 1.39 (0.79–2.46) | 1.32 (0.75–2.33) |
| Any breastfeeding | 169/526 (32%) | 143/448 (32%) | 0.00 | 1.05 (0.95–1.15) | 1.05 (0.95–1.15) | 1.11 (0.90–1.38) | 1.11 (0.90–1.38) | 1.39 (0.79–2.45) | 1.33 (0.75–2.35) |
| Caregiver completed primary school | 287/528 (54%) | 239/451 (53%) | 0.03 | 1.04 (0.94–1.15) | 1.04 (0.94–1.15) | 1.12 (0.90–1.41) | 1.11 (0.89–1.38) | 1.40 (0.80–2.48) | 1.32 (0.75–2.33) |
| Respondent is mother | 368/519 (71%) | 284/443 (64%) | 0.15 | 1.05 (0.95–1.16) | 1.04 (0.94–1.15) | 1.13 (0.90–1.42) | 1.11 (0.89–1.38) | 1.37 (0.78–2.42) | 1.29 (0.73–2.28) |
| Household floors covered | 511/530 (96%) | 413/453 (91%) | 0.22 | 1.04 (0.94–1.15) | 1.04 (0.94–1.15) | 1.12 (0.89–1.40) | 1.12 (0.90–1.39) | 1.39 (0.79–2.47) | 1.32 (0.74–2.34) |
| Household walls made of sturdy material | 370/530 (70%) | 272/453 (60%) | 0.21 | 1.04 (0.94–1.15) | 1.04 (0.94–1.15) | 1.12 (0.89–1.40) | 1.11 (0.89–1.38) | 1.41 (0.80–2.48) | 1.32 (0.75–2.33) |
| Drinking water source in compound | 386/522 (74%) | 367/448 (82%) | 0.19 | 1.03 (0.93–1.15) | 1.03 (0.93–1.14) | 1.08 (0.85–1.36) | 1.05 (0.83–1.33) | 1.65 (0.89–3.06) | 1.59 (0.85–2.95) |
| Faeces visible around compound grounds | 282/521 (54%) | 171/447 (38%) | 0.32 | 1.03 (0.93–1.13) | 1.03 (0.93–1.13) | 1.14 (0.91–1.43) | 1.12 (0.90–1.40) | 1.43 (0.81–2.54) | 1.35 (0.76–2.40) |
| Compound floods when it rains | 348/533 (65%) | 255/454 (56%) | 0.19 | 1.04 (0.94–1.15) | 1.04 (0.94–1.15) | 1.12 (0.89–1.40) | 1.11 (0.89–1.38) | 1.41 (0.80–2.49) | 1.32 (0.74–2.33) |
| Latrine drop-hole has cover | 278/521 (53%) | 274/447 (61%) | 0.16 | 1.04 (0.94–1.15) | 1.03 (0.93–1.15) | 1.11 (0.88–1.40) | 1.08 (0.85–1.36) | 1.74 (0.92–3.30) | 1.69 (0.89–3.20) |
| Latrine has ceramic/concrete slab or pedestal | 181/518 (35%) | 176/447 (39%) | 0.09 | 1.04 (0.94–1.15) | 1.04 (0.93–1.15) | 1.10 (0.87–1.39) | 1.07 (0.85–1.35) | 1.71 (0.90–3.24) | 1.65 (0.87–3.14) |
| Latrine walls made of sturdy material | 142/521 (27%) | 161/447 (36%) | 0.19 | 1.03 (0.93–1.14) | 1.03 (0.93–1.13) | 1.14 (0.91–1.43) | 1.12 (0.90–1.40) | 1.42 (0.80–2.51) | 1.33 (0.75–2.37) |
| Standing water observed around compound | 14/521 (2.7%) | 56/447 (13%) | 0.38 | 1.03 (0.93–1.14) | 1.03 (0.93–1.13) | 1.14 (0.91–1.42) | 1.12 (0.90–1.39) | 1.42 (0.80–2.51) | 1.34 (0.75–2.38) |
| Leaking or standing wastewater observed around grounds | 363/521 (70%) | 241/447 (54%) | 0.33 | 1.03 (0.93–1.14) | 1.03 (0.93–1.13) | 1.14 (0.91–1.43) | 1.12 (0.90–1.40) | 1.42 (0.80–2.51) | 1.34 (0.75–2.38) |
| Compound has electricity that normally functions | 467/533 (88%) | 420/454 (93%) | 0.16 | 1.04 (0.94–1.15) | 1.04 (0.94–1.15) | 1.11 (0.89–1.39) | 1.11 (0.89–1.38) | 1.41 (0.80–2.48) | 1.32 (0.75–2.34) |
| Any animal observed in compound | 318/533 (60%) | 303/454 (67%) | 0.15 | 1.04 (0.95–1.15) | 1.04 (0.95–1.15) | 1.13 (0.91–1.41) | 1.13 (0.91–1.40) | 1.39 (0.79–2.44) | 1.29 (0.73–2.28) |
| Dog observed | 28/533 (5.3%) | 46/454 (10%) | 0.18 | 1.05 (0.95–1.15) | 1.04 (0.95–1.15) | 1.13 (0.90–1.41) | 1.12 (0.90–1.39) | 1.38 (0.79–2.40) | 1.30 (0.75–2.27) |
| Chicken or duck observed | 70/533 (13%) | 60/454 (13%) | 0.00 | 1.05 (0.95–1.15) | 1.05 (0.95–1.16) | 1.12 (0.90–1.41) | 1.12 (0.90–1.40) | 1.37 (0.78–2.40) | 1.27 (0.72–2.23) |
| Cat observed | 287/533 (54%) | 259/454 (57%) | 0.06 | 1.05 (0.95–1.16) | 1.04 (0.95–1.15) | 1.14 (0.91–1.42) | 1.13 (0.91–1.41) | 1.39 (0.79–2.45) | 1.30 (0.74–2.29) |
| Compound density, terciles | | | 0.40 | .. | .. | .. | .. | .. | .. |
| 0 (least dense) | 199/519 (38%) | 120/447 (27%) | .. | 1.05 (0.95–1.16) | 1.05 (0.95–1.16) | 1.10 (0.88–1.38) | 1.10 (0.89–1.38) | 1.43 (0.81–2.50) | 1.32 (0.75–2.33) |

*Appendix 1—table 9 continued on next page*

*Appendix 1—table 9 continued*

| Variable | n/N (%) or mean (SD) at Baseline Control | Intervention | Std diff.* | Primary outcome Unadjusted† Comparator PR: 1.04 (0.94–1.15) | Primary outcome Adjusted‡ Comparator aPR: 1.04 (0.94–1.15) | Any STH Unadjusted† Comparator PR: 1.12 (0.89–1.40) | Any STH Adjusted‡ Comparator aPR: 1.11 (0.90–1.38) | Diarrhea Unadjusted† Comparator PR: 1.41 (0.80–2.48) | Diarrhea Adjusted‡ Comparator aPR: 1.32 (0.75–2.33) |
|---|---|---|---|---|---|---|---|---|---|
| 1 | 191/519 (37%) | 137/447 (31%) | .. | .. | .. | .. | .. | .. | .. |
| 2 (most dense) | 129/519 (25%) | 190/447 (43%) | .. | .. | .. | .. | .. | .. | .. |
| Child age at survey, days | 700 (405) | 694 (403) | 0.02 | .. | .. | .. | .. | 1.33 (0.76–2.34) | 1.32 (0.75–2.33) |
| Child age at sample, days | 659 (396) | 669 (391) | 0.03 | 1.04 (0.94–1.14) | 1.04 (0.94–1.15) | 1.09 (0.88–1.36) | 1.11 (0.89–1.38) | - | - |
| Cumulative monthly rainfall at survey, mm | 22 (23) | 23 (24) | 0.07 | .. | .. | .. | .. | 1.39 (0.79–2.44) | 1.30 (0.74–2.29) |
| Cumulative monthly rainfall at sample, mm | 25 (30) | 32 (38) | 0.19 | 1.04 (0.94–1.15) | 1.04 (0.95–1.15) | 1.13 (0.90–1.41) | 1.13 (0.91–1.40) | .. | .. |
| Survey collected during rainy season | 155/526 (29%) | 222/448 (50%) | 0.42 | .. | .. | .. | .. | 1.44 (0.81–2.54) | 1.34 (0.76–2.38) |
| Sample collected during rainy season | 136/409 (33%) | 183/370 (49%) | 0.33 | 1.05 (0.95–1.16) | 1.05 (0.95–1.16) | 1.12 (0.90–1.40) | 1.12 (0.90–1.39) | .. | .. |
| Wealth score | 0.44 (0.1) | 0.43 (0.1) | 0.16 | 1.04 (0.94–1.15) | 1.04 (0.94–1.15) | 1.12 (0.90–1.40) | 1.11 (0.89–1.38) | 1.39 (0.79–2.46) | 1.32 (0.75–2.33) |
| Number of household residents | 5.7 (2.7) | 6.6 (3.4) | 0.32 | 1.04 (0.94–1.15) | 1.04 (0.94–1.15) | 1.13 (0.90–1.41) | 1.12 (0.90–1.39) | 1.38 (0.78–2.44) | 1.31 (0.74–2.31) |
| Number of Compound residents | 16 (7.3) | 25 (19) | 0.64 | 1.04 (0.94–1.15) | 1.04 (0.94–1.15) | 1.10 (0.88–1.37) | 1.09 (0.88–1.35) | 1.39 (0.79–2.45) | 1.31 (0.74–2.32) |
| Number of households in compound | 4.1 (2.5) | 6.1 (5.9) | 0.42 | 1.04 (0.94–1.15) | 1.04 (0.94–1.15) | 1.11 (0.89–1.37) | 1.09 (0.88–1.36) | 1.40 (0.79–2.46) | 1.31 (0.74–2.32) |
| Number of compound latrines | 1.0 (0.22) | 1.2 (0.9) | 0.33 | 1.04 (0.94–1.15) | 1.04 (0.94–1.15) | 1.13 (0.91–1.40) | 1.12 (0.90–1.39) | 1.40 (0.79–2.47) | 1.33 (0.75–2.35) |
| Number of compound waterpoints | 0.99 (0.98) | 1.9 (2.4) | 0.47 | 1.03 (0.93–1.14) | 1.03 (0.93–1.14) | 1.13 (0.91–1.42) | 1.12 (0.90–1.39) | 1.45 (0.82–2.56) | 1.37 (0.77–2.43) |

*Standardized difference between arms in baseline covariates. † Compared with 12-month unadjusted prevalence ratio (12 month difference-in-difference estimator). ‡ Compared with 12-month prevalence ratio adjusted for a priori covariates child age, sex, caregiver education, and poverty (wealth score).

**Appendix 1—table 10.** Effect estimates (prevalence ratios) for main analyses and all sub-group analyses adjusted for a priori covariates and age-squared.

| | Main analysis, all children† | | Sub-group analysis, children born after intervention* | | Sub-group analysis, children with repeated (longitudinal) measurements† | | Age stratified, children aged > 24 months old‡ |
|---|---|---|---|---|---|---|---|
| | 12 month | 24 month | 12 month | 24 month | 12 month | 24 month | 24 month |
| Any bacterial or protozoan infection | 1.05 (0.96–1.15), p=0.29 | 1.00 (0.92–1.09), p=0.97 | 0.95 (0.64–1.42), p=0.81 | 0.97 (0.79–1.18), p=0.73 | 1.02 (0.91–1.14), p=0.73 | 0.99 (0.89–1.11), p=0.89 | 0.98 (0.91–1.05), p=0.57 |
| Any STH infection | 1.16 (0.93–1.43), p=0.18 | 0.94 (0.77–1.15), p=0.54 | 1.38 (0.35–5.44), p=0.65 | 0.48 (0.26–0.92), p=0.026 | 1.20 (0.91–1.59), p=0.20 | 1.22 (0.85–1.75), p=0.27 | 1.04 (0.83–1.32), p=0.72 |
| Diarrhea | 1.73 (0.91–3.28), p=0.094 | 0.84 (0.46–1.51), p=0.55 | 1.66 (0.32–8.68), p=0.55 | 1.32 (0.45–3.90), p=0.61 | 1.71 (0.79–3.71), p=0.17 | 0.68 (0.31–1.48), p=0.33 | 0.82 (0.36–1.87), p=0.64 |
| Any Bacteria | 1.10 (0.96–1.26), p=0.15 | 1.01 (0.88–1.16), p=0.87 | 1.23 (0.75–2.02), p=0.42 | 0.88 (0.66–1.16), p=0.37 | 1.02 (0.86–1.20), p=0.85 | 1.02 (0.85–1.22), p=0.85 | 0.96 (0.84–1.11), p=0.61 |
| *Shigella* | 1.14 (0.94–1.38), p=0.18 | 0.97 (0.81–1.16), p=0.75 | 0.87 (0.25–3.02), p=0.83 | 0.48 (0.28–0.84), p=0.009 | 1.09 (0.87–1.35), p=0.47 | 0.96 (0.75–1.23), p=0.76 | 1.02 (0.85–1.23), p=0.82 |
| ETEC | 0.97 (0.70–1.35), p=0.86 | 0.83 (0.57–1.20), p=0.32 | 0.80 (0.33–1.95), p=0.63 | 0.84 (0.47–1.49), p=0.55 | 0.86 (0.58–1.29), p=0.47 | 0.86 (0.52–1.40), p=0.53 | 0.75 (0.47–1.20), p=0.23 |
| *Campylobacter* | 1.70 (0.83–3.49), p=0.15 | 1.29 (0.63–2.64), p=0.49 | 2.67 (0.59–12.00), p=0.2 | 1.63 (0.59–4.54), p=0.35 | 1.51 (0.60–3.76), p=0.38 | 1.52 (0.60–3.83), p=0.38 | 0.98 (0.30–3.21), p=0.97 |
| *C. difficile* | 2.06 (0.76–5.53), p=0.15 | 1.38 (0.45–4.20), p=0.57 | 1.42 (0.43–4.65), p=0.57 | 1.45 (0.40–5.25), p=0.57 | 1.35 (0.23–7.78), p=0.74 | 0.23 (0.02–2.67), p=0.24 | ..‡ |
| *E. coli* O157 | 0.47 (0.18–1.23), p=0.13 | 0.52 (0.17–1.59), p=0.25 | 0.00 (0.00–0.01), p=0.00 | 0.52 (0.07–4.14), p=0.54 | 0.68 (0.22–2.07), p=0.50 | 0.58 (0.12–2.86), p=0.51 | 0.48 (0.13–1.78), p=0.27 |
| STEC | 0.15 (0.03–0.71), p=0.017 | 0.24 (0.06–1.03), p=0.055 | ..‡ | 0.05 (0.00–1.26), p=0.069 | 0.11 (0.01–1.32), p=0.082 | 0.58 (0.07–5.00), p=0.62 | 1.70 (0.14–20.35), p=0.67 |
| *Y. enterocolitica* | ..‡ | ..‡ | ..‡ | ..‡ | ..‡ | ..‡ | ..‡ |
| *V. cholerae* | ..‡ | ..‡ | ..‡ | ..‡ | ..‡ | ..‡ | ..‡ |
| Any Protozoa | 1.05 (0.89–1.23), p=0.6 | 0.92 (0.78–1.09), p=0.34 | 0.42 (0.14–1.26), p=0.12 | 0.86 (0.60–1.23), p=0.41 | 1.20 (0.97–1.48), p=0.095 | 0.92 (0.73–1.16), p=0.49 | 0.94 (0.80–1.10), p=0.45 |
| *Giardia* | 1.07 (0.91–1.26), p=0.43 | 0.95 (0.80–1.12), p=0.51 | 0.46 (0.15–1.47), p=0.19 | 0.89 (0.62–1.28), p=0.52 | 1.19 (0.96–1.47), p=0.11 | 0.92 (0.73–1.16), p=0.47 | 0.96 (0.81–1.13), p=0.6 |
| *Cryptosporidium* | 0.89 (0.24–3.33), p=0.86 | 0.53 (0.13–2.17), p=0.38 | 0.33 (0.02–6.28), p=0.46 | 0.51 (0.09–2.78), p=0.44 | 1.46 (0.21–10.18), p=0.7 | 0.59 (0.06–5.45), p=0.64 | 0.20 (0.02–2.28), p=0.19 |
| *E. histolytica* | ..‡ | ..‡ | ..‡ | ..‡ | ..‡ | ..‡ | ..‡ |
| Any virus | 0.75 (0.44–1.28), p=0.29 | 1.03 (0.57–1.86), p=0.92 | 0.37 (0.14–1.03), p=0.056 | 0.79 (0.35–1.78), p=0.57 | 1.09 (0.52–2.29), p=0.83 | 0.95 (0.41–2.19), p=0.91 | 1.44 (0.61–3.38), p=0.41 |
| Norovirus GI/GII | 0.68 (0.36–1.28), p=0.23 | 1.10 (0.55–2.18), p=0.79 | 0.42 (0.12–1.41), p=0.16 | 1.25 (0.47–3.29), p=0.66 | 0.86 (0.37–2.00), p=0.73 | 0.74 (0.29–1.90), p=0.53 | 1.16 (0.45–3.04), p=0.76 |

*Continued on next page*

*Appendix 1—table 10 continued*

| | Main analysis, all children† | | Sub-group analysis, children born after intervention* | | Sub-group analysis, children with repeated (longitudinal) measurements† | | Age stratified, children aged > 24 months old‡ |
|---|---|---|---|---|---|---|---|
| | 12 month | 24 month | 12 month | 24 month | 12 month | 24 month | 24 month |
| Adenovirus 40/41 | 1.26 (0.32–4.95), p=0.74 | 0.96 (0.18–5.20), p=0.96 | 0.85 (0.09–8.30), p=0.89 | ..‡ | 3.77 (0.48–29.56), p=0.21 | 6.17 (0.51–75.19), p=0.15 | 7.51 (0.72–77.98), p=0.091 |
| Rotavirus A | ..‡ | ..‡ | ..‡ | ..‡ | ..‡ | ..‡ | ..‡ |
| Coinfection,≥2 GPP pathogens | 1.10 (0.93–1.30), p=0.27 | 0.94 (0.80–1.11), p=0.49 | 0.75 (0.33–1.71), p=0.49 | 0.83 (0.58–1.17), p=0.29 | 1.15 (0.93–1.42), p=0.19 | 0.97 (0.78–1.21), p=0.81 | 0.93 (0.78–1.11), p=0.44 |
| *Trichuris* | 1.05 (0.83–1.32), p=0.68 | 0.85 (0.67–1.08), p=0.17 | 0.99 (0.23–4.27), p=0.98 | 0.24 (0.10–0.60), p=0.002 | 1.11 (0.80–1.52), p=0.54 | 1.14 (0.76–1.70), p=0.54 | 0.99 (0.77–1.27), p=0.92 |
| *Ascaris* | 1.38 (0.95–1.99), p=0.088 | 0.83 (0.54–1.26), p=0.37 | 3.11 (0.30–32.54), p=0.34 | 0.65 (0.29–1.47), p=0.3 | 1.20 (0.76–1.92), p=0.43 | 0.86 (0.42–1.75), p=0.68 | 0.86 (0.51–1.44), p=0.56 |
| Coinfection,≥2 STH | 1.21 (0.78–1.85), p=0.39 | 0.62 (0.37–1.06), p=0.079 | 1.76 (0.15–21), p=0.66 | 0.12 (0.01–1.06), p=0.057 | 1.01 (0.53–1.93), p=0.97 | 0.70 (0.30–1.62), p=0.40 | 0.72 (0.40–1.29), p=0.27 |

All effect estimates are presented as prevalence ratios (ratio of ratios) with 95% confidence intervals and estimated using generalized estimating equations to fit Poisson regression models with robust standard errors. All models are adjusted for a priori covariates (age, sex, wealth, caregiver education) and age squared to assess the impact of the age squared term on effect estimates. †Results from main analyses examining intervention effects among all enrolled children at 12 month and 24 month visits. Effect estimates compared with 12 month and 24 month results in *Table 2*.

*Results from sub-group analyses which compared children born after the intervention was implemented with children of a similar age at baseline. Effect estimates compared with results in *Table 3* (24 month sub-group analysis results) and *Appendix 1—table 13* (12 month sub-group analysis results).

† Results from sub-group analyses including children with repeated measures at baseline and the 12 month phase or baseline and the 24 month phase. Effect estimates compared with results in *Appendix 1—tables 14* and *15*.

‡ Results from sub-group analysis comparing children aged >2 years old at baseline and 24 month phase. Effect estimates compared with results in *Appendix 1—table 12*.

**Appendix 1—table 11.** Comparison of effect estimates (prevalence ratios) at 12- and 24-month adjusted for a priori covariates only and for a priori covariates and seasonality.

| | 12-month prevalence ratio (95% CI) | | 24-month prevalence ratio (95% CI) | |
|---|---|---|---|---|
| | Adjusted (a priori only)* | Adjusted + seasonality† | Adjusted (a priori only)* | Adjusted + seasonality† |
| Any bacterial or protozoan infection | 1.04 (0.94–1.15), p=0.41 | 1.05 (0.95–1.15), p=0.37 | 0.99 (0.91–1.09), p=0.89 | 1.00 (0.91–1.10), p=0.95 |
| Any STH infection | 1.11 (0.89–1.38), p=0.35 | 1.12 (0.90–1.39), p=0.31 | 0.95 (0.77–1.17), p=0.62 | 0.94 (0.76–1.15), p=0.54 |
| Diarrhea | 1.69 (0.89–3.21), p=0.11 | 1.67 (0.88–3.17), p=0.12 | 0.84 (0.47–1.51), p=0.56 | 0.81 (0.44–1.46), p=0.48 |
| Any bacteria | 1.09 (0.95–1.26), p=0.20 | 1.10 (0.96–1.26), p=0.18 | 1.00 (0.87–1.15), p=0.95 | 1.03 (0.89–1.18), p=0.71 |
| *Shigella* | 1.12 (0.92–1.38), p=0.27 | 1.12 (0.91–1.37), p=0.28 | 0.95 (0.79–1.16), p=0.64 | 0.97 (0.80–1.17), p=0.72 |

*Continued on next page*

*Appendix 1—table 11 continued*

| | 12-month prevalence ratio (95% CI) | | 24-month prevalence ratio (95% CI) | |
| --- | --- | --- | --- | --- |
| | Adjusted (a priori only)* | Adjusted + seasonality† | Adjusted (a priori only)* | Adjusted + seasonality† |
| ETEC | 0.96 (0.69–1.33), p=0.81 | 0.98 (0.70–1.35), p=0.89 | 0.83 (0.57–1.19), p=0.31 | 0.88 (0.61–1.26), p=0.47 |
| *Campylobacter* | 1.68 (0.82–3.45), p=0.16 | 1.72 (0.84–3.49), p=0.14 | 1.28 (0.62–2.62), p=0.5 | 1.33 (0.65–2.71), p=0.43 |
| *C. difficile* | 2.09 (0.77–5.64), p=0.15 | 2.17 (0.81–5.86), p=0.13 | 1.41 (0.46–4.30), p=0.54 | 1.44 (0.48–4.37), p=0.52 |
| *E. coli* O157 | 0.46 (0.18–1.21), p=0.12 | 0.48 (0.18–1.26), p=0.14 | 0.52 (0.17–1.59), p=0.25 | 0.57 (0.19–1.74), p=0.32 |
| STEC | 0.15 (0.03–0.70), p=0.016 | 0.15 (0.03–0.74), p=0.019 | 0.24 (0.05–1.01), p=0.052 | 0.25 (0.06–1.06), p=0.061 |
| *Y. enterocolitica* | ..‡ | ..‡ | ..‡ | ..‡ |
| *V. cholerae* | ..‡ | ..‡ | ..‡ | ..‡ |
| Any Protozoa | 1.03 (0.86–1.22), p=0.76 | 1.03 (0.87–1.23), p=0.72 | 0.91 (0.76–1.09), p=0.29 | 0.91 (0.76–1.09), p=0.31 |
| *Giardia* | 1.05 (0.88–1.25), p=0.58 | 1.06 (0.88–1.26), p=0.54 | 0.93 (0.78–1.11), p=0.43 | 0.93 (0.78–1.12), p=0.45 |
| *Cryptosporidium* | 0.89 (0.24–3.31), p=0.86 | 0.83 (0.22–3.11), p=0.78 | 0.53 (0.13–2.14), p=0.37 | 0.46 (0.12–1.73), p=0.25 |
| *E. histolytica* | ..‡ | ..‡ | ..‡ | ..‡ |
| Any virus | 0.75 (0.44–1.27), p=0.29 | 0.74 (0.43–1.26), p=0.26 | 1.03 (0.57–1.86), p=0.92 | 0.97 (0.54–1.75), p=0.91 |
| Norovirus GI/GII | 0.68 (0.36–1.27), p=0.23 | 0.67 (0.35–1.27), p=0.22 | 1.10 (0.55–2.18), p=0.79 | 1.04 (0.53–2.07), p=0.90 |
| Adenovirus 40/41 | 1.24 (0.32–4.83), p=0.76 | 1.29 (0.33–5.13), p=0.71 | 0.97 (0.18–5.19), p=0.97 | 1.01 (0.19–5.30), p=0.99 |
| Rotavirus | ..‡ | ..‡ | ..‡ | ..‡ |
| Coinfection,≥2 GPP pathogens | 1.08 (0.91–1.29), p=0.37 | 1.09 (0.91–1.30), p=0.35 | 0.93 (0.79–1.10), p=0.41 | 0.94 (0.79–1.12), p=0.49 |
| *Trichuris* | 1.01 (0.79–1.28), p=0.96 | 1.02 (0.81–1.30), p=0.86 | 0.86 (0.67–1.10), p=0.22 | 0.85 (0.67–1.09), p=0.21 |
| *Ascaris* | 1.33 (0.92–1.93), p=0.13 | 1.35 (0.93–1.95), p=0.11 | 0.83 (0.54–1.27), p=0.39 | 0.81 (0.53–1.25), p=0.34 |
| Coinfection,≥2 STH | 1.17 (0.76–1.79), p=0.49 | 1.20 (0.78–1.83), p=0.40 | 0.63 (0.37–1.07), p=0.084 | 0.62 (0.36–1.06), p=0.079 |

All effect estimates are presented as prevalence ratios (ratio of ratios) with 95% confidence intervals and estimated using generalized estimating equations to fit Poisson regression models with robust standard errors.

*Models are adjusted for a priori covariates age, sex, caregiver's education, and wealth and presented for comparison with seasonality-adjusted models.

†Models are adjusted for a priori covariates and seasonality using sine/cosine terms based on the date of sample (or survey) collection.

**Appendix 1—table 12.** Effect of the intervention on enteric infection and diarrhea in children > 2 years old after 24 months.

| | Prevalence | | Prevalence ratio (95% CI), p-value | |
| --- | --- | --- | --- | --- |
| | Baseline, aged > 2 years | 24 month, aged > 2 years | Unadjusted | Adjusted* |
| **Any bacterial or protozoan infection†** | | | | |
| Control | 155/164 (95%) | 315/340 (93%) | .. | .. |
| Intervention | 149/160 (93%) | 312/344 (91%) | 0.99 (0.93–1.07), p=0.86 | 0.98 (0.91–1.05), p=0.60 |
| **Any STH infection†** | | | | |
| Control | 103/155 (66%) | 113/175 (65%) | .. | .. |
| Intervention | 86/146 (59%) | 121/208 (58%) | 1.03 (0.82–1.30), p=0.79 | 1.05 (0.83–1.32), p=0.69 |
| **Diarrhea‡** | | | | |
| Control | 21/243 (8.6%) | 33/273 (12%) | .. | .. |
| Intervention | 16/210 (7.6%) | 31/303 (10%) | 0.96 (0.45–2.07), p=0.93 | 0.82 (0.36–1.86), p=0.63 |
| **Any Bacteria** | | | | |
| Control | 129/164 (79%) | 267/340 (79%) | .. | .. |
| Intervention | 125/160 (78%) | 266/344 (77%) | 1.00 (0.87–1.15), p=0.98 | 0.97 (0.84–1.11), p=0.64 |
| *Shigella* | | | | |
| Control | 112/164 (68%) | 227/340 (67%) | .. | .. |
| Intervention | 103/160 (64%) | 223/344 (65%) | 1.05 (0.87–1.26), p=0.63 | 1.03 (0.85–1.24), p=0.79 |
| *ETEC* | | | | |
| Control | 46/164 (28%) | 93/340 (27%) | .. | .. |
| Intervention | 52/160 (33%) | 100/344 (29%) | 0.88 (0.56–1.38), p=0.58 | 0.74 (0.46–1.20), p=0.22 |
| *Campylobacter* | | | | |
| Control | 12/164 (7.3%) | 33/340 (9.7%) | .. | .. |
| Intervention | 7/160 (4.4%) | 20/344 (5.8%) | 0.97 (0.33–2.90), p=0.96 | 1.00 (0.30–3.28), p=0.99 |
| *C. difficile* | | | | |
| Control | 2/164 (1.2%) | 6/340 (1.8%) | .. | .. |
| Intervention | 0/160 (0.0%) | 4/344 (1.2%) | ..‡ | ..‡ |
| *E. coli O157* | | | | |
| Control | 6/164 (3.7%) | 21/340 (6.2%) | .. | .. |
| Intervention | 9/160 (5.6%) | 13/344 (3.8%) | 0.39 (0.11–1.40), p=0.15 | 0.47 (0.13–1.78), p=0.27 |
| **STEC** | | | | |
| Control | 2/164 (1.2%) | 15/340 (4.4%) | .. | .. |
| Intervention | 1/160 (0.63%) | 13/344 (3.8%) | 1.54 (0.12–19.19), p=0.74 | 1.73 (0.14–20.75), p=0.67 |
| *Y. enterocolitica* | | | | |
| Control | 0/164 (0.0%) | 0/340 (0.0%) | .. | .. |
| Intervention | 0/160 (0.0%) | 1/344 (0.29%) | ..‡ | ..‡ |
| *V. cholerae* | | | | |
| Control | 0/164 (0.0%) | 0/340 (0.0%) | .. | .. |

*Continued on next page*

*Appendix 1—table 12 continued*

| | Prevalence | | Prevalence ratio (95% CI), p-value | |
|---|---|---|---|---|
| | Baseline, aged > 2 years | 24 month, aged > 2 years | Unadjusted | Adjusted* |
| Intervention | 0/160 (0.0%) | 0/344 (0.0%) | ..‡ | ..‡ |
| **Any Protozoa** | | | | |
| Control | 123/164 (75%) | 250/340 (74%) | .. | .. |
| Intervention | 121/160 (76%) | 245/344 (71%) | 0.96 (0.82–1.13), p=0.66 | 0.94 (0.80–1.11), p=0.47 |
| *Giardia* | | | | |
| Control | 122/164 (74%) | 244/340 (72%) | .. | .. |
| Intervention | 118/160 (74%) | 240/344 (70%) | 0.99 (0.84–1.16), p=0.86 | 0.96 (0.81–1.13), p=0.62 |
| *Cryptosporidium* | | | | |
| Control | 1/164 (0.61%) | 9/340 (2.6%) | .. | .. |
| Intervention | 4/160 (2.5%) | 8/344 (2.3%) | 0.20 (0.02–2.27), p=0.19 | 0.21 (0.02–2.46), p=0.21 |
| *E. histolytica* | | | | |
| Control | 0/164 (0.0%) | 2/340 (0.59%) | .. | .. |
| Intervention | 3/160 (1.9%) | 10/344 (2.9%) | ..‡ | ..‡ |
| **Any virus** | | | | |
| Control | 19/164 (12%) | 39/340 (11%) | .. | .. |
| Intervention | 16/160 (10%) | 43/344 (13%) | 1.24 (0.55–2.78), p=0.6 | 1.44 (0.61–3.38), p=0.41 |
| **Norovirus GI/GII** | | | | |
| Control | 12/164 (7.3%) | 34/340 (10%) | .. | .. |
| Intervention | 13/160 (8.1%) | 37/344 (11%) | 0.96 (0.39–2.34), p=0.92 | 1.17 (0.45–3.03), p=0.75 |
| **Adenovirus 40/41** | | | | |
| Control | 6/164 (3.7%) | 2/340 (0.59%) | .. | .. |
| Intervention | 2/160 (1.3%) | 6/344 (1.7%) | 11 (0.97–119), p=0.053 | 7.5 (0.72–79), p=0.92 |
| **Rotavirus A** | | | | |
| Control | 1/164 (0.61%) | 3/340 (0.88%) | .. | .. |
| Intervention | 1/160 (0.63%) | 1/344 (0.29%) | ..‡ | ..‡ |
| **Coinfection,$\geq$2 GPP pathogens** | | | | |
| Control | 114/164 (70%) | 243/340 (71%) | .. | .. |
| Intervention | 111/160 (69%) | 236/344 (69%) | 0.97 (0.82–1.15), p=0.71 | 0.93 (0.78–1.12), p=0.45 |
| *Trichuris* | | | | |
| Control | 91/155 (59%) | 102/175 (58%) | .. | .. |
| Intervention | 76/146 (52%) | 110/208 (53%) | 1.04 (0.81–1.33), p=0.78 | 0.99 (0.77–1.27), p=0.96 |
| *Ascaris* | | | | |
| Control | 50/155 (32%) | 61/175 (35%) | .. | .. |
| Intervention | 39/146 (27%) | 47/208 (23%) | 0.78 (0.47–1.29), p=0.33 | 0.86 (0.51–1.44), p=0.57 |
| **Coinfection,$\geq$2 STH** | | | | |
| Control | 38/155 (25%) | 50/175 (29%) | .. | .. |
| Intervention | 29/146 (20%) | 36/208 (17%) | 0.74 (0.42–1.28), p=0.28 | 0.72 (0.41–1.29), p=0.27 |

Analysis includes children >2 year old at baseline or the 24 month visit. Prevalence results are presented as (n/N (%)). All effect estimates are presented as prevalence ratios (ratio of ratios) with 95% confidence intervals and estimated using generalized estimating equations to fit Poisson regression models with robust standard errors.

* Pathogen outcomes adjusted for child age and sex, caregiver's education, and household wealth index, reported diarrhea also adjusted for baseline presence of a drop-hole cover and reported use of a tap on compound grounds as primary drinking water source.

† Models did not converge due to sparse data.

**Appendix 1—table 13.** Effect of intervention on enteric infection and reported diarrhea in children born into study sites post implementation (post-baseline) and before 12 month visit compared with children of a similar age at baseline (<1 year old).

| | Prevalence | | Prevalence ratio | |
|---|---|---|---|---|
| | Baseline, children < 1 year old | 12 month, children born-in and <1 year old | unadjusted | adjusted† |
| **Any bacterial or protozoan infection** | | | | |
| Control | 57/109 (52%) | 31/48 (65%) | .. | .. |
| Intervention | 51/99 (52%) | 32/55 (58%) | 0.89 (0.60–1.33), p=0.58 | 0.97 (0.65–1.45), p=0.90 |
| **Any STH infection** | | | | |
| Control | 17/93 (18%) | 3/25 (12%) | .. | .. |
| Intervention | 13/92 (14%) | 4/32 (13%) | 1.31 (0.32–5.42), p=0.71 | 1.38 (0.35–5.45), p=0.65 |
| **Diarrhea** | | | | |
| Control | 19/138 (14%) | 6/50 (12%) | .. | .. |
| Intervention | 18/120 (15%) | 13/69 (19%) | 1.38 (0.47–4.01), p=0.56 | 1.80 (0.35–9.31), p=0.48 |
| **Any Bacteria** | | | | |
| Control | 53/109 (49%) | 24/48 (50%) | .. | .. |
| Intervention | 41/99 (41%) | 29/55 (53%) | 1.22 (0.75–1.98), p=0.43 | 1.28 (0.78–2.10), p=0.33 |
| *Shigella* | | | | |
| Control | 10/109 (9.2%) | 9/48 (19%) | .. | .. |
| Intervention | 9/99 (9.1%) | 9/55 (16%) | 0.87 (0.26–2.91), p=0.82 | 0.85 (0.26–2.81), p=0.79 |
| *ETEC* | | | | |
| Control | 25/109 (23%) | 12/48 (25%) | .. | .. |
| Intervention | 22/99 (22%) | 11/55 (20%) | 0.82 (0.34–1.99), p=0.66 | 0.80 (0.33–1.92), p=0.62 |
| *Campylobacter* | | | | |
| Control | 14/109 (13%) | 4/48 (8.3%) | .. | .. |
| Intervention | 8/99 (8.1%) | 5/55 (9.1%) | 1.76 (0.38–8.09), p=0.47 | 2.68 (0.59–12.2), p=0.20 |
| *C. difficile* | | | | |
| Control | 13/109 (12%) | 7/48 (15%) | .. | .. |
| Intervention | 10/99 (10%) | 9/55 (16%) | 1.37 (0.42–4.45), p=0.60 | 1.49 (0.46–4.89), p=0.51 |
| *E. coli O157* | | | | |
| Control | 4/109 (3.7%) | 1/48 (2.1%) | .. | .. |

*Continued on next page*

*Appendix 1—table 13 continued*

| | Prevalence | | Prevalence ratio | |
|---|---|---|---|---|
| | Baseline, children < 1 year old | 12 month, children born-in and <1 year old | unadjusted | adjusted† |
| Intervention | 2/99 (2%) | 0/55 (0.0%) | 0.01 (0.00–0.19), p=0.001 | ..‡ |
| **STEC** | | | | |
| Control | 0/109 (0.0%) | 0/48 (0.0%) | .. | .. |
| Intervention | 3/99 (3%) | 1/55 (1.8%) | ..‡ | ..‡ |
| *Y. enterocolitica* | | | | |
| Control | 0/109 (0.0%) | 0/48 (0.0%) | .. | .. |
| Intervention | 0/99 (0.0%) | 0/55 (0.0%) | ..‡ | ..‡ |
| *V. cholerae* | | | | |
| Control | 0/109 (0.0%) | 0/48 (0.0%) | .. | .. |
| Intervention | 0/99 (0.0%) | 0/55 (0.0%) | ..‡ | ..‡ |
| **Any Protozoa** | | | | |
| Control | 14/109 (13%) | 15/48 (31%) | .. | .. |
| Intervention | 22/99 (22%) | 9/55 (16%) | 0.35 (0.12–1.02), p=0.055 | 0.40 (0.13–1.20), p=0.10 |
| *Giardia* | | | | |
| Control | 12/109 (11%) | 13/48 (27%) | .. | .. |
| Intervention | 16/99 (16%) | 8/55 (15%) | 0.41 (0.13–1.24), p=0.11 | 0.44 (0.14–1.40), p=0.17 |
| *Cryptosporidium* | | | | |
| Control | 2/109 (1.8%) | 2/48 (4.2%) | .. | .. |
| Intervention | 8/99 (8.1%) | 2/55 (3.6%) | 0.25 (0.02–3.70), p=0.31 | 0.40 (0.02–7.9), p=0.55 |
| *E. histolytica* | | | | |
| Control | 0/109 (0.0%) | 1/48 (2.1%) | .. | .. |
| Intervention | 1/99 (1%) | 0/55 (0.0%) | ..‡ | ..‡ |
| **Any virus** | | | | |
| Control | 15/109 (14%) | 12/48 (25%) | .. | .. |
| Intervention | 21/99 (21%) | 7/55 (13%) | 0.33 (0.12–0.92), p=0.033 | 0.37 (0.14–1.03), p=0.056 |
| **Norovirus GI/GII** | | | | |
| Control | 12/109 (11%) | 9/48 (19%) | .. | .. |
| Intervention | 15/99 (15%) | 6/55 (11%) | 0.43 (0.13–1.40), p=0.16 | 0.44 (0.13–1.47), p=0.18 |
| **Adenovirus 40/41** | | | | |
| Control | 4/109 (3.7%) | 4/48 (8.3%) | .. | .. |
| Intervention | 3/99 (3%) | 2/55 (3.6%) | 0.56 (0.06–5.05), p=0.61 | 0.91 (0.09–9.49), p=0.94 |
| **Rotavirus A** | | | | |
| Control | 0/109 (0.0%) | 0/48 (0.0%) | .. | .. |
| Intervention | 3/99 (3%) | 0/55 (0.0%) | ..‡ | ..‡ |
| **Coinfection, ≥2 GPP pathogens** | | | | |
| Control | 23/109 (21%) | 16/48 (33%) | .. | .. |

*Continued on next page*

*Appendix 1—table 13 continued*

|  | Prevalence | | Prevalence ratio | |
| --- | --- | --- | --- | --- |
|  | Baseline, children < 1 year old | 12 month, children born-in and <1 year old | unadjusted | adjusted† |
| Intervention | 25/99 (25%) | 15/55 (27%) | 0.73 (0.31–1.71), p=0.47 | 0.74 (0.33–1.69), p=0.48 |
| *Trichuris* | | | | |
| Control | 10/93 (11%) | 3/25 (12%) | .. | .. |
| Intervention | 10/92 (11%) | 4/32 (13%) | 1.04 (0.21–5.01), p=0.96 | 0.98 (0.23–4.29), p=0.98 |
| *Ascaris* | | | | |
| Control | 12/93 (13%) | 1/25 (4%) | .. | .. |
| Intervention | 9/92 (9.8%) | 3/32 (9.4%) | 2.87 (0.30–27.85), p=0.36 | 3.10 (0.30–32.5), p=0.35 |
| **Coinfection,≥2 STH** | | | | |
| Control | 5/93 (5.4%) | 1/25 (4%) | .. | .. |
| Intervention | 6/92 (6.5%) | 3/32 (9.4%) | 1.90 (0.16–22.73), p=0.61 | 1.76 (0.15–21.0), p=0.66 |

Analysis includes children < 1 year old at baseline and children born into the study after baseline and <1 year old at the time of the 12-month visit. Prevalence results are presented as (n/N (%)). All effect estimates are presented as prevalence ratios (ratio of ratios) with 95% confidence intervals and estimated using generalized estimating equations to fit Poisson regression models with robust standard errors.

*Pathogen outcomes adjusted for child age and sex, caregiver's education, and household wealth index, reported diarrhea also adjusted for baseline presence of a drop-hole cover and reported use of a tap on compound grounds as primary drinking water source.

† Models did not converge due to sparse data.

**Appendix 1—table 14.** Effect of the intervention on children with repeated observations at baseline and 12-month visit.

|  | Prevalence | | Prevalence ratio | |
| --- | --- | --- | --- | --- |
|  | Baseline | 12 month | Unadjusted | Adjusted† |
| **Any bacterial or protozoan infection** | | | | |
| Control | 161/207 (78%) | 187/207 (90%) | .. | .. |
| Intervention | 174/228 (76%) | 207/228 (91%) | 1.02 (0.91–1.16), p=0.70 | 1.01 (0.90–1.14), p=0.84 |
| **Any STH infection** | | | | |
| Control | 67/132 (51%) | 80/132 (61%) | .. | .. |
| Intervention | 63/154 (41%) | 91/154 (59%) | 1.22 (0.92–1.61), p=0.17 | 1.16 (0.87–1.55), p=0.31 |
| **Diarrhea** | | | | |
| Control | 36/277 (13%) | 17/277 (6.1%) | .. | .. |
| Intervention | 42/279 (15%) | 34/279 (12%) | 1.71 (0.78–3.77), p=0.18 | 1.71 (0.79–3.70), p=0.17 |
| **Any Bacteria** | | | | |
| Control | 141/207 (68%) | 165/207 (80%) | .. | .. |
| Intervention | 142/228 (62%) | 170/228 (75%) | 1.02 (0.86–1.22), p=0.8 | 1.01 (0.85–1.20), p=0.92 |

*Continued on next page*

*Appendix 1—table 14 continued*

| | Prevalence | | Prevalence ratio | |
| --- | --- | --- | --- | --- |
| | Baseline | 12 month | Unadjusted | Adjusted† |
| ***Shigella*** | | | | |
| Control | 89/207 (43%) | 128/207 (62%) | | |
| Intervention | 90/228 (39%) | 142/228 (62%) | 1.10 (0.86–1.39), p=0.45 | 1.08 (0.85–1.37), p=0.54 |
| ***ETEC*** | | | | |
| Control | 63/207 (30%) | 83/207 (40%) | | |
| Intervention | 71/228 (31%) | 79/228 (35%) | 0.84 (0.56–1.27), p=0.41 | 0.85 (0.57–1.28), p=0.44 |
| ***Campylobacter*** | | | | |
| Control | 20/207 (9.7%) | 18/207 (8.7%) | | |
| Intervention | 13/228 (5.7%) | 18/228 (7.9%) | 1.54 (0.62–3.80), p=0.35 | 1.49 (0.60–3.71), p=0.39 |
| ***C. difficile*** | | | | |
| Control | 15/207 (7.3%) | 4/207 (1.9%) | | |
| Intervention | 8/228 (3.5%) | 3/228 (1.3%) | 1.39 (0.24–8.00), p=0.71 | 1.45 (0.25–8.52), p=0.68 |
| ***E. coli O157*** | | | | |
| Control | 9/207 (4.3%) | 15/207 (7.3%) | .. | .. |
| Intervention | 9/228 (4.0%) | 10/228 (4.4%) | 0.67 (0.22–2.03), p=0.48 | 0.68 (0.22–2.06), p=0.49 |
| ***STEC*** | | | | |
| Control | 1/207 (0.48%) | 6/207 (2.9%) | .. | .. |
| Intervention | 6/228 (2.6%) | 4/227 (1.8%) | 0.11 (0.01–1.31), p=0.081 | 0.11 (0.01–1.32), p=0.082 |
| ***Y. enterocolitica*** | | | | |
| Control | 0/207 (0.0%) | 0/207 (0.0%) | .. | .. |
| Intervention | 1/228 (0.44%) | 0/227 (0.0%) | ..‡ | ..‡ |
| ***V. cholerae*** | | | | |
| Control | 0/207 (0.0%) | 0/207 (0.0%) | .. | .. |
| Intervention | 0/228 (0.0%) | 0/227 (0.0%) | ..‡ | ..‡ |
| ***Any Protozoa*** | | | | |
| Control | 109/207 (53%) | 130/207 (63%) | .. | .. |
| Intervention | 117/228 (51%) | 166/228 (73%) | 1.19 (0.95–1.48), p=0.13 | 1.18 (0.94–1.47), p=0.15 |
| ***Giardia*** | | | | |
| Control | 106/207 (51%) | 130/207 (63%) | | |
| Intervention | 113/228 (50%) | 164/228 (72%) | 1.18 (0.94–1.48), p=0.15 | 1.17 (0.93–1.47), p=0.17 |
| ***Cryptosporidium*** | | | | |
| Control | 6/207 (2.9%) | 2/207 (0.97%) | .. | .. |
| Intervention | 10/228 (4.4%) | 5/227 (2.2%) | 1.44 (0.21–9.82), p=0.71 | 1.45 (0.22–9.71), p=0.7 |

*Continued on next page*

*Appendix 1—table 14 continued*

| | Prevalence | | Prevalence ratio | |
|---|---|---|---|---|
| | Baseline | 12 month | Unadjusted | Adjusted† |
| **_E. histolytica_** | | | | |
| Control | 0/207 (0.0%) | 0/207 (0.0) | .. | .. |
| Intervention | 2/228 (0.88%) | 7/228 (3.1%) | ..‡ | ..‡ |
| **Any virus** | | | | |
| Control | 27/207 (13%) | 20/207 (9.7%) | .. | .. |
| Intervention | 31/228 (14%) | 25/228 (11%) | 1.05 (0.50–2.22), p=0.89 | 1.08 (0.51–2.26), p=0.84 |
| **Norovirus GI/GII** | | | | |
| Control | 20/207 (9.7%) | 19/207 (9.2%) | | |
| Intervention | 23/228 (11%) | 19/228 (8.3%) | 0.83 (0.36–1.94), p=0.67 | 0.86 (0.37–1.99), p=0.72 |
| **Adenovirus 40/41** | | | | |
| Control | 7/207 (3.4%) | 2/207 (0.97%) | .. | .. |
| Intervention | 6/228 (2.6%) | 6/228 (2.6%) | 3.56 (0.46–27.24), p=0.22 | 3.59 (0.46–27.91), p=0.22 |
| **Rotavirus A** | | | | |
| Control | 1/207 (0.48%) | 1/207 (0.48%) | .. | .. |
| Intervention | 4/228 (1.8%) | 1/228 (0.44%) | ..‡ | ..‡ |
| **Coinfection,≥2 GPP pathogens** | | | | |
| Control | 114/207 (55%) | 135/207 (65%) | .. | .. |
| Intervention | 115/228 (50%) | 156/228 (68%) | 1.15 (0.92–1.43), p=0.23 | 1.14 (0.91–1.42), p=0.25 |
| **_Trichuris_** | | | | |
| Control | 49/132 (37%) | 64/132 (48%) | .. | .. |
| Intervention | 53/154 (34%) | 77/154 (50%) | 1.12 (0.81–1.54), p=0.50 | 1.06 (0.76–1.48), p=0.72 |
| **_Ascaris_** | | | | |
| Control | 40/132 (30%) | 46/132 (35%) | | |
| Intervention | 35/154 (23%) | 49/154 (32%) | 1.22 (0.77–1.93), p=0.4 | 1.17 (0.73–1.86), p=0.51 |
| **Coinfection,≥2 STH** | | | | |
| Control | 22/132 (17%) | 30/132 (23%) | .. | .. |
| Intervention | 25/154 (16%) | 35/154 (23%) | 1.03 (0.55–1.93), p=0.94 | 0.97 (0.51–1.85), p=0.93 |

Analysis includes children with complete observations at baseline and 12-month visits. Prevalence results are presented as (n/N (%)). All effect estimates are presented as prevalence ratios (ratio of ratios) with 95% confidence intervals and estimated using generalized estimating equations to fit Poisson regression models with robust standard errors.

* Pathogen outcomes adjusted for child age and sex, caregiver's education, and household wealth index, reported diarrhea also adjusted for baseline presence of a drop-hole cover and reported use of a tap on compound grounds as primary drinking water source.

† Models would not converge due to sparse data.

**Appendix 1—table 15.** Effect of the intervention on children with repeated observations at baseline and 24-month visit.

| | Prevalence | | Prevalence ratio | |
| --- | --- | --- | --- | --- |
| | Baseline | 24 month | Unadjusted | Adjusted† |
| **Any bacterial or protozoan infection** | | | | |
| Control | 131/166 (79%) | 155/166 (93%) | .. | .. |
| Intervention | 151/192 (79%) | 175/192 (91%) | 0.98 (0.87–1.10), p=0.73 | 0.98 (0.87–1.10), p=0.70 |
| **Any STH infection** | | | | |
| Control | 48/95 (51%) | 65/95 (68%) | .. | .. |
| Intervention | 38/106 (36%) | 62/106 (58%) | 1.20 (0.84–1.70), p=0.31 | 1.25 (0.87–1.78), p=0.23 |
| **Diarrhea** | | | | |
| Control | 25/196 (13%) | 20/196 (10%) | .. | .. |
| Intervention | 34/221 (15%) | 20/221 (9.1%) | 0.72 (0.33–1.58), p=0.41 | 0.69 (0.31–1.50), p=0.35 |
| **Any Bacteria** | | | | |
| Control | 109/166 (66%) | 138/166 (83%) | .. | .. |
| Intervention | 120/192 (63%) | 153/192 (80%) | 1.00 (0.84–1.21), p=0.96 | 1.01 (0.83–1.21), p=0.96 |
| ***Shigella*** | | | | |
| Control | 66/166 (40%) | 121/166 (73%) | | |
| Intervention | 79/192 (41%) | 136/192 (71%) | 0.93 (0.71–1.22), p=0.60 | 0.93 (0.71–1.22), p=0.60 |
| ***ETEC*** | | | | |
| Control | 47/166 (28%) | 47/166 (28%) | | |
| Intervention | 58/192 (30%) | 52/192 (27%) | 0.90 (0.55–1.46), p=0.66 | 0.85 (0.52–1.39), p=0.52 |
| ***Campylobacter*** | | | | |
| Control | 16/166 (9.6%) | 12/166 (7.2%) | | |
| Intervention | 13/192 (6.8%) | 14/192 (7.3%) | 1.44 (0.56–3.72), p=0.45 | 1.52 (0.60–3.83), p=0.37 |
| ***C. difficile*** | | | | |
| Control | 9/166 (5.4%) | 4/166 (2.4%) | .. | .. |
| Intervention | 8/192 (4.2%) | 1/192 (0.52%) | 0.28 (0.03–2.95), p=0.29 | 0.26 (0.03–2.59), p=0.25 |
| ***E. coli O157*** | | | | |
| Control | 7/166 (4.2%) | 9/166 (5.4%) | .. | .. |
| Intervention | 9/192 (4.7%) | 8/192 (4.2%) | 0.69 (0.14–3.40), p=0.65 | 0.59 (0.12–2.93), p=0.52 |
| **STEC** | | | | |
| Control | 2/166 (1.2%) | 7/166 (4.2%) | .. | .. |
| Intervention | 3/192 (1.6%) | 7/192 (3.6%) | 0.66 (0.07–6.20), p=0.72 | 0.58 (0.07–4.89), p=0.61 |
| ***Y. enterocolitica*** | | | | |
| Control | 0/166 (0.0%) | 0/166 (0.0%) | .. | .. |
| Intervention | 0/192 (0.0%) | 1/192 (0.52%) | ..‡ | ..‡ |

*Continued on next page*

*Appendix 1—table 15 continued*

| | Prevalence | | Prevalence ratio | |
|---|---|---|---|---|
| | Baseline | 24 month | Unadjusted | Adjusted† |
| ***V. cholerae*** | | | | |
| Control | 0/166 (0.0%) | 0/166 (0.0%) | .. | .. |
| Intervention | 0/192 (0.0%) | 0/192 (0.0%) | ..‡ | ..‡ |
| **Any Protozoa** | | | | |
| Control | 89/166 (54%) | 121/166 (73%) | .. | .. |
| Intervention | 109/192 (57%) | 138/192 (72%) | 0.93 (0.73–1.19), p=0.56 | 0.90 (0.69–1.15), p=0.39 |
| ***Giardia*** | | | | |
| Control | 86/166 (52%) | 120/166 (72%) | | |
| Intervention | 104/192 (54%) | 135/192 (70%) | 0.93 (0.73–1.18), p=0.55 | 0.89 (0.69–1.15), p=0.38 |
| ***Cryptosporidium*** | | | | |
| Control | 5/166 (3%) | 3/166 (1.8%) | .. | .. |
| Intervention | 11/192 (5.7%) | 4/192 (2.1%) | 0.57 (0.06–5.38), p=0.62 | 0.55 (0.06–4.93), p=0.59 |
| ***E. histolytica*** | | | | |
| Control | 0/166 (0.0%) | 0/166 (0.0%) | .. | .. |
| Intervention | 2/192 (1%) | 8/192 (4.2%) | ..‡ | ..‡ |
| **Any virus** | | | | |
| Control | 21/166 (13%) | 18/166 (11%) | .. | .. |
| Intervention | 30/192 (16%) | 22/192 (11%) | 0.86 (0.37–1.97), p=0.72 | 0.95 (0.41–2.19), p=0.91 |
| **Norovirus GI/GII** | | | | |
| Control | 15/166 (9%) | 15/166 (9%) | .. | .. |
| Intervention | 26/192 (14%) | 17/192 (8.8%) | 0.65 (0.25–1.69), p=0.38 | 0.74 (0.28–1.90), p=0.53 |
| **Adenovirus 40/41** | | | | |
| Control | 6/166 (3.6%) | 1/166 (0.6%) | | |
| Intervention | 5/192 (2.6%) | 5/192 (2.6%) | 6.12 (0.48–78.34), p=0.16 | 6.01 (0.49–73.94), p=0.16 |
| **Rotavirus A** | | | | |
| Control | 1/166 (0.6%) | 2/166 (1.2%) | .. | .. |
| Intervention | 1/192 (0.52%) | 1/192 (0.52%) | ..‡ | ..‡ |
| **Coinfection,≥2 GPP pathogens** | | | | |
| Control | 89/166 (54%) | 120/166 (72%) | .. | .. |
| Intervention | 102/192 (53%) | 132/192 (69%) | 0.96 (0.77–1.19), p=0.69 | 0.95 (0.76–1.19), p=0.67 |
| ***Trichuris*** | | | | |
| Control | 39/95 (41%) | 62/95 (65%) | .. | .. |
| Intervention | 32/106 (30%) | 57/106 (54%) | 1.11 (0.74–1.67), p=0.60 | 1.16 (0.77–1.75), p=0.47 |
| ***Ascaris*** | | | | |
| Control | 27/95 (28%) | 34/95 (36%) | | |
| Intervention | 19/106 (18%) | 21/106 (20%) | 0.88 (0.43–1.79), p=0.72 | 0.89 (0.44–1.79), p=0.74 |
| **Coinfection,≥2 STH** | | | | |

*Continued on next page*

*Appendix 1—table 15 continued*

| | Prevalence | | Prevalence ratio | |
|---|---|---|---|---|
| | Baseline | 24 month | Unadjusted | Adjusted† |
| Control | 18/95 (19%) | 31/95 (33%) | .. | .. |
| Intervention | 13/106 (12%) | 16/106 (15%) | 0.71 (0.30–1.70), p=0.44 | 0.72 (0.31–1.69), p=0.46 |

Analysis includes children with complete observations at baseline and 24-month visits. Prevalence results are presented as (n/N (%)). All effect estimates are presented as prevalence ratios (ratio of ratios) with 95% confidence intervals and estimated using generalized estimating equations to fit Poisson regression models with robust standard errors.

* Pathogen outcomes adjusted for child age and sex, caregiver's education, and household wealth index, reported diarrhea also adjusted for baseline presence of a drop-hole cover and reported use of a tap on compound grounds as primary drinking water source.

† Models would not converge due to sparse data.

**Appendix 1—table 16.** Outcome and covariate descriptions, coding, and % missing.

| | Baseline, n = 987 | 12 month, n = 939 | 24 month, n = 1001 | | |
|---|---|---|---|---|---|
| | % missing | % missing | % missing | Variable description | Data source |
| **Outcome Data** | | | | | |
| Enteric infection outcome data available | 24 | 14 | 8.0 | Binary; 0/1 | Based on collection of stool material and successful analysis by GPP |
| STH infection outcome data available | 30 | 37 | 46 | Binary; 0/1 | Based on collection of stool material and successful analysis by Kato-Katz |
| Caregiver-reported diarrhea, 7-day recall | 1.3 | 7.8 | 20 | Binary; 0/1 | Child Survey |
| **Covariate data** | | | | | |
| Child sex, female | 2.3 | 1.3 | 7.0 | Binary; 0=male, 1=female | Child Survey |
| Respondent is child's mother | 2.5 | 7.6 | 20 | Binary; 0/1 | Child Survey |
| Caregiver completed primary school | 0.8 | 1.7 | 6.7 | Binary; 0/1 | Child Survey |
| Child breast feeds with or without complementary feeding | 1.3 | 7.7 | 20 | Binary; 0/1 | Child Survey |
| Child exclusively breastfeeds | 1.3 | 7.7 | 20 | Binary; 0/1 | Child Survey |
| Child wears a diaper | 1.4 | 7.6 | 20 | Binary; 0/1 | Child Survey |
| Child feces is disposed of in a latrine | 1.3 | 7.1 | 20 | Binary; 0/1 | Created from survey questions in Child Survey |
| Child age at sampling, days | 23 | 16 | 17 | Integer | Created from birthdate (Child Survey) and date of sampling |
| Child age at survey, days | 2.6 | 7.5 | 19 | Integer | Created from birthdate (Child Survey) and date of Survey |
| 30-day cumulative rainfall at sampling | 21 | 14 | 10 | Continuous | Created from sample date and data from data from the National Oceanic and Atmospheric Administration's National Centers for Environmental Information (https://www.ncdc.noaa.gov/cdo-web/datatools/findstation) |

*Continued on next page*

*Appendix 1—table 16 continued*

| | Baseline, n = 987 | 12 month, n = 939 | 24 month, n = 1001 | Variable description | Data source |
|---|---|---|---|---|---|
| | % missing | % missing | % missing | | |
| 30-day cumulative rainfall at survey | 1.3 | 7.1 | 19 | Continuous | Created from survey date and data from data from the National Oceanic and Atmospheric Administration's National Centers for Environmental Information (https://www.ncdc.noaa.gov/cdo-web/datatools/findstation) |
| Sample collection during rainy season | 21 | 14 | 10 | Binary; 0/1 | Created from sample date. Rainy season defined as November – April. |
| Survey collection during rainy season | 1.3 | 7.1 | 19 | Binary; 0/1 | Created from survey date. Rainy season defined as November – April. |
| Household crowding, >3 persons/room | 0.4 | 0.3 | 2.7 | Binary; 0/1 | Created from questions in Household Survey |
| Household floor is covered | 0.4 | 0.3 | 2.7 | Binary; 0/1 | Observation |
| Household walls made of concrete, bricks or similar | 0.4 | 0.3 | 2.7 | Binary; 0/1 | Observation |
| Household population | 0.3 | 0.3 | 1.6 | Integer | Household survey |
| Number of rooms in household | 0.4 | 0.3 | 2.3 | Integer | Created from questions in Household Survey |
| Wealth score, 0 (poorest) - 1 (wealthiest), unitless | 0.4 | 0.3 | 2.7 | Continuous | Created from questions in Household Survey using Simple Poverty Scorecard for Mozambique (http://www.simplepovertyscorecard.com/MOZ_2008_ENG.pdf). Questions referencing latrine removed from 12 month and 24 month score. All scores normalized by total number of points available. |
| Household uses tap in compound as primary drinking water source | 1.7 | 1.0 | 2.0 | Binary 0/1 | Created from drinking water source question in Household Survey |
| Latrine has drop-hole cover | 1.9 | 0.0 | 0.0 | Binary; 0/1 | Observation |
| Latrine has a ventpipe | 1.8 | 0.0 | 0.0 | Binary; 0/1 | Observation |
| Latrine has a ceramic, tile, or concrete pedestal or slab | 2.2 | 0.1 | 0.1 | Binary; 0/1 | Observation |
| Latrine has sturdy walls made of concrete, bricks, or similar | 1.9 | 0.0 | 0.0 | Binary; 0/1 | Observation |
| Compound population | 0.0 | 0.0 | 0.0 | Integer | Compound Survey, enrollment checklists |
| Number of households in compound | 0.0 | 0.0 | 0.0 | Integer | Compound Survey, enrollment checklists |
| Number of latrines present in the compound | 0.1 | 0.0 | 0.0 | Integer | Compound Survey |
| Persons per latrine | 1.8 | 0.1 | 0.3 | Continuous | Created by dividing the compound population by the number of latrines/drop-holes |
| Households per latrine | 1.8 | 0.1 | 0.3 | Continuous | Created by dividing the number of households in the compound by the number of latrines in the compound |
| Number of water taps present in the compound | 1.1 | 0.0 | 0.0 | Integer | Compound Survey |

*Continued on next page*

*Appendix 1—table 16 continued*

| | Baseline, n = 987 | 12 month, n = 939 | 24 month, n = 1001 | Variable description | Data source |
|---|---|---|---|---|---|
| | % missing | % missing | % missing | | |
| Standing water visible around compound grounds | 1.9 | 0.3 | 0.0 | Binary; 0/1 | Observation |
| Standing or leaking wastewater visible around compound grounds | 1.9 | 0.3 | 0.0 | Binary; 0/1 | Observation |
| Faeces or used diapers observed around compound grounds or in solid waste | 1.9 | 0.3 | 0.0 | Binary; 0/1 | Observation |
| Compound floods when it rains | 0.0 | 0.0 | 0.0 | Binary; 0/1 | Compound Survey |
| Compound has electricity that normally functions | 0.0 | 0.0 | 0.0 | Binary; 0/1 | Compound Survey |
| Compound-level population density | 2.2 | 1.5 | 1.5 | Continuous, persons/m$^2$ | Created by dividing the population of the compound by the measured area of the compound |
| Any animal present in the compound | 0.0 | 0.4 | 0.0 | Binary; 0/1 | Observation |
| Dog(s) present in the compound | 0.0 | 0.4 | 0.0 | Binary; 0/1 | Observation |
| Chicken(s) and/or duck(s) present in the compound | 0.0 | 0.4 | 0.0 | Binary; 0/1 | Observation |
| Cat(s) present in the compound | 0.0 | 0.4 | 0.0 | Binary; 0/1 | Observation |
| Any other animal(s) present in the compound | 0.0 | 0.4 | 0.0 | Binary; 0/1 | Observation |

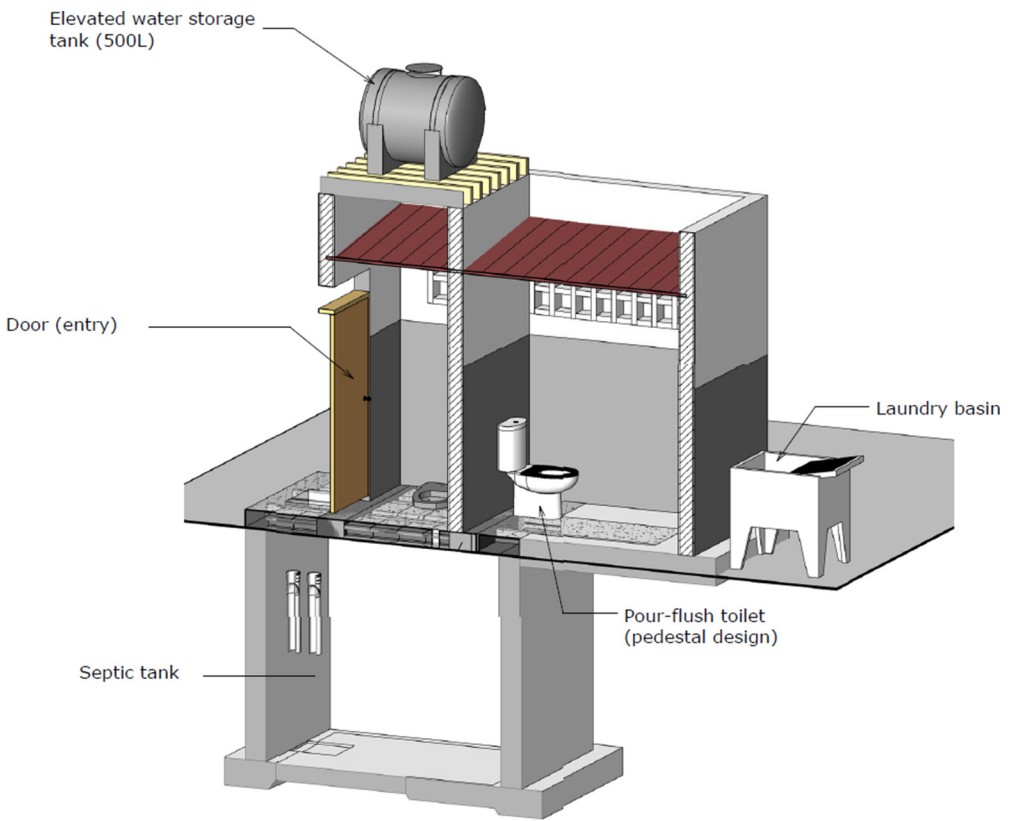

**Appendix 1—figure 5.** Schematic of communal sanitation block design from the NGO (Water and Sanitation for the Urban Poor). Pictured: two latrine stalls, two pour-flush toilets, septic tank, elevated water storage tank, laundry basin, door. Not pictured: soakaway pit. Source: Water and Sanitation for the Urban Poor.

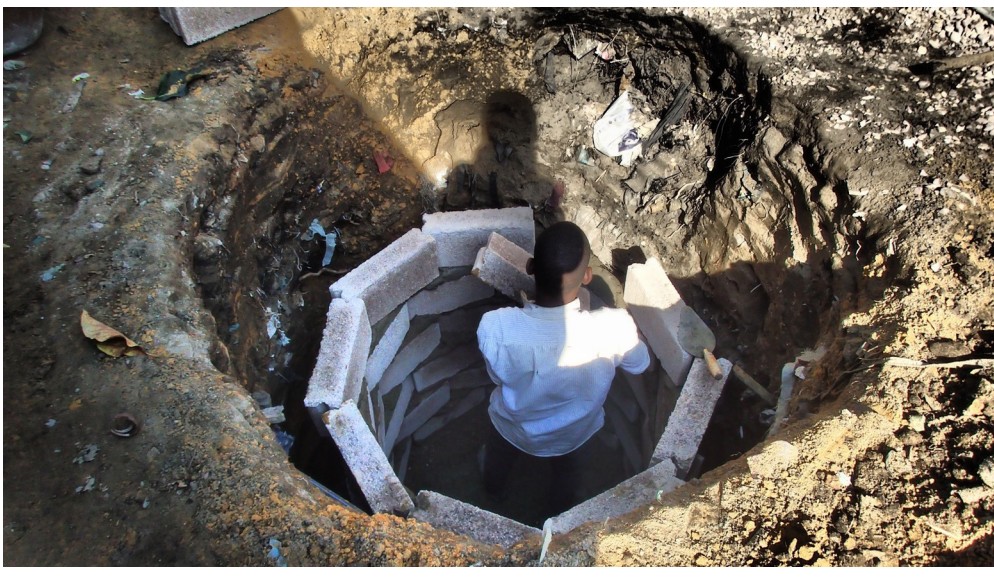

**Appendix 1—figure 6.** Construction of a soakaway pit for discharge of liquid effluent from intervention latrines.

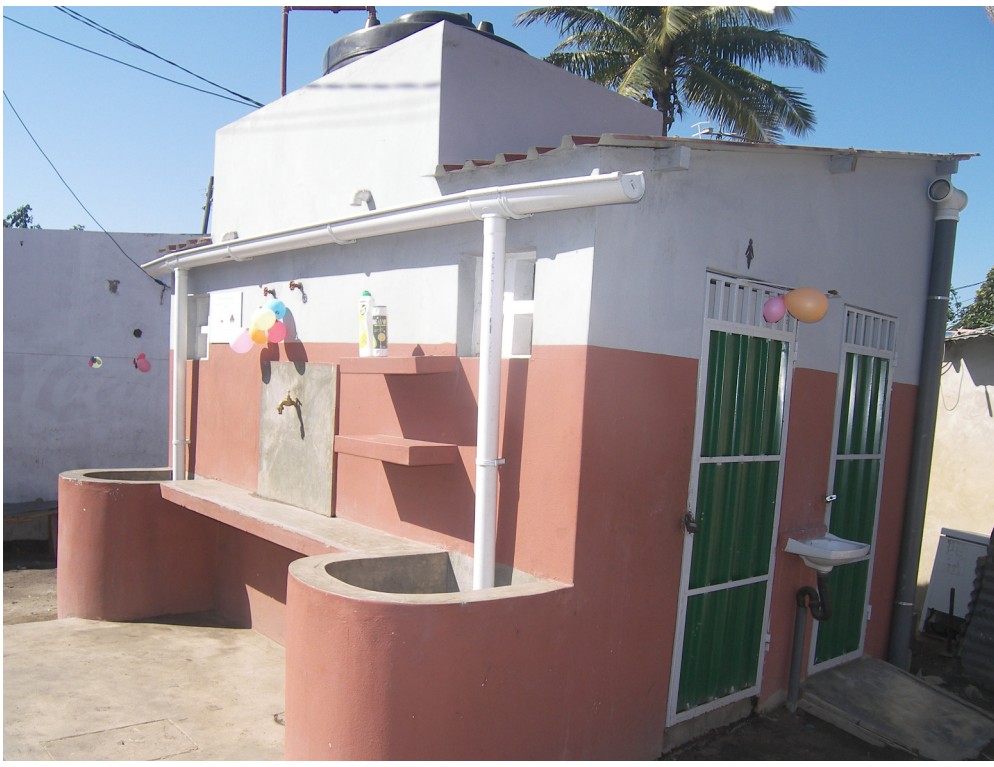

**Appendix 1—figure 7.** Photo of communal sanitation block as constructed.

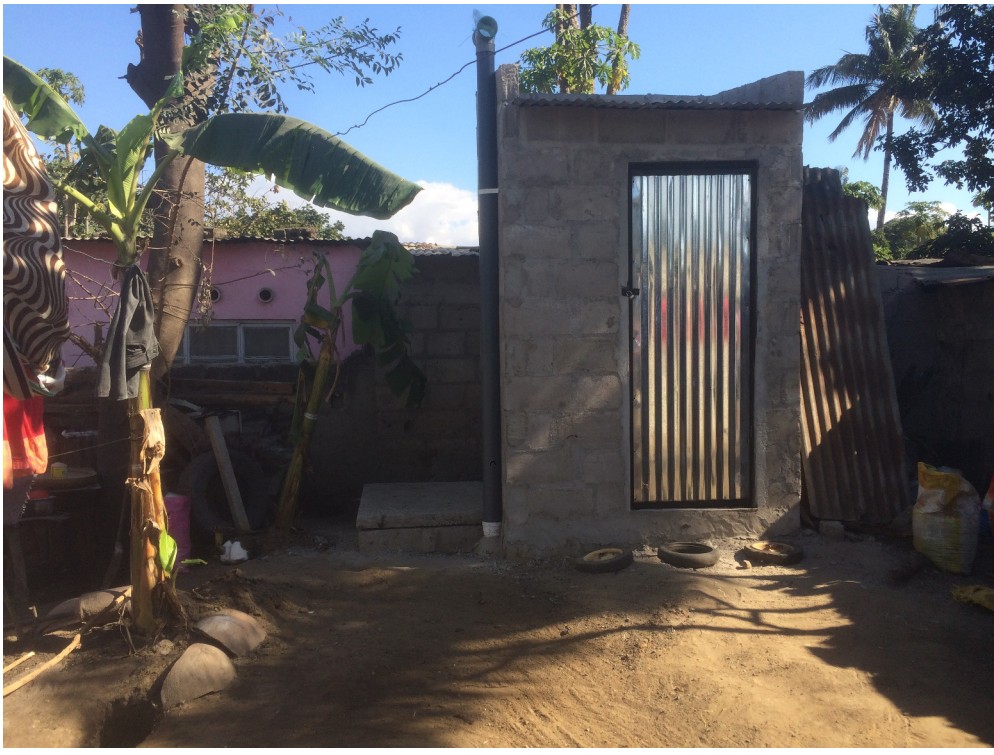

**Appendix 1—figure 8.** Photo of shared latrine as constructed.

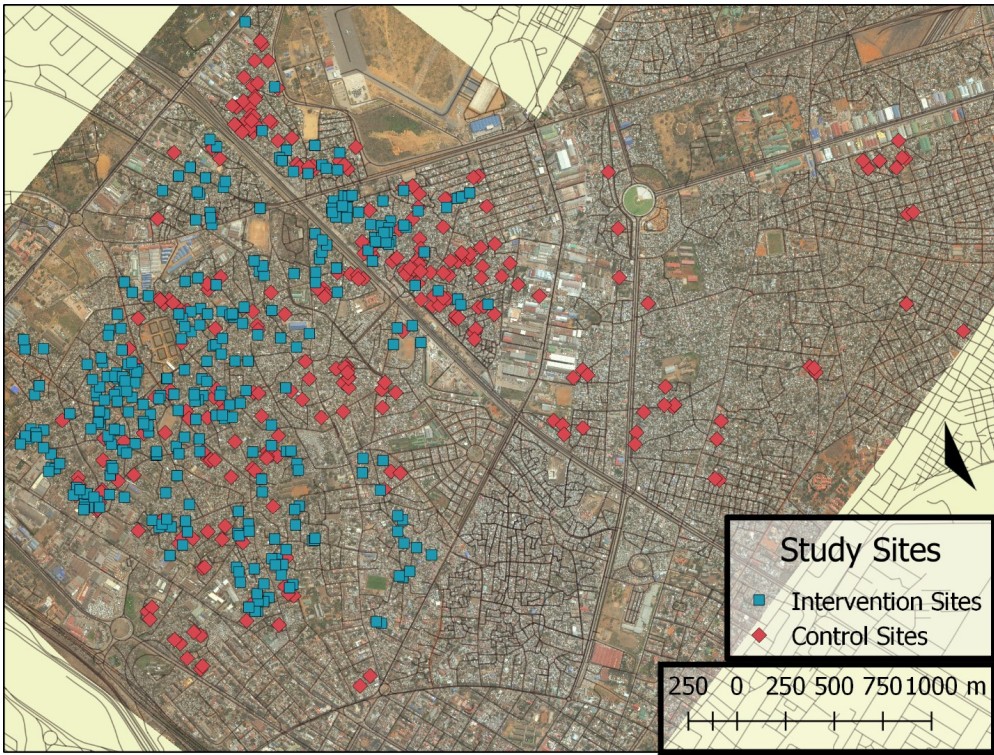

**Appendix 1—figure 9.** Map illustrating locations of intervention (n=208) and control sites (n=287) compounds.

