## [Decision Letter]

**Acceptance summary:**

This is an important study in a difficult and under-studied (specifically urban) setting, which adds to the growing body of evidence challenging the effectiveness of urban sanitation interventions to address the burden of infectious diarrhea in low income countries.

**Decision letter after peer review:**

Thank you for submitting your article "Effects of an urban sanitation intervention on childhood enteric infection and diarrhoea in Mozambique" for consideration by *eLife*. Your article has been reviewed by 3 peer reviewers, including Joseph Lewnard as the Reviewing Editor and Reviewer #1, and the evaluation has been overseen by a Senior Editor. The following individual involved in review of your submission has agreed to reveal their identity: James Platts Mills (Reviewer #3).

As is customary in *eLife*, the reviewers have discussed their critiques with one another. What follows below is the Reviewing Editor's edited compilation of the essential and ancillary points provided by reviewers in their critiques and in their interaction post-review. Please submit a revised version that addresses these concerns directly. Although we expect that you will address these comments in your response letter, we also need to see the corresponding revision in the text of the manuscript. Some of the reviewers' comments may seem to be simple queries or challenges that do not prompt revisions to the text. Please keep in mind, however, that readers may have the same perspective as the reviewers. Therefore, it is essential that you attempt to amend or expand the text to clarify the narrative accordingly.

Given the present situation, we will give authors as much time as they need to submit revised manuscripts. We are also offering, if you choose, to post the manuscript to bioRxiv (if it is not already there) along with this decision letter and a formal designation that the manuscript is "in revision at *eLife*". Please let us know if you would like to pursue this option. (If your work is more suitable for medRxiv, you will need to post the preprint yourself, as the mechanisms for us to do so are still in development.)

Summary:

This is an important study in a difficult (and under-studied, specifically urban) setting, which adds to the growing body of evidence challenging the effectiveness of WASH interventions to address the burden of infectious diarrhea. The findings regarding cohorts born into the intervention clusters are particularly novel and merit a bit more attention than what is presently afforded.

Essential revisions:

1. The analysis strategy for a non-randomized intervention is generally appropriate but some more descriptive details about both design (e.g. control group selection) and analysis are needed.

2. We feel a bit more information on the predictors of infection would be appropriate to offer some insight into why the intervention failed, and what this teaches us. The absence of an effect is interesting insofar is it further chips away at the notion long prevailing in international development circles and others that WASH interventions are a solution to diarrhea burden and suggests routes of transmission that remain poorly explicated, although this paper in its current form does not shed light on determinants of infection. As rich covariate data were collected this type of analysis would strengthen the paper and make it more suitable for *eLife*, where mechanism is of greater interest, as compared to a clinical journal which would care about an yes-no answer to whether the intervention worked. Are there covariates which tell us what may contribute to infection and therefore offer insight into why the intervention failed?

3. The reviewers felt the excruciating delineation of "primary" and "secondary" and "exploratory" outcomes is not of much scientific value. In fact, the analyses described as "exploratory" pack the most interesting and novel information learned in this trial (i.e. effect specifically among kids born into the intervention clusters). Address limitations of power or design as appropriate for these outcomes. Even if the study is underpowered to show significant effects against each individual pathogen the directional consistency across multiple point estimates has a low probability of occurring under the null hypothesis of no effect. We would like to see this finding from the trial and its biological implications better explicated-at present no attempt is made to account for the observation and it is dismissed out of hand as a non-primary analysis. One hypothesis could be that the prevention of early life infections/reduction in early life exposure among those born into intervention clusters (and thus adverse immune/developmental sequelae e.g. environmental enteric dysfunction) abrogates pathways that would enhance susceptibility later in life. Many others are also possible.

4. Some considerations about how the analyses were undertaken should be addressed:

a. An important assumption is that the age and seasonal distribution of stool collection was similar between the two groups both at baseline and 24 months, and if that it was not, adjusting for age (presumably as a linear effect) and considering 30-day cumulative rainfall as a potential covariate in the adjusted analysis was sufficient to account for this. Supplemental Figure 1 appears to demonstrate that there were significant changes in the rate of enrollment over the course of the study, and that these rates differed between the intervention and control groups (both at baseline and during follow-up).

b. For age – the relationship between age and pathogen prevalence is non-linear for many pathogens (roughly increases from birth and then peaks between 6-18 months of age and then declines). Were higher order terms or splines considered to adjust for age, both for the primary analysis and especially for the <24 month sub-group analysis? It would also be helpful to analyze and present the results of the other age-based sub-group (24-48 months at baseline and 24-48 months old at 24-month follow-up) – why was overall Shigella prevalence not different between groups if it was so strikingly different for the youngest children (was the effect reversed in older children, and was this confounded by the age distribution between the groups)?

c. For season – rainfall is not the only observable parameter; if there is a difference between the seasonal distribution of stool collection between the groups (as Figure 1a, 1b, and 1c would suggest), were splines or sine/cosine terms considered to adjust for seasonal variation in stool collection? Clearly this is a complicated modeling topic, but it would be helpful to understand how the study group thoughts about this.

5. All reviewers were in agreement that more extensive discussion is needed to situate this work, and these findings, on the context of previous literature. It's important in scientific literature to build on previous findings and look for supporting evidence from other studies to back up your findings or new hypotheses that have arisen from the findings. Notably, *eLife* does not limit citations, so relevant work by other groups should be referenced. For instance:

a. line 369-372 in discussion – It would be helpful for the authors to do a more nuanced review of the previous literature on the impact of sanitation interventions on enteric infections (given it's the primary outcome here), even if they are mostly rural. In particular, it would be nice to see a discussion of the WASH Benefits results of the impact of the WASH interventions on protozoa and STH infections (these outcomes were measured in both the Bangladesh and Kenya trials), and the results of the SHINE WASH intervention on enteric infections. For context, relatively few studies of WASH interventions have used enteropathogen outcomes (usually diarrhea, and more recently growth)

b. It is an exploratory analysis, and multiple comparisons are made, but if the findings of the sub-group analysis are believable and plausible this clearly still strengthens the inference that can be made. For example – Shigella, STHs, and Cryptosporidium (though underpowered and not statistically significant) show reductions in the children born in intervention compounds. These are all anthroponotic fecal-oral pathogens with relatively high inoculation sizes. Campylobacter and ETEC and Giardia (other high prevalence bacterial/protozoal pathogens) may also come from animals and contaminated water, thus perhaps would be less likely to be reduce by a sanitation intervention alone. Viral pathogens may be more likely to be spread by direct contact and crowding, and thus handwashing may be more important, etc.

6. Several key limitations were suggested by the reviewers which should be discussed explicitly; the most important of these are:

a. The intervention was not delivered at the community level, which has been a major criticism of previous sanitation trials given that neighboring compound sanitation conditions can affect the risk of child exposure to fecal contamination and pathogens.

b. The first primary outcome – >= 1 bacterial or parasitic infection – was likely largely biased towards the null because of the very high prevalence of this outcome especially in older children. Combined with this, the xTAG system is qualitative, and thus cannot distinguish between what can be million-fold difference in pathogen quantities between stools. Thus one child with a trace detection of one pathogen is considered to have the outcome, as is a child with multiple high-quantity detections. As a result, ~80% or more of children at baseline and during follow-up had the outcome. Even in this sub-group analysis of children < 24 months at enrolment and born during follow-up, the outcome was very common. This limitation should be more clearly stated. Relatedly, the authors should note a limitation of the study is not having quantitative results of enteric infection, which has been shown to be valuable in previous work.

c. The second primary outcome, STH, was only assessed in a subset of children (per Tables 2 and 3) – especially during follow-up. The reason for this should be explained and any potential limitations discussed.

d. The intervention was mixed, including both CSBs and SLs, and thus some of the benefit may have been not from the improved latrine but instead other aspects of the CSB. This includes a handwashing station, but the majority of intervention compounds received SLs, which did not. I am also not clear as to whether the water storage is for flush toilets alone or also for drinking water. This could be made clear, and the heterogeneity of the intervention could be discussed as a limitation.

e. A few aspects of the compound selection differed between intervention and control compounds. I particular, intervention compounds appear to have at baseline been more crowded. While the difference-in-difference approach intrinsically adjusts for associated differences in pathogen carriage between these compounds at baseline, crowding has been identified as an independent risk factor for transmission of enteric pathogens, and thus this may have biased the intervention effect towards the null. This should be discussed as a limitation.

---

## [Author Response]

Essential revisions:1. The analysis strategy for a non-randomized intervention is generally appropriate but some more descriptive details about both design (e.g. control group selection) and analysis are needed.

We have addressed this in the revised version.

2. We feel a bit more information on the predictors of infection would be appropriate to offer some insight into why the intervention failed, and what this teaches us. The absence of an effect is interesting insofar is it further chips away at the notion long prevailing in international development circles and others that WASH interventions are a solution to diarrhea burden and suggests routes of transmission that remain poorly explicated, although this paper in its current form does not shed light on determinants of infection. As rich covariate data were collected this type of analysis would strengthen the paper and make it more suitable for eLife, where mechanism is of greater interest, as compared to a clinical journal which would care about an yes-no answer to whether the intervention worked. Are there covariates which tell us what may contribute to infection and therefore offer insight into why the intervention failed?

We agree that additional information on predictors of infection among the study population would help contextualize and explain our findings. We added the following to the Discussion section in response this comment (As additions were extensive, only descriptions and line numbers are provided here).

– Description of results from two risk factor analyses performed on data collected at baseline. The analyses aimed to identify risk factors for enteric infection an environmental fecal contamination in the study population/site prior to introduction of the intervention. (Lines 328–358)

– Description of results from a forthcoming manuscript which evaluated the impact of the intervention on measures of fecal indicators in various environmental samples. We also describe these results within the context of previous studies which have similarly evaluated the impact of sanitation interventions on environmental fecal contamination. (Lines 357-367)

3. The reviewers felt the excruciating delineation of "primary" and "secondary" and "exploratory" outcomes is not of much scientific value. In fact, the analyses described as "exploratory" pack the most interesting and novel information learned in this trial (i.e. effect specifically among kids born into the intervention clusters). Address limitations of power or design as appropriate for these outcomes. Even if the study is underpowered to show significant effects against each individual pathogen the directional consistency across multiple point estimates has a low probability of occurring under the null hypothesis of no effect. We would like to see this finding from the trial and its biological implications better explicated-at present no attempt is made to account for the observation and it is dismissed out of hand as a non-primary analysis. One hypothesis could be that the prevention of early life infections/reduction in early life exposure among those born into intervention clusters (and thus adverse immune/developmental sequelae e.g. environmental enteric dysfunction) abrogates pathways that would enhance susceptibility later in life. Many others are also possible.

As this is a registered clinical trial with pre-defined outcomes, we feel it is important to classify our results in accordance with the clinical trial registration given the number of statistical tests that are possible in this dataset and the potential for type I error. That said, we agree that the sub-group analysis of children born in to the study by the 24-month visit deserves further discussion. We have added the following text to the second paragraph of the discussion to address the ‘limitations of power or design’ (lines 256-262):

“The trial was neither designed nor powered to detect differences in sub-groups of children such as those born after the intervention was implemented, potentially limiting our ability to detect small effects in such analyses. […] However, the magnitude of the effect estimates for the outcomes of any STH, Trichuris, and Shigella observed among children born into the study by the 24-month visit, and the directional consistency of effect estimates among most other outcomes in this sub-group analysis, strengthens the plausibility of these findings.”

We have also added additional text to the discussion detailing the potential biological

implications of our sub-group analysis findings (lines 263-280).

“There are several reasons we observed suggestive evidence of an effect for some outcomes among this sub-group of young children but not among older children or in the main analyses. […] The intervention would have no effect on such infections, highlighting the potentially important role of protection from birth.”

4. Some considerations about how the analyses were undertaken should be addressed:a. An important assumption is that the age and seasonal distribution of stool collection was similar between the two groups both at baseline and 24 months, and if that it was not, adjusting for age (presumably as a linear effect) and considering 30-day cumulative rainfall as a potential covariate in the adjusted analysis was sufficient to account for this. Supplemental Figure 1 appears to demonstrate that there were significant changes in the rate of enrollment over the course of the study, and that these rates differed between the intervention and control groups (both at baseline and during follow-up).b. For age – the relationship between age and pathogen prevalence is non-linear for many pathogens (roughly increases from birth and then peaks between 6-18 months of age and then declines). Were higher order terms or splines considered to adjust for age, both for the primary analysis and especially for the <24 month sub-group analysis? It would also be helpful to analyze and present the results of the other age-based sub-group (24-48 months at baseline and 24-48 months old at 24-month follow-up) – why was overall Shigella prevalence not different between groups if it was so strikingly different for the youngest children (was the effect reversed in older children, and was this confounded by the age distribution between the groups)?

We considered including a higher order age term (age-squared) in our main and sub-group analyses but ultimately decided to present the simpler models containing only the age (in days) term. We have added text to the results describing results from sensitivity analyses which included an age-squared term and present those results Appendix 1-table 10. To demonstrate the non-linear relationship between age and prevalence for many pathogens measured in this study, we have added Appendix 1-figure 4 that presents the age-prevalence relationship for most pathogen outcomes (excluding those with very low prevalence throughout the study, e.g. *V. cholerae*) at each phase and stratified by study arm. Please see lines 206-209.

“While the relationship between age and pathogen prevalence appeared to be non-linear for many pathogens (Appendix 1- figure 4), the inclusion of a higher order age term (age squared) did not meaningfully change effect estimates in the main or sub-group analyses (Appendix 1- table 10).”

As suggested, we have also included in Appendix 1 the results of a sub-group analysis of children aged >24 months old at baseline and the 24-month follow-up phase. These data represent intervention effect estimates on older children who were born before the intervention was implemented. In general, effect estimates for this older sub-group of children were closer to the null value and we observed no significant intervention effects. In a few cases, the direction of effect estimate was opposite to estimates from the analysis children <2 years (e.g. Any STH outcome, Shigella, diarrhea). It’s likely the results of the older children served to skew effect estimates towards the null value in the main analyses, particularly as there were three times as many data points for children >2 years (n=340 control, n=344 intervention) than for children <2 years (n=106 control, n=107 intervention) at the 24-month phase. We have added text to the Results section to describe these findings. Please see lines 233-235.

“These effects were attenuated in sub-group analyses restricted to older children (>24 months) who were born before the intervention was implemented and present at the 24-month phase (Appendix 1-table 12).”

Further, the age distributions of intervention and control children were similar at each phase. We have added text to the results and a figure of age distributions at each phase (kernel density plots) to Appendix 1 to illustrate this point. Lines 139-141.

“The age distributions of intervention and control children were similar at baseline and both follow-up phases (Appendix 1- figure 3).”

c. For season – rainfall is not the only observable parameter; if there is a difference between the seasonal distribution of stool collection between the groups (as Figure 1a, 1b, and 1c would suggest), were splines or sine/cosine terms considered to adjust for seasonal variation in stool collection? Clearly this is a complicated modeling topic, but it would be helpful to understand how the study group thoughts about this.

In addition to the measure of cumulative rainfall, we have also assessed a binary variable defining the wet (November – April) and dry (May – October) seasons as a potential confounder. Neither the rainfall nor the binary seasonality variable met inclusion criteria for multivariable models as their inclusion in the main difference-in-difference models had limited impact on effect estimates. Further, as suggested by the reviewers, we included sine and cosine terms (e.g. sin[(2*pi/365)*t] and cos[(2*pi/365)*t] where t=day of the year). in our main models. Again, inclusion of these terms in the 12- and 24-month analyses had no meaningful effect on prevalence ratios of any outcome measured and were excluded from final multivariable models. We have included details of the assessment for all three seasonality measures in the methods, results, and discussion text and in Appendix 1- tables 9 and 11.

Methods Text (lines 677-682)

“We assessed the potential impact of seasonality on our results in three ways: (1) inclusion of binary indicator of wet (November – April) and dry (May – October) season in multivariable models, (2) inclusion of a variable representing cumulative rainfall (mm) 30 days prior to sample or survey collection in multivariable models, and (3) inclusion of sine and cosine functions of sample and survey dates in multivariable models (Appendix 1-table 9 and Appendix 1-table 11).”

Results text (lines 209-217)

“Three measures of seasonality were considered for inclusion in multivariable models to adjust for any difference in seasonal distributions of data collection: (1) a binary variable defining the ‘rainy’ (November – April) and ‘dry’ seasons (May – October) in Maputo, (2) a measure of cumulative rainfall (mm) in the 30 days prior to data collection, and (3) sine and cosine terms representing dates of sample collection. While there was some imbalance between arms in data collected during the wet and dry seasons at baseline (Appendix 1- table 9), no measure of seasonality meaningfully changed effect estimates in the 12- and 24-month analyses and seasonality was excluded from final multivariable models (Appendix 1-table 9 and Appendix 1-table 11).”

Discussion text (lines 516-520):

“We had limited ability to evaluate the impact of seasonality or weather-related trends on our effect estimates due to drought conditions during the 2015/2016 rainy season. We adjusted models for cumulative 30-day rainfall, a binary indicator of wet/dry season, and sine/cosine terms of sample collection date (Stolwijk, Straatman, and Zielhuis, 1999) but excluded all seasonality terms from final multivariable models because they did not meaningfully change effect estimates.”

5. All reviewers were in agreement that more extensive discussion is needed to situate this work, and these findings, on the context of previous literature, it's important in scientific literature to build on previous findings and look for supporting evidence from other studies to back up your findings or new hypotheses that have arisen from the findings. Notably, eLife does not limit citations, so relevant work by other groups should be referenced. For instance:a. line 369-372 in discussion – It would be helpful for the authors to do a more nuanced review of the previous literature on the impact of sanitation interventions on enteric infections (given it's the primary outcome here), even if they are mostly rural. In particular, it would be nice to see a discussion of the WASH Benefits results of the impact of the WASH interventions on protozoa and STH infections (these outcomes were measured in both the Bangladesh and Kenya trials), and the results of the SHINE WASH intervention on enteric infections. For context, relatively few studies of WASH interventions have used enteropathogen outcomes (usually diarrhea, and more recently growth)

We agree a more thorough discussion of previous literature would be useful in contextualizing our results. We have made extensive additions to the Discussion section. Please see lines 290-328.

b. It is an exploratory analysis, and multiple comparisons are made, but if the findings of the sub-group analysis are believable and plausible this clearly still strengthens the inference that can be made. For example – Shigella, STHs, and Cryptosporidium (though underpowered and not statistically significant) show reductions in the children born in intervention compounds. These are all anthroponotic fecal-oral pathogens with relatively high inoculation sizes. Campylobacter and ETEC and Giardia (other high prevalence bacterial/protozoal pathogens) may also come from animals and contaminated water, thus perhaps would be less likely to be reduce by a sanitation intervention alone. Viral pathogens may be more likely to be spread by direct contact and crowding, and thus handwashing may be more important, etc.

We have added text to the discussion highlighting the biological plausibility of our findings for Shigella and STHs (namely Trichuris) (lines 281-289)

“Notably, both Shigella and Trichuris are primarily anthroponotic, and infection with both is strongly age-dependent in this study population (Knee et al., 2018). […] This supports the hypothesis that the intervention may have reduced the overall frequency of exposure enough to impact Shigella and Trichuris infection among young children but not older children.”

6. Several key limitations were suggested by the reviewers which should be discussed explicitly; the most important of these are:a. The intervention was not delivered at the community level, which has been a major criticism of previous sanitation trials given that neighboring compound sanitation conditions can affect the risk of child exposure to fecal contamination and pathogens.

This is an important point and one hypothesis as to why we saw only a limited effect of the intervention. Rather than list it as a limitation, we have expanded and clarified the discussion of this point as a potential reason this intervention was ineffective. Please see the paragraph starting on line 385-407.

“The intervention was delivered at the compound level, not the community level, and was not designed to achieve any specified threshold of sanitation coverage in the study neighborhoods. […] In addition to neighborhood-level exposures, the transience of the study population meant that trips to and from provinces outside of Maputo, where exposures were varied and unmeasured, were common.”

b. The first primary outcome – >= 1 bacterial or parasitic infection – was likely largely biased towards the null because of the very high prevalence of this outcome especially in older children. Combined with this, the xTAG system is qualitative, and thus cannot distinguish between what can be million-fold difference in pathogen quantities between stools. Thus one child with a trace detection of one pathogen is considered to have the outcome, as is a child with multiple high-quantity detections. As a result, ~80% or more of children at baseline and during follow-up had the outcome. Even in this sub-group analysis of children < 24 months at enrolment and born during follow-up, the outcome was very common. This limitation should be more clearly stated.

We have added more explicit description of this limitation in the discussion. Lines 491-499.

“Our ability to detect an effect on our primary outcome, the prevalence of ≥1 bacterial or protozoan infection, may have been limited by (1) the extended duration of shedding of some pathogens following active infection; (2) the overall high burden of disease in our study population, particularly among older children; and (3) residual confounding by age given the strong observed relationship between age and infection status (particularly for protozoan pathogens), all of which may have biased our results toward the null. Further, the intervention may have impacted the concentration of pathogens shed (Grembi et al., 2020; Lin et al., 2019), but our binary outcome was not sensitive to such differences The qualitative nature of the GPP did not allow us to interrogate this question.”

Relatedly, the authors should note a limitation of the study is not having quantitative results of enteric infection, which has been shown to be valuable in previous work.

We have added additional text to the discussion to highlight the utility of quantitative pathogen detection. Lines 476-480.

ׅ“Quantitative results, like those produced by multiplex quantitative PCR panels, can be used to aid identification of etiologic agents of diarrhea, especially in cases of coinfection, and to differentiate between low-level enteric pathogen detection of unknown clinical relevance and higher concentration shedding which is more clearly associated with disease (Liu et al., 2014, 2016; Platts-Mills, Liu, and Houpt, 2013).”

c. The second primary outcome, STH, was only assessed in a subset of children (per Tables 2 and 3) – especially during follow-up. The reason for this should be explained and any potential limitations discussed.

STH analysis was limited to whole stool samples as we relied on the Kato Katz method for detection. Therefore, diarrheal diaper samples and rectal swabs, the latter of which we began collecting from the 12-month phase and onward, could not be tested for STH by Kato Katz. Further, when a limited volume of whole stool was collected, we prioritized shipment to Georgia Tech for molecular analysis of enteric pathogens. We have added text to the discussion explaining the discrepancy in sample numbers for the primary outcome and STH outcomes as well as a discussion of the potential limitations it introduced. Lines 500-508.

“We analyzed a smaller number of stool samples for STH than for other enteric pathogens due to requirements of the Kato-Katz method used for STH detection. […] This limited the potential impact that sample type could have on our results.”

d. The intervention was mixed, including both CSBs and SLs, and thus some of the benefit may have been not from the improved latrine but instead other aspects of the CSB. This includes a handwashing station, but the majority of intervention compounds received SLs, which did not. I am also not clear as to whether the water storage is for flush toilets alone or also for drinking water. This could be made clear, and the heterogeneity of the intervention could be discussed as a limitation.

We have clarified the intended use of the shared water connection installed with the CSBs in the methods section. These connections were a part of the municipal piped water system and supplied the same water most compounds used for drinking and other domestic purposes. Lines 557-559.

“Shared piped water connections were part of the municipal water system and could be used for drinking in addition to other domestic purposes. Rainwater was intended for cleaning and flushing but not drinking.”

The differences in design features between CSBs and SLs could modify the effect of the intervention, as you describe. Unfortunately, our study was not powered to detect intervention effects in a stratified analysis. Further, even within the two broad categories of intervention type, there exists heterogeneity. Water and Sanitation for the Urban Poor, the NGO that implemented the intervention, encouraged all compound members to independently upgrade and invest in their facility over time. We have added a paragraph addressing these points to the discussion. Lines 421-427.

“The two intervention designs we evaluated in this study – communal sanitation blocks and shared latrines – utilized the same basic sanitation technology but differed in the number of cabins and amenities available. […] Moreover, all intervention compounds were encouraged to independently upgrade their facilities by adding features like electricity and handwashing stations, or by connecting existing handwashing stations to the water supply, resulting in heterogeneity even within the two broad categories of intervention type.”

e. A few aspects of the compound selection differed between intervention and control compounds. I particular, intervention compounds appear to have at baseline been more crowded. While the difference-in-difference approach intrinsically adjusts for associated differences in pathogen carriage between these compounds at baseline, crowding has been identified as an independent risk factor for transmission of enteric pathogens, and thus this may have biased the intervention effect towards the null. This should be discussed as a limitation.

As part of the compound selection process, we attempted to enroll intervention and control compounds with similar numbers of residents However, many of the largest eligible compounds in the study area were selected by the NGO as intervention sites, resulting in the observed difference in intervention and control resident populations at baseline. The DID analysis coupled with our CBA design allows us to adjust for baseline differences in infection between arms, which may be the most relevant predictor of future infection. Further, we found limited evidence that crowding was associated with risk of infection at baseline. We have added the following text to the limitations section of the discussion to address these concerns. Lines 440-451.

“While we attempted to enroll intervention and control compounds with comparable numbers of residents, the NGO which identified and implemented the intervention selected most of the largest eligible compounds for intervention. […] We consider our analysis to be robust to small differences in study arms at baseline, however, we cannot exclude the possibility of residual confounding due to such differences, a limitation of non-randomized designs.”

References:

Barreto, M. L., Genser, B., Strina, A., Teixeira, M. G., Marlucia, A., Assis, O., Rego, R. F., Teles, C. A., Prado, M. S., Matos, S. M. A., Santos, D. N., Santos, L. A. Dos, and Cairncross, S. (2007). Effect of city-wide sanitation programme on reduction in rate of childhood diarrhea in northeast Brazil: assessment by two cohort studies. In *www.thelancet.com* (Vol. 370). www.thelancet.comBenjamin-Chung, J., Pilotte, N., Ercumen, A., Grant, J. R., Maasch, J. R. M. A., Gonzalez, A. M., Ester, A. C., Arnold, B. F., Rahman, M., Haque, R., Hubbard, A. E., Luby, S. P., Williams, S. A., and Colford, J. M. (2020). Comparison of multi-parallel qPCR and double-slide kato-katz for detection of soil-transmitted helminth infection among children in rural Bangladesh. PLoS Neglected Tropical Diseases, 14(4), 1–23. https://doi.org/10.1371/journal.pntd.0008087Berendes, D. M., Kirby, A. E., Clennon, J. A., Agbemabiese, C., Ampofo, J. A., Armah, G. E., Baker, K. K., Liu, P., Reese, H. E., Robb, K. A., Wellington, N., Yakubu, H., and Moe, C. L. (2018). Urban sanitation coverage and environmental fecal contamination: Links between the household and public environments of Accra, Ghana. PLOS ONE, 13(7), e0199304. https://doi.org/10.1371/journal.pone.0199304Chhabra, P., Gregoricus, N., Weinberg, G. A., Halasa, N., Chappell, J., Hassan, F., Selvarangan, R., Mijatovic-Rustempasic, S., Ward, M. L., Bowen, M., Payne, D. C., and Vinjé, J. (2017). Comparison of three multiplex gastrointestinal platforms for the detection of gastroenteritis viruses. Journal of Clinical Virology, 95, 66–71. https://doi.org/10.1016/j.jcv.2017.08.012Cools, P., Vlaminck, J., Albonico, M., Ame, S., Ayana, M., José Antonio, B. P., Cringoli, G., Dana, D., Keiser, J., Maurelli, M. P., … Levecke, B. (2019). Diagnostic performance of a single and duplicate Kato-Katz, Mini-FLOTAC, FECPAKG2 and qPCR for the detection and quantification of soil-transmitted helminths in three endemic countries. PLOS Neglected Tropical Diseases, 13(8), e0007446. https://doi.org/10.1371/journal.pntd.0007446Deng, J., Luo, X., Wang, R., Jiang, L., Ding, X., Hao, W., Peng, Y., Jiang, C., Yu, N., and Che, X. (2015). A comparison of Luminex xTAGGastrointestinal Pathogen Panel (xTAG GPP) and routine tests for the detection of enteropathogens circulating in Southern China. Diagnostic Microbiology and Infectious Disease, 83(3), 325–330. https://doi.org/10.1016/j.diagmicrobio.2015.07.024Duong, V. T., Phat, V. V., Tuyen, H. T., Dung, T. T. N., Trung, P. D., Minh, P. Van, Phuong Tu, L. T., Campbell, J. I., Phuc, H. Le, Thanh Ha, T. T., … Baker, S. (2016). Evaluation of luminex xTAG gastrointestinal pathogen panel assay for detection of multiple diarrheal pathogens in fecal samples in Vietnam. Journal of Clinical Microbiology, 54(4), 1094–1100. https://doi.org/10.1128/JCM.03321-15Else, K. J., Keiser, J., Holland, C. V., Grencis, R. K., Sattelle, D. B., Fujiwara, R. T., Bueno, L. L., Asaolu, S. O., Sowemimo, O. A., and Cooper, P. J. (2020). Whipworm and roundworm infections. In Nature Reviews Disease Primers (Vol. 6, Issue 1, pp. 1–23). Nature Research. https://doi.org/10.1038/s41572-020-0171-3Freeman, K., Tsertsvadze, A., Taylor-Phillips, S., McCarthy, N., Mistry, H., Manuel, R., and Mason, J. (2017). Agreement between gastrointestinal panel testing and standard microbiology methods for detecting pathogens in suspected infectious gastroenteritis: Test evaluation and meta-analysis in the absence of a reference standard. PLOS ONE, 12(3), e0173196. https://doi.org/10.1371/journal.pone.0173196Fuller, J. A., and Eisenberg, J. N. S. (2016). Herd Protection from Drinking Water, Sanitation, and Hygiene Interventions. The American Journal of Tropical Medicine and Hygiene, 95(5), 1201–1210. https://doi.org/10.4269/ajtmh.15-0677Fuller, J. A., Villamor, E., Cevallos, W., Trostle, J., and Eisenberg, J. N. S. (2016). I get height with a little help from my friends: herd protection from sanitation on child growth in rural Ecuador. International Journal of Epidemiology. https://doi.org/10.1093/ije/dyv368Grembi, J. A., Lin, A., Karim, M. A., Islam, M. O., Miah, R., Arnold, B. F., McQuade, E. T. R., Ali, S., Rahman, M. Z., Hussain, Z., … Luby, S. P. (2020). Effect of Water, Sanitation, Handwashing, and Nutrition Interventions on Enteropathogens in Children 14 Months Old: A Cluster-Randomized Controlled Trial in Rural Bangladesh. The Journal of Infectious Diseases. https://doi.org/10.1093/infdis/jiaa549Harris, M., Alzua, M. L., Osbert, N., and Pickering, A. (2017). Community-Level Sanitation Coverage More Strongly Associated with Child Growth and Household Drinking Water Quality than Access to a Private Toilet in Rural Mali. Environmental Science and Technology, 51(12), 7219–7227. https://doi.org/10.1021/acs.est.7b00178Huang, R. S. P., Johnson, C. L., Pritchard, L., Hepler, R., Ton, T. T., and Dunn, J. J. (2016). Performance of the Verigene enteric pathogens test, Biofire FilmArray gastrointestinal panel and Luminex xTAG gastrointestinal pathogen panel for detection of common enteric pathogens. Diagnostic Microbiology and Infectious Disease, 86(4), 336–339. https://doi.org/10.1016/j.diagmicrobio.2016.09.013Jung, Y. T., Lou, W., and Cheng, Y.-L. (2017). Exposure-response relationship of neighborhood sanitation and children’s diarrhea. Tropical Medicine and International Health, 22(7), 857–865. https://doi.org/10.1111/tmi.12886Keusch, G. T., Denno, D. M., Black, R. E., Duggan, C., Guerrant, R. L., Lavery, J. V., Nataro, J. P., Rosenberg, I. H., Ryan, E. T., Tarr, P. I., Ward, H., Bhutta, Z. A., Coovadia, H., Lima, A., Ramakrishna, B., Zaidi, A. K. M., Hay Burgess, D. C., and Brewer, T. (2014). Environmental Enteric Dysfunction: Pathogenesis, Diagnosis, and Clinical Consequences. Clinical Infectious Diseases, 59(suppl_4), S207–S212. https://doi.org/10.1093/cid/ciu485Knee, J., Sumner, T., Adriano, Z., Berendes, D., de Bruijn, E., Schmidt, W.-P., Nalá, R., Cumming, O., and Brown, J. (2018). Risk factors for childhood enteric infection in urban Maputo, Mozambique: A cross-sectional study. PLOS Neglected Tropical Diseases, 12(11), e0006956. https://doi.org/10.1371/journal.pntd.0006956Lazar, V., Ditu, L. M., Pircalabioru, G. G., Gheorghe, I., Curutiu, C., Holban, A. M., Picu, A., Petcu, L., and Chifiriuc, M. C. (2018). Aspects of gut microbiota and immune system interactions in infectious diseases, immunopathology, and cancer. In Frontiers in Immunology (Vol. 9, Issue AUG, p. 1). Frontiers Media S.A. https://doi.org/10.3389/fimmu.2018.01830Lin, A., Ali, S., Arnold, B. F., Rahman, M. Z., Alauddin, M., Grembi, J., Mertens, A. N., Famida, S. L., Akther, S., Hossen, M. S., … Luby, S. P. (2019). Effects of Water, Sanitation, Handwashing, and Nutritional Interventions on Environmental Enteric Dysfunction in Young Children: A Cluster-randomized, Controlled Trial in Rural Bangladesh. Clinical Infectious Diseases. https://doi.org/10.1093/cid/ciz291Liu, J., Platts-Mills, J. A., Juma, J., Kabir, F., Nkeze, J., Okoi, C., Operario, D. J., Uddin, J., Ahmed, S., Alonso, P. L., … Houpt, E. R. (2016). Use of quantitative molecular diagnostic methods to identify causes of diarrhea in children: a reanalysis of the GEMS case-control study. The Lancet, 388(10051), 1291–1301. https://doi.org/https://doi.org/10.1016/S0140-6736(16)31529-XMedgyesi, D., Sewell, D., Senesac, R., Cumming, O., Mumma, J., and Baker, K. K. (2019). The landscape of enteric pathogen exposure of young children in public domains of low-income, urban Kenya: The influence of exposure pathway and spatial range of play on multi-pathogen exposure risks. PLOS Neglected Tropical Diseases, 13(3), e0007292. https://doi.org/10.1371/journal.pntd.0007292Nhampossa, T., Mandomando, I., Acacio, S., Quinto, L., Vubil, D., Ruiz, J., Nhalungo, D., Sacoor, C., Nhabanga, A., Nhacolo, A., … Alonso, P. (2015). Diarrheal Disease in Rural Mozambique: Burden, Risk Factors and Etiology of Diarrheal Disease among Children Aged 0-59 Months Seeking Care at Health Facilities. Plos One, 10(5), e0119824. https://doi.org/10.1371/journal.pone.0119824Platts-Mills, J. A., Liu, J., and Houpt, E. R. (2013). New concepts in diagnostics for infectious diarrhea. Mucosal Immunology, 6(5), 876–885. https://doi.org/10.1038/mi.2013.50Platts-Mills, James A, Liu, J., Rogawski, E. T., Kabir, F., Lertsethtakarn, P., Siguas, M., Khan, S. S., Praharaj, I., Murei, A., Nshama, R., … Bessong, P. O. (2018). Use of quantitative molecular diagnostic methods to assess the aetiology, burden, and clinical characteristics of diarrhea in children in low-resource settings: a reanalysis of the MAL-ED cohort study. The Lancet Global Health, 6, e1309–e1318. https://doi.org/10.1016/S2214-109X(18)30349-8Prendergast, A. J., and Kelly, P. (2016). Interactions between intestinal pathogens, enteropathy and malnutrition in developing countries. Current Opinion in Infectious Diseases, 29(3), 229–236. https://doi.org/10.1097/QCO.0000000000000261Robertson, R. C., Manges, A. R., Finlay, B. B., and Prendergast, A. J. (2019). The Human Microbiome and Child Growth – First 1000 Days and Beyond. Trends in Microbiology, 27(2), 131–147. https://doi.org/10.1016/j.tim.2018.09.008Rogawski, E. T., Bartelt, L. A., Platts-Mills, J. A., Seidman, J. C., Samie, A., Havt, A., Babji, S., Rengifo Trigoso, D., Qureshi, S., Shakoor, S., … MAL-ED Network Investigators, the. (2017). Determinants and Impact of Giardia Infection in the First 2 Years of Life in the MAL-ED Birth Cohort. Journal of the Pediatric Infectious Diseases Society Giardia Epidemiology and Impact • JPIDS, 2017(6), 153–160. https://doi.org/10.1093/jpids/piw082Rogawski, E. T., Liu, J., Platts-Mills, J. A., Kabir, F., Lertsethtakarn, P., Siguas, M., Khan, S. S., Praharaj, I., Murei, A., Nshama, R., … Nyathi, E. (2018). Use of quantitative molecular diagnostic methods to investigate the effect of enteropathogen infections on linear growth in children in low-resource settings: longitudinal analysis of results from the MAL-ED cohort study. The Lancet Global Health, 6(12), e1319–e1328. https://doi.org/10.1016/S2214-109X(18)30351-6Rogawski McQuade, E. T., Platts-Mills, J. A., Gratz, J., Zhang, J., Moulton, L. H., Mutasa, K., Majo, F. D., Tavengwa, N., Ntozini, R., Prendergast, A. J., Humphrey, J. H., Liu, J., and Houpt, E. R. (2019). Impact of Water Quality, Sanitation, Handwashing, and Nutritional Interventions on Enteric Infections in Rural Zimbabwe: The Sanitation Hygiene Infant Nutrition Efficacy (SHINE) Trial. The Journal of Infectious Diseases. https://doi.org/10.1093/infdis/jiz179Satterthwaite, D., Beard, V. A., Mitlin, D., and Du, J. (2019). Untreated and Unsafe : Solving the Urban Sanitation Crisis in the Global South Solving the Urban Sanitation Crisis in the Global South Untreated and Unsafe : www.citiesforall.orgSpears, D., Ghosh, A., and Cumming, O. (2013). Open Defecation and Childhood Stunting in India: An Ecological Analysis of New Data from 112 Districts. PLoS ONE, 8(9), e73784. https://doi.org/10.1371/journal.pone.0073784Stolwijk, A. M., Straatman, H., and Zielhuis, G. A. (1999). Studying seasonality by using sine and cosine functions in regression analysis. Journal of Epidemiology and Community Health, 53(4), 235–238. https://doi.org/10.1136/jech.53.4.235Wardwell, L. H., Huttenhower, C., and Garrett, W. S. (2012). Current Concepts of the Intestinal Microbiota and the Pathogenesis of Infection. Curr Infect Dis Rep, 13(1), 28–34. https://doi.org/10.1007/s11908-010-0147-7.CurrentWolf, J., Hunter, P. R., Freeman, M. C., Cumming, O., Clasen, T., Bartram, J., Higgins, J. P. T., Johnston, R., Medlicott, K., Boisson, S., and Prüss-Ustün, A. (2018). Impact of drinking water, sanitation and handwashing with soap on childhood diarrheal disease: updated meta-analysis and meta-regression. Tropical Medicine and International Health, 23(5), 508–525. https://doi.org/10.1111/tmi.13051Zhuo, R., Cho, J., Qiu, Y. Y., Parsons, B. D., Lee, B. E., Chui, L., Freedman, S. B., and Pang, X. (2017). High genetic variability of norovirus leads to diagnostic test challenges. Journal of Clinical Virology, 96(May), 94–98. https://doi.org/10.1016/j.jcv.2017.10.003